# MMIU: MULTIMODAL MULTI-IMAGE UNDERSTANDING FOR EVALUATING LARGE VISION-LANGUAGE MODELS

**Fanqing Meng**[*,1,2]**, Jin Wang**[*,3,2]**, Chuanhao Li**[*,2]**, Quanfeng Lu**[1,2]**, Hao Tian**[4]
**Jiaqi Liao**[2]**, Xizhou Zhu**[5,2,4]**, Jifeng Dai**[5,2] **Yu Qiao**[2]**, Ping Luo**[2,3]**, Kaipeng Zhang**[2†]
**Wenqi Shao**[2†]

[1]Shanghai Jiao Tong University    [2]Shanghai AI Laboratory [3]The University of Hong Kong
[4]SenseTime Research    [5]Tsinghua University
Project Page: https://mmiu-bench.github.io

## ABSTRACT

The capability to process multiple images is crucial for Large Vision-Language Models (LVLMs) to develop a more thorough and nuanced understanding of a scene. Recent multi-image LVLMs have begun to address this need. However, their evaluation has kept pace with their development. To fill this gap, we introduce the Multimodal Multi-image Understanding (MMIU) benchmark, a comprehensive evaluation suite designed to assess LVLMs across a wide range of multi-image tasks. MMIU encompasses 7 types of multi-image relationships, 52 tasks, 77K images, and 11K meticulously curated multiple-choice questions, making it the most extensive benchmark of its kind. Our evaluation of nearly 30 popular LVLMs, including both open-source and proprietary models, reveals significant challenges in multi-image comprehension, particularly in tasks involving spatial understanding. Even the most advanced models, such as GPT-4o, achieve only 55.7% accuracy on MMIU. Through multi-faceted analytical experiments, we identify key performance gaps and limitations, providing valuable insights for future model and data improvements. We aim for MMIU to advance the frontier of LVLM research and development. We release the data and code at `https://github.com/OpenGVLab/MMIU`.

## 1 INTRODUCTION

The capability to process multiple images is crucial for multimodal large models, as a single image captures information from a specific angle and moment, limiting the model's ability to understand and reason about the entire scene (Song et al., 2024; Wang et al., 2024). Multiple images, on the other hand, provide rich information from different perspectives and time points, enabling the model to synthesize this data and achieve a more comprehensive understanding, such as analyzing consecutive images for action prediction (Lu et al., 2024b) or utilizing multi-view images in 3D navigation (Dai et al., 2017). The ability to process multiple images allows Large Vision-Language Models (LVLMs) to understand and handle complex visual tasks, thereby facilitating real-world applications.

Due to the great importance of multi-image understanding, recent LVLMs have improved such a capability by pre-training on various image-text interleaved data such as M4-Instruct (Li et al., 2024a), Mantis-Instruct (Jiang et al., 2024b), and OmniCorpus (Li et al., 2024b). However, the evaluation of multi-image LVLMs significantly lags behind their development. A good multi-image evaluation benchmark can help identify tasks that lead to poor performance and guide future model design data collection. Prior datasets such as LVLM-eHub (Xu et al., 2023) and MMBench (Liu et al.,

---

† Corresponding Authors: shaowenqi@pjlab.org.cn; zhangkaipeng@pjlab.org.cn

∗ Equal contribution

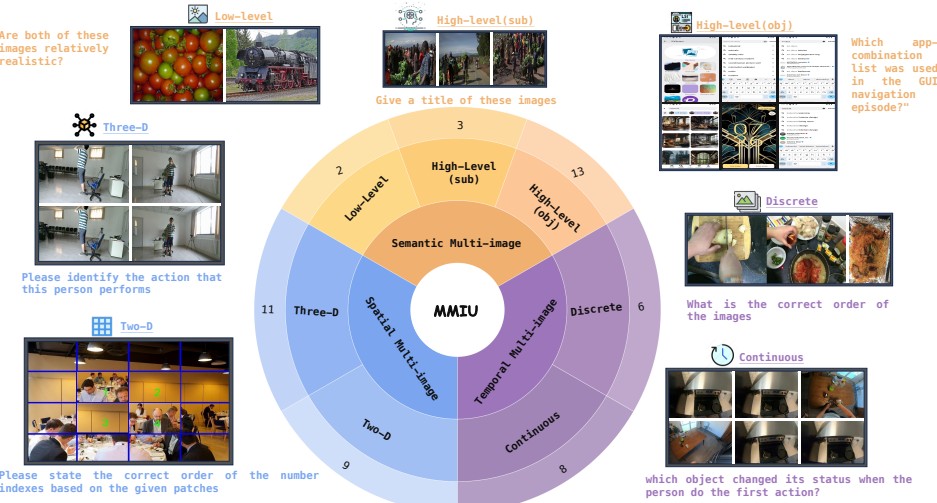

Figure 1: Visualization of MMIU. Our MMIU contains 77,659 images, 7 types of image relationships, and 5 image modalities, along with 11,698 multiple-choice questions, providing a comprehensive evaluation for 52 multi-image understanding tasks. Each example comes from a task from each multi-image relationship. We construct MMIU by adopting a top-down hierarchy where image relationships of interest are enumerated and multiple tasks are associated with each relationship.

2023) focus on single-image tasks (Xu et al., 2023), which cannot capture the complexity in multi-image scenarios. Although several recent benchmarks have attempted to evaluate the multi-image performance of LVLMs, they have limited coverage of multi-image tasks while capturing a few relationships between multiple images as shown in Table 1. For example, Video-MME (Fu et al., 2024a) focuses solely on temporal relationships and MUIRBENCH (Wang et al., 2024) does not consider spatial relationships between objects in multiple images, which is crucial in multi-image applications such as 3D navigation. Other works such as SlideQA (Tanaka et al., 2023) and MMMU (Yue et al., 2024) focus on understanding and reasoning within specific input types or disciplines, preventing them from providing a general evaluation for multi-image capabilities.

To build a comprehensive multi-image evaluation benchmark, we connect multi-image comprehension with manipulating information in working memory in cognitive psychology (Baddeley, 2000). As pointed out by Multiple Trace Theory (MTT) (Moscovitch et al., 2006), working memories are categorized into episodic memory which captures sequential information and can arrange events in the order they occur, semantic memory enabling concept comprehension, and spatial memory which helps understand spatial environments. Multiple images can be deemed as a visual memory. Understanding such a visual memory requires models to handle the semantic content, understand spatial relationships, and track temporal sequences of multiple images, closely mirroring human memory mechanisms. This inspires us to construct the evaluation benchmark to measure how well LVLMs tackle multi-image tasks from temporal, semantic and spatial perspectives.

This work introduces the Multimodal Multi-image Understanding (MMIU) benchmark, designed to comprehensively evaluate large visual language models (LVLMs) in multi-image task understanding. As shown in Table 1, we collect evaluation data through a top-down hierarchy, starting with the enumeration of image relationships spanning temporal, semantic, and spatial correspondences, and subsequently assigning multiple multi-image tasks to each relationship. The comprehensiveness of MMIU is twofold. First, it has the widest coverage of multi-image evaluation data to date, encompassing 7 types of multi-image relationships, 52 tasks (*e.g.* multi-view action recognition), 77k images, and 11.6k carefully curated multi-choice questions, which is 1.81 times larger than Miles-Bench (Song et al., 2024). Second, MMIU involves more diverse multi-image analysis tools than previous benchmarks, including performance comparison over image relationships, in- and out-of-domain task discovery by task map, and task learning difficulty by supervised fine-tuning (SFT). The multi-faceted analyses provide useful insights for model and data improvement.

We test 24 popular LVLMs on our MMIU, including closed-source models such as GPT-4o (OpenAI, 2024) and Gemini1.5 (Reid et al., 2024), and open-source models such as GLM4V (GLM et al., 2024) and InternVL-Chat (Chen et al., 2024b). These LVLMs contain both multi-image (support

Table 1: The comparison between MMIU and existing multi-image evaluation benchmarks including Video-MME (Fu et al., 2024a), MIRB (), MUIRBENCH (Wang et al., 2024), and MileBench (Song et al., 2024). We summarize the image relationships in previous benchmarks according to seven categories defined in Figure 1. 'Y&N' indicates that our MMIU comprises both answerable and unanswerable questions. I, T, V, D and P represent image, text, video, depth map and point cloud, respectively. Compared with prior datasets, MMIU involves massive test samples spanning 52 multimodal tasks and 5 modalities, and comprehensive multi-image analyses by image relationships, task map and supervised fine-tuning (SFT).

| Benchmark | Data Statistics | | | | | | Multi-image Analysis | | |
|---|---|---|---|---|---|---|---|---|---|
| | # Sample | # Imgs. | # Relation | # Task | # Modality | Answerable? | Relation | Task Map | SFT |
| Video-MME | 2.7K | - | *1* | *30* | T,V | Y | - | ✗ | ✗ |
| MIRB | 0.9K | 3.5k | 3 | 11 | I,T,V | Y | ✓ | ✗ | ✗ |
| MUIRBENCH | 2.6K | 11k | 4 | 12 | I,T,V | Y&N | ✓ | ✗ | ✗ |
| MileBench | 6.4K | 97k | 4 | 28 | I,T,V | Y | ✓ | ✗ | ✗ |
| **MMIU** | 11.6K | 77k | 7 | 52 | I,T,V,P,D | Y&N | ✓ | ✓ | ✓ |

multi-image input) and single-image (support only single-image input) models. For single-image models, we employ image concatenation to obtain the evaluation performance. The experimental results show that even the most advanced model, GPT-4o (OpenAI, 2024), achieves only 55.7% accuracy on MMIU, highlighting the inherent difficulty of these tasks. Other than the diverse analytical tools in Table 1, we conduct ablation studies to investigate the impact of unanswerable questions and multi-image concatenation methods on model performance. We summarize our findings as follows:

- The best-performing model for multi-image tasks is GPT-4o, with InternVL2-pro (Chen et al., 2024b) being the strongest among open-source models. The best closed-source model GPT-4o leads the best open-source model InternVL2 by a large margin, (*i.e.* 5.4% accuracy). However, GPT-4o achieves only 55.7% accuracy on MMIU, indicating a substantial challenge in our benchmark.

- Some powerful LVLMs like InternVL1.5 (Chen et al., 2024b) and GLM4V (GLM et al., 2024) whose pre-training data do not contain multi-image content even outperform many multi-image models which undergo multi-image supervised fine-tuning (SFT), indicating the strong capacity in single-image understanding is the foundation of multi-image comprehension.

- By comparing performance at the level of image relationships, we conclude that LVLM excels at understanding semantic content in multi-image scenarios but has weaker performance in comprehending temporal and spatial relationships in multi-image contexts.

- The analysis based on the task map reveals that models perform better on high-level understanding tasks such as video captioning which are in-domain tasks, but struggle with 3D perception tasks such as 3D detection and temporal reasoning tasks such as image ordering which are out-of-domain tasks.

- By task learning difficulty analysis, tasks involving ordering, retrieval and massive images cannot be overfitted by simple SFT, suggesting that additional pre-training data or training techniques should be incorporated for improvement.

In summary, this paper makes three key contributions. First, we introduce and open-source the Multimodal Multi-image Understanding (MMIU) benchmark, a comprehensive evaluation suite that addresses various complex multi-image tasks, thereby filling a critical gap in multi-image comprehension. Second, our evaluation results demonstrate that current large visual language models (LVLMs), including proprietary models like GPT-4o, encounter significant challenges in solving multi-image tasks, particularly those involving spatial understanding. Third, we conduct multi-faceted analytical experiments, shedding light on the limitations and performance gaps of current models from various perspectives. We hope that MMIU will push the boundaries of LVLM research and development, bringing us closer to the realization of advanced multimodal multi-image user interactions.

## 2 RELATED WORK

### 2.1 LARGE VISION-LANGUAGE MODELS

With the advancements in large language models (LLMs) (Touvron et al., 2023; Jiang et al., 2024a), a series of studies have begun exploring multimodal LLMs capable of simultaneously interpreting visual and linguistic information. Through visual pre-training and instruction fine-tuning, LVLMs have demonstrated outstanding performance in understanding multimodal image-text inputs (Li et al., 2024a; Lu et al., 2024a; Bai et al., 2023). However, most LVLM training data consist primarily of single image-text pairs or pure text data, limiting their ability to comprehend multi-image inputs. Therefore, researchers have considered using large-scale interleaved image-text corpora, such as MMC4 (Zhu et al., 2024) and Omnicorpus (Li et al., 2024b), during the pre-training phase of LVLMs. This approach has led to the development of models like DeepSeek-VL (Lu et al., 2024a) and Idefics (Laurençon et al., 2024), which exhibit notable performance in multi-image tasks. Building on this foundation, recent studies have applied instruction tuning with extensive multi-image data, resulting in models that handle multi-image tasks more effectively while utilizing fewer resources. Notable examples of these advancements include Mantis (Jiang et al., 2024b) and LLaVA-Next-interleave (Li et al., 2024a). Nonetheless, the evaluation of these models' capabilities in handling multiple images has mainly been qualitative, and quantitative assessments of different models' performance across a broad range of multi-image tasks remain insufficiently explored.

### 2.2 LARGE VISION-LANGUAGE MODELS BENCHMARKS

Benchmarking large vision-language models (LVLMs) is crucial for identifying model limitations and guiding their development (Xu et al., 2023; Ying et al., 2024; Liu et al., 2023). Despite the existence of numerous benchmarks aimed at evaluating the perception or reasoning abilities of LVLMs, most of these benchmarks focus solely on single-image scenarios. Although some benchmarks include multi-image examples (Jiang et al., 2024b; Fu et al., 2024a), they usually address limited capabilities. For instance, MANTIS-Eval (Jiang et al., 2024b) focuses on assessing a model's ability to perceive size, while Video-MME (Fu et al., 2024a) emphasizes image sequences and their temporal relationships. Recently, researchers have been dedicated to developing more holistic multi-image evaluation benchmarks, such as MileBench (Song et al., 2024) and MUIRBench (Wang et al., 2024), to provide a more thorough assessment of multi-image cognition. However, these benchmarks fall short in terms of task depth and breadth. For instance, MILEBENCH (Wang et al., 2024) provides a relatively comprehensive multi-image evaluation but lacks important multi-image tasks such as 3D spatial understanding and low-level semantics, which are essential for drawing complete conclusions. In contrast, MMIU offers a benchmark that combines both task depth and breadth, covering a wider range of image relationships, task types, and image categories. This enables a more comprehensive assessment of model capabilities.

## 3 MMIU

This section presents the proposed MMIU benchmark. MMIU is a comprehensive evaluation dataset encompassing 11K multi-choice questions for multi-image comprehension. We first give a brief overview of MMIU in Section 3.1. Then, we describe the construction process of MMIU in Section 3.2. We provide two versions of MMIU: test and testmini, with the latter being 1/10th the size of the former for quick testing, which has 1040 samples. In this paper, we primarily conduct experiments and analyses on the test set, while results on testmini are recorded in the Table 15 in the appendix for comparison.

Table 2: Key statistics for the MMIU

| Statistic | Number |
|---|---|
| Total samples | 11698 |
| Total images | 77659 |
| Total tasks | 52 |
| Img. relations | 7 |
| Average images | 6.64 |
| Average question words | 27.9 |
| Range of images | 2~32 |
| **Image Num Level** | **Number** |
| - Few (2~5) | 7446 |
| - Medium (6~15) | 2574 |
| - Many (16~32) | 1666 |
| **Unanswerable set** | **Percentage** |
| - Replace keyword | 21% |
| - Replace answer image | 47% |
| - Replace other images | 53% |
| - Shuffle all images | 53% |
| - Irrelevant question/image set | 79% |

### 3.1 BENCHMARK OVERVIEW

MMIU is designed to measure multi-image understanding for LVLMs. It has two advantages compared with pre-

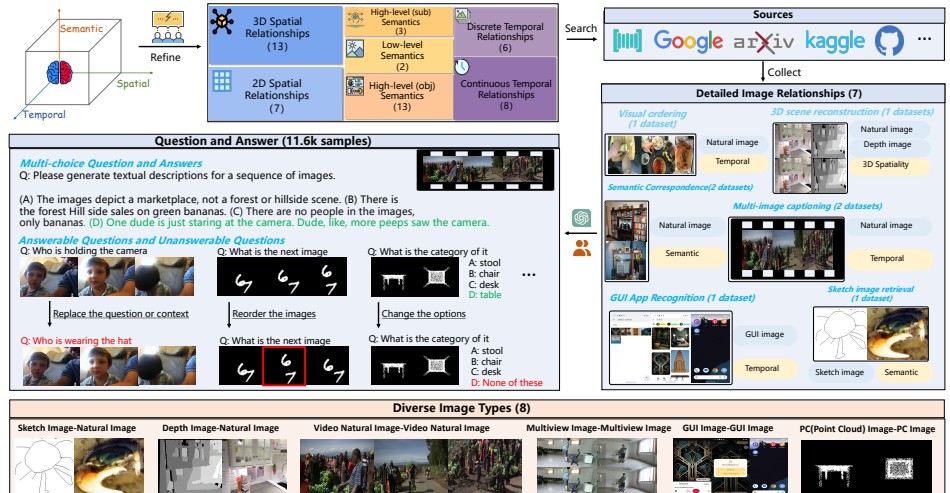

Figure 2: An illustration of our data collection process. First, we refine multi-image tasks and collect task data based on cognitive psychology. Then, we standardize these datasets into a uniform format—metadata. Next, we generate multiple-choice samples with answerable and unanswerable questions from the metadata using either manually designed rules or GPT-4o. Our benchmarks include capability evaluations across various image types.

vious multi-image evaluation benchmarks as illustrated in

Table 1. First, MMIU provides a comprehensive evaluation by encompassing massive test samples spanning various multi-image tasks and image relationships. Specifically, MMIU consists of 77,659 images and 11,698 multi-choice questions (1.81 times more than MileBench (Song et al., 2024) which previously had the most multi-image test samples) with an average of 6.64 images per instance. It tests 7 distinctive multi-image relationships covering 52 diverse multi-image tasks, 1.73 times more than VideoMME (Fu et al., 2024a) which previously contained the most multi-image tasks. In addition, we also create an unanswerable set comprising 19 tasks with each task containing 40 questions, considering that LVLMs cannot answer all questions in real scenarios. More detailed statistics of MMIU can be found in Table 2. The diverse evaluation data requires the model to be capable enough to deeply understand semantical, temporal, and spatial clues in multi-images with various input types (Figure 2).

Second, MMIU offers thorough analyses in multi-image understanding by utilizing multi-faceted analytical tools. 1) Thanks to the top-down hierarchy in collecting data, MMIU can compare performance across image relationships. 2) The extensive coverage of multi-image tasks enables evaluating on a task map, facilitating the discovery of in- and out-of-domain tasks. 3) The evaluation samples can be adapted to multi-image instruction tuning data. By SFT, the task learning difficulty can be acquired, which is crucial for the practitioner to improve the model and data.

## 3.2 DATA CURATION PROCESS

Multi-image understanding is essential for Large Vision-Language Models (LVLMs) due to its prevalence in real-world applications. We treat a sequence of images as visual memories whose semantic, temporal, and spatial segments are crucial in retrieving information (Moscovitch et al., 2006). As shown in Figure 2, our Multi-Modal Image Understanding (MMIU) benchmark is constructed using a top-down approach. We first categorize multi-image relationships into semantic, spatial, and temporal types, further divided into seven subtypes. We then collect data for each relationship type and standardize its format. Finally, we create multiple-choice questions based on these relationships to evaluate LVLMs' multi-image understanding capabilities.

**Relationships → Tasks.** We categorize multi-image relationships into semantic, spatial, and temporal aspects. Semantic relationships are further divided into: *1) Low-level semantic relationships* comparing visual features like illumination, quality, and saturation; *2) High-level (objective) relationships* involving objects, attributes, and interactions (*e.g.*, a person hitting a ball, a person catching a ball); and *3) High-level (subjective) relationships* such as thematic, cultural, and emotional asso-

ciations (*e.g.*, the emotions expressed in these images). Temporal relationships are refined into: *4) Continuous temporal relationships* for video frame sequence tasks, and *5) Discrete event sequence relationships* for understanding multi-step tutorials. Spatial relationships are categorized as: *6) 2D spatial relationships* including rotation, translation, and symmetry; and *7) 3D spatial relationships* involving different camera perspectives and depth variations. Detailed information on each image relationship is provided in Appendix A, with corresponding multi-image tasks presented in Table 4 of the Appendix.

**Tasks → Data.** We conduct extensive searches for relevant datasets using resources like Google, Paper With Code, and Kaggle, guided by the proposed tasks. After downloading, we thoroughly evaluate each dataset's appropriateness for the specific task, ensuring usability and relevance. We then organize the data into a standardized metadata format, which includes task description, question, answer, input context, and images for each sample. This format facilitates the creation of visual questions and answers. We manually verify the accuracy of this information and its convertibility into a multiple-choice question format. For efficient evaluation, we limit each task to a maximum of 200 randomly selected samples, except for tasks with insufficient data. The detailed description of this metadata format is provided in Appendix A.4.

**Question and Answer Generation.** For each subtask, we create multiple-choice visual questions (with a maximum of eight options, depending on the task), with the choices and answers derived from their metadata. Specifically, depending on the task, we either manually design rules or use GPT-4o (OpenAI, 2024) with carefully crafted prompts to ensure efficient and high-quality generation. For example, in 3D question-answering tasks, we instruct GPT-4o to generate plausible but incorrect options based on the question and the correct answer. For image retrieval tasks, we randomly select incorrect images from the metadata as the wrong options. Additionally, we select 19 tasks and create 40 unanswerable samples for each task to construct an unanswerable set for robust evaluation. More details in unanswerable question generation are provided in Appendix A.4.

**Challenges.** Designing accurate question templates: Questions must provide all necessary information for LVLMs to derive correct answers. For example, in 3D object detection, questions should include detailed camera pose information and specify the coordinate system for detected objects. Obtaining and verifying correct answers: This is particularly challenging for 3D spatial relationships. In 3D pose estimation, relative camera pose between images is not inherently provided in datasets like ScanNet (Dai et al., 2017). We address this by: a) Transforming original camera poses to relative poses through matrix multiplication. b) Carefully examining the correctness by applying relative poses to image pairs and ensuring proper correspondence matching. These challenges highlight the significant effort required to establish MMIU as a comprehensive multi-image evaluation benchmark.

## 4 EXPERIMENT

This section first introduces the experimental setup in Section 4.1, including the testing methods and models used. Following this, we present the main results and multi-faceted analyses in Section 4.2 and Section 4.3, respectively. Ablation studies are included in Section 4.5. We put more detailed information and analysis in the Appendix B.

### 4.1 EXPERIMENT SETUP

**LVLM Models.** Specifically, we select 9 closed-source models: GPT-4o (OpenAI, 2024), and the other 3 models. Additionally, we evaluate 19 open-source models that support multiple image inputs: Mantis (Jiang et al., 2024b), InternVL2 (Chen et al., 2024b), LLaVA-Next-Interleave (Li et al., 2024a), and other 16 models. Furthermore, we include 7 models that only support single image input including LLaVA-V1.5-7B (Liu et al., 2024a), Monkey-Chat (Li et al., 2024c) and other 5 models. Finally, we also evaluate the performance of 4 LLMs, including two approaches: direct questioning without images and questioning after generating image captions using GPT-4o. The detailed description of all models can be found in Appendix B.1.

**Evaluation Method.** With OpenCompass (Contributors, 2023), we first match the model's response to the corresponding options. If a match cannot be made, we mark it as Z (Yue et al., 2023). The accuracy is used as the metric. Specifically: 1) For cases where the input token is too long for the tested model, we randomly sample images until it can be tested. 2) For closed-source models, if the model refuses to respond due to copyright issues with the images, we discard those samples.

Table 3: Quantitative results across 7 image relationships on test set are summarized. Accuracy is the metric, and the Overall score is computed across all tasks. The maximum value of each task is bolded. The caption used for LLMs is generated by GPT-4o.

| Model | Overall (testmini) | Overall | Discrete | Continuous | Low-level | High-level-sub | High-level-obj | Two-D | Three-D |
|---|---|---|---|---|---|---|---|---|---|
| Frequency | 30.9 | 31.5 | 29.5 | 29.5 | 38.1 | 29.6 | 36.7 | 27.8 | 30.2 |
| Random | 27.1 | 27.4 | 22.1 | 25.4 | 33.7 | 20.7 | 32.8 | 24.3 | 28.4 |
| *Closed-source LVLMs* | | | | | | | | | |
| GPT-4o | **55.6** | **55.5** | **58.2** | **53.7** | **84.0** | **69.2** | 57.5 | 41.7 | **55.4** |
| Claude3.5 | 54.3 | 53.4 | 55.3 | 47.9 | 77.2 | 64.8 | 64.5 | 41.9 | 45.1 |
| Gemini1.5 | 54.5 | 53.4 | 54.2 | 50.1 | 76.1 | 63.9 | **64.9** | **43.3** | 43.0 |
| Gemini1.0 | 41.2 | 40.2 | 45.8 | 49.8 | 57.9 | 36.7 | 36.7 | 29.7 | 36.7 |
| *Adequate Multi-Image SFT LVLMs* | | | | | | | | | |
| Mantis-idefics2-8B | 45.3 | 45.6 | 37.3 | 43.4 | 58.4 | 54.8 | 56.4 | 37.8 | 40.4 |
| Mantis-SigCLIP-8B | 41.8 | 42.6 | 37.2 | 39.3 | 69.5 | 46.2 | 52.9 | 30.2 | 40.2 |
| LLaVA-Next-Interleave-7B | 33.5 | 32.4 | 35.3 | 30.7 | 33.7 | 35.7 | 33.3 | 34.7 | 27.4 |
| xGen-MM-Interleaved-4B | 28.2 | 28.7 | 28.1 | 29.2 | 37.1 | 39.8 | 24.2 | 30.9 | 27.6 |
| *Multi-Image input LVLMs* | | | | | | | | | |
| InternVL2-Pro | 49.8 | 50.3 | 53.8 | 46.3 | 72.7 | 70.6 | 58.5 | 38.1 | 42.1 |
| Internvl1.5-chat | 38.7 | 37.4 | 43.6 | 46.4 | 42.9 | 59.1 | 26.0 | 33.6 | 37.0 |
| InternVL2-8B | 34.0 | 34.8 | 34.2 | 43.4 | 36.7 | 47.3 | 32.1 | 30.0 | 32.2 |
| Mini-InternVL-chat-1.5-4B | 32.5 | 32.1 | 30.6 | 42.2 | 35.4 | 47.2 | 29.2 | 27.2 | 30.5 |
| Mini-InternVL-chat-1.5-2B | 31.8 | 30.5 | 33.1 | 38.6 | 30.9 | 37.6 | 28.7 | 27.4 | 25.7 |
| idefics2-8B | 27.2 | 27.8 | 22.9 | 19.3 | 42.4 | 45.2 | 26.8 | 33.4 | 25.7 |
| DeepSeek-VL-7B | 24.6 | 24.6 | 16.4 | 10.3 | 39.1 | 32.3 | 34.2 | 32.9 | 16.7 |
| xGen-MM-Single-4B | 25.4 | 24.5 | 25.8 | 25.7 | 23.9 | 28.8 | 24.4 | 24.5 | 22.1 |
| XComposer2-7B | 23.4 | 23.5 | 31.9 | 31.6 | 23.4 | 34.3 | 20.0 | 18.7 | 18.0 |
| DeepSeek-VL-1.3B | 23.8 | 23.2 | 14.6 | 9.2 | 33.3 | 24.9 | 30.8 | 32.7 | 19.0 |
| flamingov2 | 22.7 | 22.3 | 20.8 | 19.5 | 29.6 | 24.6 | 26.9 | 17.2 | 21.7 |
| XComposer2-1.8B | 22.0 | 21.9 | 29.4 | 32.9 | 22.5 | 36.2 | 15.3 | 20.9 | 14.6 |
| Qwen-chat | 18.0 | 15.9 | 14.7 | 19.5 | 22.3 | 21.3 | 14.8 | 10.5 | 17.1 |
| idefics-9b-instruct | 13.2 | 12.8 | 23.6 | 7.2 | 11.6 | 27.0 | 12.3 | 12.2 | 8.7 |
| Qwen-Base | 4.8 | 5.2 | 13.2 | 2.6 | 5.3 | 10.1 | 4.6 | 2.8 | 3.8 |
| *Single-Image input LVLMs* | | | | | | | | | |
| GLM-4v-9b | 26.6 | 27.0 | 23.3 | 30.9 | 45.4 | 43.6 | 29.8 | 20.9 | 20.1 |
| LLaVA-next-vicuna-7b | 22.9 | 22.2 | 21.7 | 26.0 | 24.0 | 32.5 | 20.7 | 21.3 | 19.4 |
| MiniCPM-Llama3-V-2-5 | 21.7 | 21.6 | 22.2 | 30.8 | 30.5 | 43.3 | 15.7 | 12.7 | 21.4 |
| LLaVA-v1.5-7b | 18.7 | 19.2 | 20.4 | 16.6 | 20.5 | 23.8 | 19.1 | 19.2 | 19.1 |
| sharegpt4v-7b | 18.4 | 18.5 | 19.4 | 17.5 | 22.5 | 26.1 | 15.4 | 19.9 | 18.5 |
| sharecaptioner | 16.4 | 16.1 | 15.9 | 9.8 | 33.4 | 30.0 | 15.6 | 16.4 | 14.0 |
| monkey-chat | 14.3 | 13.7 | 10.7 | 13.0 | 14.7 | 17.0 | 12.9 | 15.4 | 14.6 |
| *Pure text LLMs* | | | | | | | | | |
| GPT-4o | 21.8 | 23.1 | 33.3 | 24.1 | 16.2 | 40.8 | 10.4 | 20.0 | 30.5 |
| + Caption | 48.9 | 48.7 | 60.8 | 49.1 | 66.7 | 58.9 | 52.1 | 32.8 | 44.6 |
| Qwen2.5 | 30.1 | 30.9 | 31.7 | 26.2 | 31.2 | 37.5 | 35.4 | 30.6 | 26.8 |
| + Caption | 41.1 | 40.3 | 50.0 | 45.6 | 61.3 | 51.7 | 42.7 | 27.8 | 31.8 |
| Llama3.1 | 30.1 | 29.5 | 32.5 | 26.9 | 38.7 | 30.0 | 30.2 | 24.4 | 31.4 |
| + Caption | 38.3 | 39.2 | 40.9 | 45.4 | 51.7 | 50.3 | 38.9 | 31.2 | 35.4 |
| InternLM2.5 | 32.1 | 31.5 | 39.2 | 28.8 | 38.7 | 42.2 | 28.7 | 27.8 | 35.9 |
| + Caption | 39.8 | 38.2 | 36.7 | 45.0 | 45.0 | 52.5 | 41.3 | 30.0 | 31.8 |

The detailed setup can be found in Appendix B.2. 3) To prevent the model consistently selecting a same option (e.g. single-image input models), we shuffle the original options and retest. A result is considered correct only if both tests yield the correct answer. 4) For both the answerable set and the unanswerable set, we randomly shuffle the options to ensure that the correct answer's position does not introduce any position bias.

## 4.2 MAIN RESULTS

As shown in Table 3, we report the average accuracy of all models across 7 image relationships alongside Random Choice and Frequent Choice baselines, with "overall" representing the average accuracy on all tasks. We record the model's performance across all tasks in Table 14 in the Appendix. Specifically, we have the following findings.

**Multi-image tasks remain challenging.** GPT-4o leads with 55.7% accuracy, followed by Gemini1.5-Flash and Claude3.5-Sonnet at 53.4%. Among open-source models, InternVL2-Pro tops at 50.3%, outperforming Gemini1.0 Pro Vision. *A 5.4% accuracy gap exists between closed-source and open-source models in multi-image comprehension.* Notably, in single-image tasks, open-source models like InternVL2-Pro match or exceed closed-source models such as GPT-4o (Yue et al., 2023; Liu et al., 2023; Ying et al., 2024).

**The strong capability in single-image understanding is the foundation of multi-image comprehension.** Several advanced models such as InternVL1.5-Chat * which have been trained with only single-image data can achieve good performance in MMIU. For instance, GLM4V reaches 37.4%

---

*Notice that although InternVL1.5-chat supports multiple image inputs, its training phase did not incorporate multi-image data.

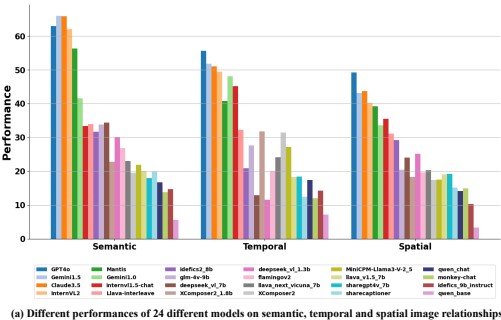 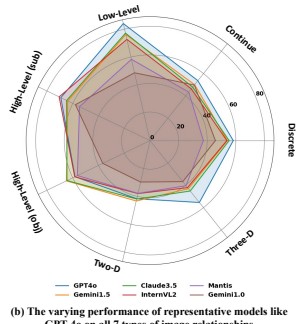

(a) Different performances of 24 different models on semantic, temporal and spatial image relationships

(b) The varying performance of representative models like GPT-4o on all 7 types of image relationships.

Figure 3: (a): The average performance comparison of representative models on 3 main image relationships. (b): The average performance comparison of representative models on 7 specific image relationships.

accuracy, surpassing multi-image models LLaVA-Next-Interleave and Idefics2. Such success stems from its powerful capability in single-image multimodal understanding. Besides, GLM-4V also outperforms many multi-image models such as DeepSeek-VL. This is because GLM-4V supports an ultra-high resolution of 1120*1120, allowing it to understand concatenated images and to reason. For instance, in the video-captioning task, its accuracy reaches 76%.

**Adequate multi-image supervised fine-tuning (SFT) can improve the performance of models on multi-image tasks.** Notably, we have observed that many models trained extensively with multi-image data during the pre-training phase did not achieve satisfactory results, such as idefics2 and DeepSeek-VL. However, Mantis and LLaVA-Next-Interleave stand out among all models. Their common feature is extensive multi-image instruction fine-tuning during the SFT phase. For instance, although idefics2 is trained with a large amount of multi-image data during the pre-training phase, it is trained by a few multi-image data during the SFT phase. Mantis, after performing multi-image SFT on the basis of idefics2, achieved a 17.8% accuracy improvement.

### 4.3 MULTITASK ANAYSIS

#### 4.3.1 PERFORMANCE ACROSS IMAGE RELATIONSHIPS

As shown in Figure 3, models exhibit varying capabilities across different image relationships. More detailed visualizations can be found in Figure 8 in the Appendix. In general, LVLMs excel at understanding semantic content in multi-image scenarios, perform moderately in temporal tasks, and obtain the worst performance incomprehending spatial relationships in multi-image contexts.
**1) In semantic relationships,** models generally perform well on multi-image semantic tasks involving low-level relationships. However, they struggle with high-level tasks, for subjective tasks such as Causality Reasoning and Emotion Recognition, which require the identification and reasoning of implicit visual information, highlighting a gap between model performance and human visual cognition. As for objective tasks such retrieval tasks, most models fail to tackle them. **2) In temporal relationships,** models can handle discrete and continuous temporal relationships relatively well but show mediocre performance on reasoning-intensive multi-image tasks. For instance, in sorting tasks, GPT-4o achieves only 28% and 21.5% accuracy in temporal ordering and visual ordering tasks, respectively. **3) In spatial relationships,** we find that models struggle with understanding both 2D and 3D positional relations. This is consistent with the observation in the previous single-image evaluation benchmark Ying et al. (2024) where they find that LVLMs fall short in localization and detection tasks requiring spatial reasoning. The tasks involving spatial relationships in MMIU become more challenging because models need to gather spatial information in multiple images and to reason.

#### 4.3.2 ANALYSIS ON THE TASK MAP

Task map is an effective tool for multi-task analysis Ying et al. (2024); Ilharco et al. (2023). Thanks to extensive coverage of multi-image tasks in MMIU, we build a task map to analyze the relationships between different tasks, allowing us to identify in- and out-of-domain tasks for current LVLMs. Following MMT-Bench Ying et al. (2024), we use QwenVL-chat to construct a task map where the

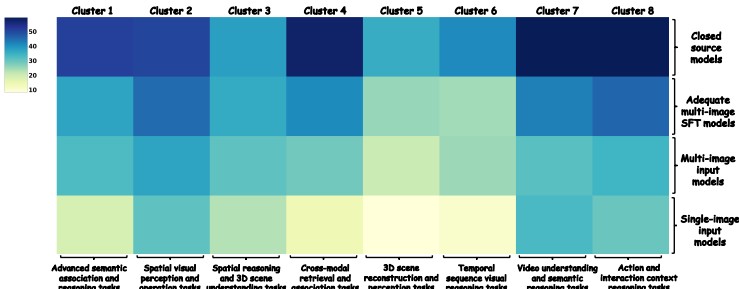

Figure 4: Visualization of 4 categories of LVLMs performance across various clusters.

distance between two tasks is given and cluster all tasks into 8 categories. Detailed construction process, the description of the task map, and more analysis can be found in Appendix C.

**Tasks involving recognition or captioning are in-domain tasks** which can be handled by most current multimodal large models. For multi-image tasks, models generally struggle to achieve satisfactory results, obtaining good performance on a limited number of tasks. Specifically, for tasks in clusters 7, 8, and some tasks in cluster 2, which involve recognition or captioning (e.g., video captioning, action recognition), models perform relatively well. This is because these multi-image tasks focus on overall image perception, requiring less comparison and reasoning between images.

**Tasks involving temporal ordering and 3D spatial reasoning are out-of-domain Tasks** where most models perform poorly. Specifically, models struggle with tasks in clusters 4, 5, and 6. Clusters 4 and 6 involve modelling semantic relationships or sequential order among multiple images, requiring memorizing detailed long-context content and strong reasoning skills. Most LVLMs underperform on these tasks such as temporal ordering tasks). Tasks in cluster 5 pertain to 3D visual tasks such as 3D detection and restruction. This may be due to the lack of 3D vision-language data in training LVLMs.

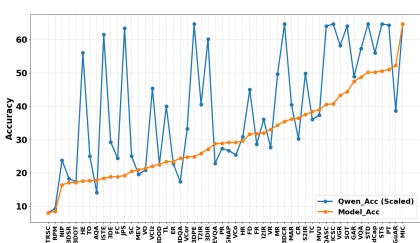

Figure 5: The performance of $Acc_{Model}$ and $Acc_{SFT}$ across different tasks, sorted by $Acc_{Model}$ in descending order.

### 4.3.3 TASK LEARNING DIFFICULTY

We analyze task learning difficulty by SFT with all evaluation samples in MMIU being instruction tuning data. In this way, we can identify tasks which cannot be improved by simple SFT. To this end, we fine-tune QwenVL-chat on each task for 20 epochs and obtain the accuracy of QwenVL-chat on each task, denoted as $Acc_{SFT}$. The lower accuracy reflects the larger fitting difficulty of the task. Meanwhile, we also obtain the average accuracy of all tested models on each task, denoted as $Acc_{Model}$. This accuracy reflects the difficulty current models face in handling these tasks.

As shown in Figure 5, we find that the Spearman correlation coefficient between $Acc_{SFT}$ and $Acc_{Model}$ is 0.66, indicating a high correlation. This suggests that both measures can reflect task difficulty to some extent. More importantly, we need to focus on tasks where both $Acc_{SFT}$ and $Acc_{Model}$ are low. A low $Acc_{SFT}$ indicates that the task is difficult to overfit even with SFT, suggesting that additional pre-training data or training techniques might be necessary. These tasks include 1) Ordering and retrieval tasks, which require strong memory and reasoning abilities—capabilities that are generally weak in large multimodal models. 2) Tasks involving a large number of images, such as EVQA, MEV, and GNAP, require models to support longer context lengths and possess strong memory capabilities. This indicates that future model designs should consider the ability to handle long contexts and emphasize the inclusion of multi-image data during the pre-training.

### 4.4 ERROR ANALYSIS

We analyzed error patterns of some Large Vision-Language Models (LVLMs): GPT-4o, Claude 3.5 Sonnet, and InternVL2-Pro on MMIU. Our approach involved systematically sampling up to 5 incorrect answers per subtask for each model. Task-specific experts then examined these errors to

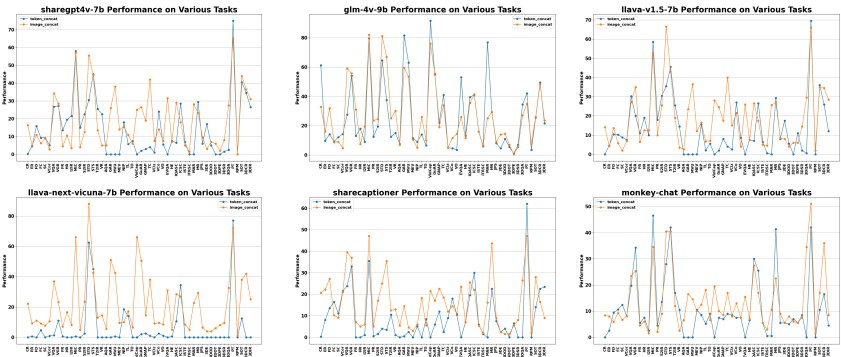

Figure 7: Comparison of the performance of different single-image models on various tasks in the MMIU when tested with image stitching or visual token stitching methods.

identify their root causes. The error distribution is shown in Figure 6. For more detailed analysis and description on the four error categories, refer to Appendix B.4.

Specifically, GPT-4o's errors are primarily perception (39%) and reasoning (42%), indicating room for improvement in visual capabilities and logical reasoning based on visual cues. However, it excels in following instructions due to strong semantic understanding and long context handling. GPT-4o shows 17% lack of capacity errors, mainly in 3D and 2D visual tasks, suggesting its visual capabilities are still insufficient in these areas. Similarly, as a representative closed-source model, Claude 3.5 Sonnet exhibits an error distribution similar to GPT-4o,

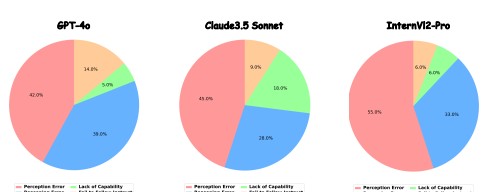

Figure 6: Distribution of error types for GPT-4o, Claude3.5 Sonnet and InternVL2-Pro.

but with more "Lack of capacity" errors. We find this is mainly due to Claude's weaker visual capabilities compared to GPT-4o. InternVI2-Pro, as an open-source model, achieves excellent results on MMIU. However, it shows more perception errors, indicating that open-source models may be weaker in multi-image and text alignment compared to closed-source models.

## 4.5 ABLATION STUDY

Here we have conduct a detailed ablation study for different test methods. For more results, please refer to Appendix B.5.

**Impact of Different Testing Methods on Model Performance.** For single-image input models handling multi-image tasks, one approach is to concatenate the images into a single image and feed it to the models. Besides, we explore an alternative method: concatenating all output visual embeddings before feeding them into LLMs. As shown in Figure 7, we observe that for these models, testing using concatenated visual tokens does not perform better than directly concatenating images. This is especially true for the LLaVA series, where concatenating images significantly outperform concatenating visual tokens. In contrast, GLM-4V exhibits relatively consistent performance under both testing methods.

## 5 CONCLUSION

In this paper, we present MMIU, a benchmark dedicated to comprehensively evaluating the performance of LVLMs on multi-image tasks. MMIU includes seven types of image relationships, such as 3D spatial relations, 52 tasks, and various image modalities, filling a gap in this field. We test 24 popular LVLMs on MMIU and analyzed the results using various analytical tools, including task maps. The experimental results indicate that current models, including GPT-4, struggle to handle complex multi-image tasks. We hope that MMIU will promote the development of more generalized capabilities in future models within the multi-image domain.

## 6 ACKNOWLEDGEMENT

This paper is partially supported by the National Key R&D Program of China (NO.2022ZD0160102, NO.2022ZD0161000), and the General Research Fund of Hong Kong No.17200622 and 17209324. This work was done during his internship at Shanghai Artificial Intelligence Laboratory.

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

## A  MMIU DETAILS

### A.1  MULTI-IMAGE RELATIONS

Overall, inspired by cognitive psychology, MMIU encompasses three broad types of image relationships: semantic, temporal, and spatial. Furthermore, we refine all detailed types as follows:

- Low-level semantic relations: This mainly refers to multi-image comparisons of low-level visual features, such as lighting, quality, and saturation.
- High-level semantic relationship (objective): This refers to the objective assessment of high-level image features, such as objects (e.g., dog), attributes (e.g., number), and relationships between objects (e.g., person serving a ball, person catching a ball).
- High-level semantic relationship (subjective): This refers to the subjective assessment of high-level image features, such as thematic association (e.g., determining whether a set of images conveys a theme) or emotional association (e.g., identifying the emotions expressed in the images).
- Discrete time (event) temporal relationship: Compared to continuous video frames, this mainly refers to discrete event/time sequence image tasks, such as the analysis and reasoning of multi-step tutorials.
- Continuous time temporal relationship: It mainly refers to video frame sequence tasks, including perception (e.g., action classification) and reasoning (e.g., action prediction).
- Two-dimensional spatial relationship: This mainly refers to two-dimensional spatial multi-image relationships, such as rotation, translation, and symmetry.
- Three-dimensional spatial relationship: This mainly refers to multi-image relationships in three-dimensional spatial contexts, such as different perspectives and depth variations.

### A.2  HIERARCHICAL STRUCTURE OF MMIU

**Image relationships and corresponding tasks.** We present all 7 types of image relationships in MMIU, totaling 52 tasks. Table 4 includes the distribution of tasks for each type of image relationship.

Table 4: Details of tasks classified by image relationship of our MMIU.

| Image Relationship | Task | # Number |
|---|---|---|
| Two-dimensional spatial relationship | ravens-progressive-matrices, jigsaw-puzzle-solving, image-captioning-with-spatial-context, icon-question-answering-with-spatial-context, image-text-retrieval-with-spatial-context, image-spatial-transformation-estimation, homography-estimation, point-tracking, single-object-tracking | 9 |

Table 4 – continued from previous page

| Image Relationship | Task | # Number |
|---|---|---|
| Three-dimensional spatial relationship | threeD-scene-reconstruction, threeD-object-detection, egocentric-video-question-answering, threeD-object-tracking, threeD-pose-estimation, multiview-reasoning, multiview-action-recognition, threeD-depth-estimation, threeD-question-answering, threed-cad-recognition, threed-indoor-recognition | 11 |
| Discrete time (event) temporal relationship | visual-coherence, textual-cloze, gui-app-recognition, gui-next-action-prediction, visual-cloze, visual-ordering | 6 |
| Continuous time temporal relationship | general-action-recognition, video-captioning, next-img-prediction, temporal-ordering, meme-video-understanding, action-quality-assessment, temporal-localization, mevis | 8 |
| Low-level semantic relations | visual-quality-assessment, forensic-detection | 2 |
| High-level semantic relationship (objective) | visually-grounded-reasoning, image2image-retrieval, sketch2image-retrieval, vehicle-retrieval, text2image-retrieval, face-retrieval, handwritten-retrieval, person-reid, spot-the-diff, spot-the-similarity, visual-correspondence, semantic-correspondence, functional-correspondence | 13 |
| High-level semantic relationship (subjective) | emotion-recognition, casuality-reasoning, multiple-image-captioning | 3 |

**Tasks and corresponding datasets.** To introduce MMIU more thoroughly, we need to introduce all 52 tasks in MMIU, their specific descriptions, and which datasets they come from. Table 5, Table 5, and Table 7 respectively show the specific descriptions and data sources of tasks corresponding to temporal relationships, spatial relationships, and semantic relationships.

Table 5: Task descriptions and corresponding datasets for multi-image tasks in temporal relationships

| Task Name | Task Description | Dataset |
|---|---|---|
| Action Quality Assessment | Action Quality Assessment involves evaluating the quality of an action or movement depicted in a sequence of natural images. Given a sequence of natural images capturing the action, the task requires assessing the quality of the action or movement. | Olympic (Parmar & Tran Morris, 2017), AQA-7 (Parmar & Morris, 2019) |
| Action Recognition | General Action Recognition is a vision task that involves recognizing and classifying the actions or activities depicted in a sequence of natural images. | Kinetics (Kay et al., 2017) |
| Meme Video Understanding | Meme Video Understanding task involves understanding and interpreting the content and context of a meme video, where the visual input consists of a sequence of synthetic images. The task requires providing an explanation of the meme video content or context. | FunQA (Xie et al., 2023) |
| Mevis | MeVIS involves localizing objects of interest within a series of natural images. | MeVIS (Ding et al., 2023) |
| Next Image Prediction | Next Image Prediction refers to predicting the image at the next moment based on a given series of images in chronological order. | Moving MNist (Srivastava et al., 2015) |
| Temporal Localization | Temporal Localization involves identifying the instance or target in a sequence of frames or a video at a specific time or time range. The task requires analyzing a sequence of natural images and determining the identifier of the target instance in the sequence. | YouCook2 (Zhou et al., 2018), THUMOS14 (Wang et al., 2014) |
| Temporal Ordering | Temporal Ordering is a vision task that involves arranging a sequence of shuffled natural images in the correct temporal order. | Penn-Action (Chiu et al., 2019) |
| Video Captioning | Video Captioning involves generating textual descriptions for a sequence of video frames, providing a narrative or informative explanation for the visual content. | MSVD (Chen & Dolan, 2011), MSRVTT (Xu et al., 2016) |
| visual close | Visual cloze style questions test a skill similar to that of textual cloze task with the difference that the missing information in this task reside in the visual domain | RecipeQA (Yagcioglu et al., 2018) |
| textual close | Textual cloze style questions test the ability to infer missing text either in the title or in the step description by taking into account the question's context which includes a set of illustrative images besides text | RecipeQA (Yagcioglu et al., 2018) |
| visual coherence | Visual coherence style questions test the capability to identify an incoherent image in an ordered set of images given the titles and descriptions of the corresponding recipe as the context. | RecipeQA (Yagcioglu et al., 2018) |

Table 5 – continued from previous page

| Task Name | Task Description | Dataset |
|---|---|---|
| visual ordering | Visual ordering questions test the ability of a system in finding a correctly ordered sequence given a jumbled set of representative images of a recipe. As in the previous visual tasks, the context of this task consists of the titles and descriptions of a recipe | RecipeQA (Yagcioglu et al., 2018) |
| gui app recognition | Identify and analyze the applications utilized in the graphical user interface (GUI) segment of the episode. | GUI-Odyssey (Lu et al., 2024b) |
| gui next action prediction | Predict the subsequent action based on the information provided in the previous screenshot and the given graphical user interface (GUI) navigation instructions. | GUI-Odyssey (Lu et al., 2024b) |

Table 6: Task descriptions and corresponding datasets for multi-image tasks in spatial relationships

| Task Name | Task Description | Dataset |
|---|---|---|
| Raven's Progressive Matrices | Raven's Progressive Matrices is a visual reasoning task involving synthetic images. Given a set of visual patterns, the task requires identifying the missing pattern from a set of options. | RAVEN (Zhang et al., 2019), PGM (Santoro et al., 2018) |
| Jigsaw Puzzle Solving | Jigsaw Puzzle Solving task involves solving a jigsaw puzzle made up of natural images. The visual input consists of a shuffled patch of a natural image, and the instruction asks to rearrange the patches to reconstruct the original image. The patches can be fed as a set of images. | MSCOCO (Lin et al., 2014), WikiArt (Saleh & Elgammal, 2015) |
| Image Spatial Transformation Estimation | Given pairs of images depicting scenes before and after a spatial transformation (e.g., rotation, translation), predict the type and magnitude of the transformation that occurred. | MSCOCO (Lin et al., 2014) |
| Image Captioning with Spatial Context | Given a set of images (in NLVR, each sample can be split into 3 images), generate one sentence consistent with all images in terms of spatial context. | NLVR (Suhr et al., 2017) |
| Icon Question Answering with Spatial Context | Answer a multi-choice question in an icon image context. | IconQA (Lu et al., 2021) (a subset of it addresses spatial reasoning with multi-image.) |
| Image Text Retrieval with Spatial Context | Given a text addressing spatial context, identify the matched image within candidates. | SPEC (Kocabas et al., 2021) |

Table 6 – continued from previous page

| Task Name | Task Description | Dataset |
|---|---|---|
| Homography Estimation | Computing the 3x3 homography matrix that maps the coordinates of points in one image to their corresponding coordinates in another image. (Two images of the same planar.) | HPatch (Balntas et al., 2017), Kaggle for HPatch |
| Single Object Tracking | Visual Tracking involves following an object or region of interest across a series of images or frames. Given a query natural image with visual annotations, the task is to track the specified object or region in subsequent natural images. | TAP-Vid-DAVIS, TAP-Vid-RGB-stacking (Doersch et al., 2022) |
| Point Tracking | Point Tracking involves locating and tracking a specific point of interest within a natural image. Given a query natural image with a visual mark indicating the initial position of the point, the task requires finding the same point within another natural image. | Mevis (Ding et al., 2023) |
| 3D Classification - CAD | 3D classification - CAD involves classifying 3D images into specific categories based on their content and features. | ModelNet40 (Wu et al., 2015) |
| 3D Classification - Indoor Point Cloud | 3D classification - indoor Point Cloud involves categorizing indoor scenes based on 3D point cloud data. | ScanObjectNN (Uy et al., 2019) |
| Multi-view Reasoning | This task is centered on evaluating the multi-view reasoning capabilities of models. The objective is to deduce the relative camera motion based on two images of an object captured from different viewpoints. | BLINK (Fu et al., 2024b) |
| 3D Object Detection and Pose Estimation | Detect objects and estimate their poses in 3D space using multiple views of the scene. Input Format: A Set of RGB images captured from different viewpoints, and a query image. Output Format: Detected objects with their 3D bounding boxes and poses based on the query image. | ScanNet (Dai et al., 2017), SceneNet (Handa et al., 2016), SUN RGB-D (Song et al., 2015), nuScenes (Caesar et al., 2020) |
| 3D Scene Reconstruction | Reconstruct the 3D geometry of a scene. Input Format: An RGB image and a depth image. Output Format: A set of images captured from different viewpoints for this scene. | ScanNet (Dai et al., 2017), Matterport3D (Chang et al., 2017), SUN RGB-D (Song et al., 2015) |
| 3D Object Tracking | Input: Sequences of RGB-D images capturing object motion over time. Task: Track the movement of objects in 3D space across multiple frames. Output: Trajectories or paths of objects in 3D space (e.g., a sequence of 3D poses (position and orientation)). | KITTI (Geiger et al., 2013), nuScenes (Caesar et al., 2020) |

Table 6 – continued from previous page

| Task Name | Task Description | Dataset |
|---|---|---|
| Multi-View Object Instance Segmentation | Estimate the instance-level segmentation map for a query image based on multiple images captured from different viewpoints. Input Format: A Set of RGB images captured from different viewpoints, and a query image. Output Format: A corresponding instance-level segmentation map for the query image. | ScanNet (Dai et al., 2017), SceneNet (Handa et al., 2016), NYU Depth Dataset (Silberman et al., 2012), SUN RGB-D (Song et al., 2015) |
| Multi-View Depth Estimation | Estimate the depth map for a query image based on multiple images captured from different viewpoints. Input Format: A Set of RGB images captured from different viewpoints, and a query image. Output Format: A corresponding depth map for the query image. | MegaDepth (Li & Snavely, 2018), SceneNet (Handa et al., 2016), SUN RGB-D (Song et al., 2015) |
| Multi-View Action Recognition | Recognize human actions or activities in a scene using information from multiple views. Input Format: A set of RGB images from multiple views. Output Format: Action labels/categories. | NTU RGB+D (Shahroudy et al., 2016), PKUMMD (Liu et al.) |
| 3D Question Answering | Given inputs of the point cloud and a question about the 3D scene (real life), the model aims to output the correct answer. | ScanQA (Azuma et al., 2022), NuScenes-QA (Qian et al., 2024), SQA3D (Ma et al., 2022) |
| Egocentric Video Question-Answering | Egocentric Video Question-Answering (EgoVQA) is a task that involves understanding and reasoning about activities and events from the first-person perspective. In this task, the model is presented with a sequence of egocentric (first-person) videos, typically captured by wearable cameras such as head-mounted cameras. The goal is to answer questions related to the content and context of the videos. | EgoTaskQA (Jia et al., 2022) |

Table 6 – continued from previous page

| Task Name | Task Description | Dataset |
|---|---|---|
| Visual Navigation and Robotics | Given a series of images captured by robots or drones in different locations, the model outputs navigation commands or robot actions based on its spatial reasoning about the environment. Outputs may include directions for navigation, obstacle avoidance strategies, or object manipulation instructions. | DriveMLM (synthetic), YouTube-VIS (Yang et al., 2019), DAVIS (Pont-Tuset et al., 2017), VOT2018 (Kristan et al., 2018) |

Table 7: Task descriptions and corresponding datasets for multi-image tasks in semantic relationships

| Task Name | Task Description | Dataset |
|---|---|---|
| Visual Quality Assessment | This task is to evaluate the visual quality of two images, such as resolution, brightness, and clarity. | Q-bench (Wu et al., 2023), VE-LOL-L (Liu et al., 2021b) |
| Forensic Detection | This task involves multiple images and requires determining which image is fake and not authentically composed. | FaceForensics++ (Rossler et al., 2019), ForgeryNet (He et al., 2021) |
| Visually Grounded Reasoning | This task involves giving a pair of images and checking if the sentence description matches the image pair. | NLVR v2 (Suhr et al., 2017), MaRVL (Liu et al., 2021a) |
| Image-to-Image Retrieval | Image-to-Image Retrieval involves retrieving the candidate image ID that is most similar to the query image. | places365 (Zhou et al., 2017), tiny-imagenet (Le & Yang, 2015) |
| Sketch-to-Image Retrieval | Sketch-to-Image Retrieval involves retrieving candidate images that are most similar to a given sketch image. | quickdraw (Ha & Eck, 2017), DomainNet (Peng et al., 2019) |

Table 7 – continued from previous page

| Task Name | Task Description | Dataset |
|---|---|---|
| Text-to-Image Retrieval | Text-to-Image task involves generating an image based on a given textual description. The visual input consists of natural images, and the task instruction example could be 'Generate an image based on the provided text description.' The output provides the identifier of the generated image. | CUB220-2011 (Wah et al., 2011), Flowers102 (Nilsback & Zisserman, 2008) |
| Person Re-Identification | Person Re-Identification involves identifying and matching a person's appearance across different camera views or over time. The task requires comparing a query image of a person with multiple candidate images to determine if the same person appears in the candidates. | Market-1501-v15 (Zheng et al., 2015) |
| Vehicle Re-Identification | Vehicle Re-Identification involves identifying a specific vehicle from a set of candidate vehicle images based on a given query image of the vehicle. | veri-776 (Liu et al., 2016) |
| Face Verification | Face verification involves recognizing the identity of a query face image by comparing it with each support face image with an annotated identity. | LFW (Huang et al., 2008), CelebA (Liu et al., 2015) |
| Handwritten Text Retrieval | Handwritten Text Retrieval and Verification involves retrieving and verifying handwritten text from a query image against candidate images containing handwritten text. | IAM (Marti & Bunke, 2002) |
| Spot the Difference | Spot the Difference task involves identifying the numeric value corresponding to the number of differences between two natural images. | spot-the-diff (Jhamtani & Berg-Kirkpatrick, 2018) |
| Spot the Similarity | Spot the Similarity involves identifying the similarity between multiple images and providing an explanation for the judgment. | TLL (Rosenfeld et al., 2018), DISC21 (Douze et al., 2021) |
| Visual Correspondence | This task involves providing several images from different angles and finding the same points in different perspectives, such as specific pixels. | BLINK (Fu et al., 2024b), ScanNet (Dai et al., 2017) |

Table 7 – continued from previous page

| Task Name | Task Description | Dataset |
|---|---|---|
| Semantic Correspondence | The task requires providing several images of different species and identifying semantically identical points across the different species, such as the head of a horse and the head of a human. | BLINK (Fu et al., 2024b), MISC210K |
| Functional Correspondence | The task requires providing several images of different tools and identifying functionally identical points across the different tools, such as the handle of a broom and the handle of a toothbrush. | BLINK (Fu et al., 2024b), FunKPoint |
| Emotion Recognition | The task is to provide multiple images, most of which depict the same emotion, and identify the one image that represents a different emotion. | FindingEmo (Mertens et al., 2024), ExpW (Zhang et al., 2018b) |
| Casuality Reasoning | The task is to provide multiple images from a video and ask about the cause leading to a specific result, such as: "Why did the little girl stop the car?" The answer might be: "She stopped to wait for her mom." | NeXTQA (Xiao et al., 2021), VideoABC (Zhao et al., 2022) |
| Multi-image Captioning | The task is to provide multiple images of discrete events and require a title for them. | SSID (Abdelhamed et al., 2018) |

## A.3 Task Abbreviations

Due to the large number of tasks and models evaluated in the benchmark, we use abbreviations to streamline the manuscript. Table 8 lists the abbreviations used throughout the paper.

## A.4 Construction

**Metadata.** We organize the dataset of each collected task into a metadata format. Specifically, the metadata is organized as a dictionary, with keys categorized into two main groups. The first group comprises essential keys for task information, including task name, task description, input format and output format. The second group consists of keys that are unique to individual samples and may vary accordingly, including data sources, input image path, input context, question, visual components and output (*i.e.*, the ground truth). This structured format helps us easily convert it into multiple-choice questions without losing information. The specific metadata format can be referenced in Table 9.

Based on the above predefined template, we obtain the detailed information of metadata primarily following two steps. In the first step, we create a Python script for each dataset to extract relevant keys directly from the original data, such as the image path, data source, and the ground truth. Since samples within a dataset tend to share similar features, our co-authors predefined elements such as the specific question template, input context template, and detailed visual components for each dataset. In the second step, our co-authors manually conduct a sample review focusing on sample-specific keys, including the predefined question templates and the output information.

**Ensuring Challenging Distractor Options in Samples**. To create difficult distractor options, we primarily rely on prompt engineering with GPT-4o, utilizing tailored prompts for specific tasks and applying in-context learning (with 2-shot or 3-shot examples). The generated options are thoroughly

Table 8: The Abbreviations of terms mentioned in this paper and their corresponding full terms.

| Abbreviation | Full Term | | Abbreviation | Full Term |
|---|---|---|---|---|
| | | Tasks | | |
| I2IR | Image2Image Retrieval | | S2IR | Sketch2Image Retrieval |
| VR | Vehicle Retrieval | | FD | Forensic Detection |
| CR | Causality Reasoning | | T2IR | Text2Image Retrieval |
| FR | Face Retrieval | | ER | Emotion Recognition |
| FC | Functional Correspondence | | HR | Handwritten Retrieval |
| VCor | Visual Correspondence | | VGR | Visually Grounded Reasoning |
| STD | Spot the Difference | | VQA | Visual Quality Assessment |
| MIC | Multiple Image Captioning | | PR | Person Re-ID |
| SC | Semantic Correspondence | | STS | Spot the Similarity |
| GAR | General Action Recognition | | AQA | Action Quality Assessment |
| NIP | Next Image Prediction | | MVU | Meme Video Understanding |
| TL | Temporal Localization | | MEV | MeVis |
| TC | Textual Cloze | | GuAR | GUI App Recognition |
| GNAP | GUI Next Action Prediction | | VO | Visual Ordering |
| VCo | Visual Coherence | | VidCap | Video Captioning |
| TO | Temporal Ordering | | VClz | Visual Cloze |
| MAR | Multiview Action Recognition | | HE | Homography Estimation |
| 3DOT | 3D Object Tracking | | ICSC | Image Captioning with Spatial Context |
| MR | Multiview Reasoning | | ITRSC | Image Text Retrieval with Spatial Context |
| IQASC | Icon Question Answering with Spatial Context | | 3DE | 3D Depth Estimation |
| RPM | Ravens Progressive Matrices | | 3DPE | 3D Pose Estimation |
| 3DSR | 3D Scene Reconstruction | | JPS | Jigsaw Puzzle Solving |
| 3DCR | 3D CAD Recognition | | 3DOD | 3D Object Detection |
| ISTE | Image Spatial Transformation Estimation | | EVQA | Egocentric Video Question Answering |
| 3DIR | 3D Indoor Recognition | | 3DQA | 3D Question Answering |
| PT | Point Tracking | | SOT | Single Object Tracking |

Table 9: The example of the metadata.

**Metadata Example**

Task Info: {
    TaskName: *Name of the task* ,
    TaskDescription: *Description of the task* ,
    Input Format: *Input data formats, such as text and images* ,
    Output Format: *Output data format, such as text or image* ,
}
Samples: [
    {
        Source: *The data set source of this sample* ,
        Input: {
            Input Image: *The path of the input image, in list format* ,
            Input Context: *The context needed to solve the problem, in text form* ,
            Question: *Input question or instruction* ,
            Visual Component: *Image type, such as depth image, natural image* ,
            }
        Output: *The actual textual output of the problem, which may be text (caption task) or image path (retrieval task), etc.* ,
    }
]

reviewed, and any inaccuracies are corrected. Notably, GPT-4o is capable of producing highly challenging options in two key ways. Firstly, the distractors are often semantically similar to the ground truth (GT), making them particularly confusing in classification tasks such as 3D Indoor Recognition. For example, if the correct answer is "sink," the distractors might include "cabinet," "bed or "bin," all of which share similar semantic background with the GT. Secondly, the distractors can exhibit ambiguity in their relation to visual content. For captioning tasks, we leverage GPT-4o to analyze images and assist in generating plausible incorrect options, thereby mitigating the hallucination issues common with text-only LLMs. In tasks involving object detection, we introduce perturbations to the GT using Python scripts to create distinct yet challenging options. Overall, we

Table 10: LPIPS between MMIU and MuirBench, MilesBench

| LPIPS ↓ | 0.1 | 0.2 | 0.3 | 0.4 |
|---|---|---|---|---|
| MuirBench | 0.1 | 1.8 | 6.4 | 11.8 |
| MilesBench | 1.6 | 3.3 | 7.8 | 12.9 |

believe that MMIU generates options with an appropriate level of difficulty, enhancing the robustness of our evaluation tasks.

**Visual Grounding of MMIU Answers**. To ensure that the generated ground truth answers in MMIU are visually grounded, we adopt the following strategies. First, during the question design process for each sample, we meticulously craft specific question templates. We avoid embedding overly specific information within the templates, ensuring that the questions heavily require reliance on visual content for accurate answers. Second, when generating answer options, we create more complex distractors to prevent models from bypassing visual information and providing responses solely based on correlations between options or prior knowledge.

**Unanswerable Set.** We consider five strategies for modifying an answerable instance into its unanswerable counterpart with minimal changes. The five strategies include replacing key words, replacing the answer image, replacing other images, shuffling all images, and using irrelevant question/image sets. For each task, we select the most suitable strategy or combination of strategies to construct the corresponding unanswerable task. The specific construction methods for each task can be referenced in Table 19.

**The difference with other benchmarks.** MMIU is a comprehensive benchmark designed to evaluate the multi-image capabilities of multimodal large models across a wide range of tasks involving semantic, temporal, and spatial relationships among multiple images. Compared to other multi-image benchmarks such as MuirBench, MMIU offers broader task coverage and more in-depth analysis of results. Moreover, MMIU is distinct from these benchmarks in its data, representing a completely independent benchmark. We assessed the similarity between the images in MMIU and those in MuirBench and MilesBench using LPIPS (Zhang et al., 2018a) for visual similarity. Many image editing works adopt LPIPS as a metric, and based on their reported results, an LPIPS value below $0.1$ can be considered as indicating that the images are relatively similar (Zhao et al., 2024; Xu et al., 2024). As shown in Table 10, when a threshold of $0.1$ is applied, the image duplication rates are as low as 0.1% and 1.6%, respectively.

Table 11: Causes of Unanswerable Tasks Considered in Their Construction: Note that a single task often corresponds to multiple causes for being unanswerable.

| Task | replace key word | replace answer image | replace other images | shuffle all images | irrelevant question/image set |
|---|---|---|---|---|---|
| CR | Yes | No | No | No | Yes |
| T2IR | Yes | Yes | No | No | Yes |
| VCo | No | Yes | No | No | No |
| VO | No | No | No | No | No |
| GAR | No | No | No | Yes | Yes |
| TL | Yes | Yes | No | No | Yes |
| TO | No | No | No | Yes | No |
| VidCap | No | No | No | Yes | Yes |
| HE | No | No | Yes | Yes | Yes |
| IQASC | Yes | Yes | Yes | No | Yes |
| ISTE | No | No | Yes | No | Yes |
| ITRSC | Yes | Yes | Yes | No | Yes |
| JPS | No | No | Yes | Yes | Yes |
| MAR | No | No | Yes | No | Yes |
| 3DE | No | Yes | Yes | Yes | Yes |
| 3DOD | No | No | Yes | Yes | No |
| 3DPE | No | No | Yes | Yes | Yes |
| 3DSR | No | Yes | Yes | Yes | Yes |
| 3DCR | No | Yes | No | No | Yes |
| 3DIR | No | Yes | No | No | Yes |

# B  EXPERIMENT DETAILS

## B.1  MODEL DETAILS

Table 12 provides an overview of the LVLMs utilized in this study, detailing their parameter sizes, visual encoders, and LLMs. It is important to mention that the evaluation process was carried out according to the protocol established by OpenCompass. (Contributors, 2023)

Table 12: Model architecture of 24 LVLMs evaluated on MMIU.

| Models | Parameters | Vision Encoder | LLM |
|---|---|---|---|
| GPT-4o (OpenAI, 2024) | - | - | - |
| Gemini1.5 flash (Team et al., 2023) | - | - | - |
| Gemini1.0 ProVision (Team et al., 2023) | - | - | - |
| Claude3.5-Sonnet (Anthropic, 2023) | - | - | - |
| Mantis-idefics2 (Jiang et al., 2024b) | 8B | CLIP ViT-L/14 | Mistral-7B-v0.1 |
| Mantis-SigCLIP (Jiang et al., 2024b) | 8B | CLIP ViT-L/14 | LLaMA-3-8B |
| xGen-MM (Xue et al., 2024) | 4B | xGen-MM-vision-encoder | Phi-3-mini |
| LLaVA-Next-Vicuna-7B (Liu et al., 2024a) | 7.1B | CLIP ViT-L/14 | Vicuna-v1.5-7B |
| LLaVA-Next-Interleave (Liu et al., 2024a) | 7.1B | CLIP ViT-L/14 | Vicuna-v1.5-7B |
| InternVL2-Pro (Chen et al., 2024a) | - | - | - |
| InternVL2 (Chen et al., 2024a) | 8B | InternViT-300M-448px | Internlm2-5-7b-chat |
| InternVL-Chat-V1.5 (Chen et al., 2024a) | 26B | InternViT-6B | InternLM2-Chat-20B |
| Mini-InternVL-Chat (Chen et al., 2024a) | 2B | InternViT-300M | InternLM2-Chat-1.8B |
| Mini-InternVL-Chat (Chen et al., 2024a) | 4B | InternViT-300M | Phi-3-mini |
| DeepSeek-VL-1.3B (Lu et al., 2024a) | 1.3B | SAM-B & SigLIP-L | DeekSeek-1.3B |
| DeepSeek-VL-7B (Lu et al., 2024a) | 7.3B | SAM-B & SigLIP-L | DeekSeek-7B |
| Monkey-chat (Li et al., 2024c) | 9.8B | CLIP-ViT-BigHuge | Qwen-7B |
| XComposer2 (Dong et al., 2024) | 7B | CLIP ViT-L/14 | InternLM2-7B |
| XComposer2-1.8b (Dong et al., 2024) | 1.8B | CLIP ViT-L/14 | InternLM2-1.8B |
| ShareGPT4V (Chen et al., 2023) | 7.2B | CLIP ViT-L/14 | Vicuna-v1.5-7B |
| SharedCaptioner (Chen et al., 2023) | 8B | EVA-G | InternLM-7B |
| LLaVA-v1.5-7B (Liu et al., 2024b) | 7.2B | CLIP ViT-L/14 | Vicuna-v1.5-7B |
| Qwen-Base (Bai et al., 2023) | 9.6B | CLIP ViT-G/16 | Qwen-7B |
| Qwen-chat (Bai et al., 2023) | 9.6B | CLIP ViT-G/16 | Qwen-7B |
| Idefics2-8b (Laurençon et al., 2024) | 8B | SigLIP-L | Mistral-7B |
| Idefics-9b-instruct (Bai et al., 2023) | 9B | CLIP-ViT-H-14 | Llama-7b |
| Monkey-chat (Li et al., 2024c) | 9.8B | Vit-BigG | QwenVL-7B |
| MiniCPM-Llama3-V-2.5 (Hu et al., 2024) | 8.4B | SigLip-L | Llama3-8B |
| FlamingoV2 (Awadalla et al., 2023) | 9.6B | CLIP ViT-G/16 | Qwen-7B |
| GLM-4V-9B (GLM et al., 2024) | 13B | - | GLM-4-9B |
| Llama3.1-Instruct (Dubey et al., 2024) | 8B | - | - |
| InternLM2.5-Instruct (Cai et al., 2024) | 7B | - | - |
| Qwen2.5-Instruct (Team, 2024) | 7B | - | - |

Table 13: The complete prompt template for MMIU

**Model Prompts**

Context: {CONTEXT}
Question: {QUESTION}
Choices:
(A) {OPTION_A}
(B) {OPTION_B}
(C) {OPTION_C}
(D) {OPTION_D}
Hint: Please answer the option directly like A, B, C, D...

## B.2  MODEL PROMPTS

According to MathVista (Lu et al., 2023), our prompt consists of four parts: the question, options, the hint indicating the answer format, and the context of this task (e.g. *Your task is to track the movement of objects in 3D space across multiple frames, select from the following choices.*). For

Table 14: Quantitative results across 52 tasks on test set are summarized. Accuracy is the metric, and the Overall score is computed across all tasks. The maximum value of each task is bolded. Notice that although InternVL1.5-chat supports multiple image inputs, its training phase did not incorporate multi-image data. The full term of task abbreviation can be found in Table 8 in Appendix.

| Model | Overall | CR | ER | FD | FC | SC | VCor | VQA | VGR | FR | HR | I2IR | MIC | PR | S2IR | STD | STS | T2IR | VR | AQA | GAR | MVU | MEV | NIP | TL | TO | VidCap |
| --- | --- | --- | --- | --- | --- | --- | --- | --- | --- | --- | --- | --- | --- | --- | --- | --- | --- | --- | --- | --- | --- | --- | --- | --- | --- | --- | --- |
| | | GuAR | GNAP | TC | VCIz | VCo | VO | EVQA | HE | IQASC | ICSC | ISTE | ITRSC | MAR | MR | JPS | 3DE | 3DOD | 3DOT | 3DPE | 3DSR | 3DQA | RPM | PT | SOT | 3DCR | 3DIR |
| Frequency | 31.5 | 32.0 | 27.7 | 27.3 | 30.0 | 30.2 | 29.6 | 49.0 | 76.5 | 29.0 | 28.0 | 27.5 | 29.0 | 30.0 | 37.0 | 51.5 | 50.0 | 26.5 | 31.0 | 32.0 | 30.0 | 29.0 | 30.0 | 28.5 | 30.1 | 27.5 | |
| | | 31.5 | 28.0 | 28.5 | 27.5 | 30.5 | 31.0 | 27.5 | 27.5 | 41.5 | 27.5 | 30.0 | 18.0 | 27.6 | 55.6 | 29.0 | 26.5 | 29.0 | 28.0 | 26.5 | 28.5 | 29.5 | 30.5 | 18.0 | 28.0 | 26.0 | 27.0 |
| Random | 27.4 | 19.0 | 23.0 | 22.3 | 26.4 | 24.7 | 29.1 | 45.0 | 50.0 | 23.0 | 26.0 | 24.0 | 20.0 | 24.5 | 37.5 | 51.0 | 55.0 | 27.5 | 28.0 | 26.5 | 24.0 | 27.5 | 23.0 | 26.9 | 24.5 | 23.0 | |
| | | 21.0 | 12.5 | 24.0 | 27.5 | 20.5 | 27.0 | 32.0 | 31.5 | 38.5 | 27.0 | 26.0 | 14.0 | 24.6 | 50.4 | 23.5 | 25.5 | 24.5 | 22.5 | 31.0 | 23.5 | 24.5 | 25.5 | 10.5 | 22.5 | 27.0 | 27.0 |
| *Closed-source LVLMs* | | | | | | | | | | | | | | | | | | | | | | | | | | | |
| GPT-4o | **55.7** | 67.8 | **46.5** | **88.8** | 42.6 | 41.5 | **72.6** | 79.2 | 61.3 | 76.0 | 42.0 | 59.5 | **93.5** | 61.5 | 67.0 | 11.0 | **84.0** | **70.5** | 68.0 | 33.5 | **91.5** | 71.5 | **35.0** | 26.5 | **50.8** | **28.0** | **92.5** |
| | | 78.0 | 46.5 | 62.5 | 43.5 | **97.5** | 21.5 | 57.5 | **29.5** | 88.0 | 58.5 | **35.0** | 17.5 | **81.9** | 46.6 | 23.5 | 24.0 | 40.5 | **94.5** | 85.0 | 22.0 | 39.0 | 55.0 | 12.5 | 56.0 | **69.0** | 49.0 |
| Gemini1.5 | 53.4 | 71.0 | 31.8 | 73.5 | 24.3 | 34.9 | 47.3 | 78.8 | 61.0 | 88.0 | **80.0** | 74.0 | 89.0 | 70.5 | **81.5** | 74.0 | 80.0 | 60.5 | 68.0 | **35.5** | 88.0 | **75.0** | 25.0 | 21.0 | 45.6 | 26.5 | 84.0 |
| | | **93.0** | 39.5 | 59.0 | 30.0 | 60.0 | **43.5** | 53.5 | 22.5 | **91.0** | **64.5** | 24.0 | 13.0 | 68.8 | 51.1 | **34.5** | 20.0 | 32.0 | 48.5 | 37.5 | 28.5 | 35.5 | 66.5 | 13.0 | 61.0 | 55.0 | 43.0 |
| Claude3.5 | 53.4 | 70.2 | 38.5 | 76.6 | 31.3 | 34.9 | 57.0 | 77.8 | 54.5 | 92.0 | 79.0 | 62.0 | 85.5 | 77.5 | 68.0 | 80.0 | 57.5 | 65.5 | 79.0 | 26.0 | 80.5 | **75.0** | 33.5 | 10.5 | 43.5 | 23.0 | 91.0 |
| | | 88.5 | **55.0** | 56.0 | 26.5 | 67.5 | 38.5 | 53.5 | 23.0 | 78.5 | 52.0 | 32.0 | 4.0 | 64.8 | 42.1 | 31.5 | 23.5 | **41.0** | 32.0 | **99.5** | 21.5 | 28.5 | 78.5 | 10.5 | **67.5** | 53.5 | 36.5 |
| Gemini1.0 | 40.2 | 63.2 | 26.5 | 36.6 | 27.5 | 28.3 | 30.3 | 60.8 | **71.0** | 25.0 | 24.5 | 28.0 | 84.0 | 21.0 | 44.0 | 71.0 | 48.0 | 27.0 | 31.5 | 34.5 | 89.0 | 73.5 | 29.0 | 21.5 | 37.3 | 23.5 | 90.0 |
| | | 87.0 | 35.5 | 62.5 | 24.5 | 42.0 | 23.0 | 45.5 | 17.0 | 53.0 | 55.0 | 22.5 | 16.0 | 71.9 | 43.6 | 28.0 | 22.0 | 28.0 | 36.0 | 7.0 | 24.5 | 39.0 | 17.0 | 12.0 | 47.0 | 53.0 | 33.5 |
| *Adequate Multi-Image SFT LVLMs* | | | | | | | | | | | | | | | | | | | | | | | | | | | |
| Mantis | 45.6 | 61.5 | 31.8 | 57.0 | 24.3 | 28.1 | 30.9 | 59.8 | 65.2 | 66.5 | 54.0 | 63.5 | 71.0 | 57.5 | 64.5 | 96.0 | 65.5 | 46.5 | 70.5 | 17.5 | 81.0 | 58.5 | 28.5 | 26.0 | 23.8 | 27.0 | 85.0 |
| | | 73.5 | 34.0 | 51.5 | 31.0 | 14.0 | 20.0 | 54.5 | 23.0 | 66.0 | 48.0 | 23.5 | 13.0 | 71.4 | 47.4 | 27.5 | 23.5 | 24.0 | 26.0 | **50.5** | 76.0 | 13.5 | 59.5 | 5.0 | 50.0 | 59.0 | 40.5 |
| Mantis-SigCLIP | 41.8 | 40.0 | 32.5 | 57.5 | 22.5 | 25.0 | 30.0 | 77.5 | 60.0 | 50.0 | 45.0 | 70.0 | 90.0 | 35.0 | 60.0 | **100.0** | 60.0 | 60.0 | 65.0 | 30.0 | 80.0 | 45.0 | 30.0 | 30.0 | 15.0 | 25.0 | 65.0 |
| | | 60.0 | 35.0 | 50.0 | 30.0 | 30.0 | 10.0 | 60.0 | 20.0 | 40.0 | 40.0 | 40.0 | 20.0 | 35.0 | 15.0 | 0.0 | 25.0 | 75.0 | 45.0 | 35.0 | 5.0 | 50.0 | 55.0 | 20.0 | | | |
| LLaVA-Next-Interleave | 32.4 | 29.5 | 24.8 | 26.3 | 23.2 | 26.4 | 25.1 | 48.8 | 49.8 | 23.5 | 25.0 | 28.0 | 57.0 | 21.5 | 33.0 | 63.5 | 54.5 | 25.0 | 26.0 | 24.0 | 27.0 | 49.5 | 29.0 | 23.0 | 25.4 | 27.5 | 32.5 |
| | | 43.0 | 34.0 | 49.0 | 29.5 | 32.0 | 26.0 | 30.0 | 21.5 | 42.0 | 47.5 | 22.5 | 14.0 | 23.6 | 32.3 | 17.5 | **28.5** | 23.0 | 17.5 | 3.0 | 31.0 | 36.0 | **79.0** | 15.0 | 60.5 | 34.5 | 42.5 |
| xGen-MM-interleaved-4B | 28.2 | 20.0 | 22.5 | 42.5 | 37.5 | 32.5 | 0.0 | 42.5 | 52.5 | 10.0 | 10.0 | 30.0 | 85.0 | 20.0 | 25.0 | 35.0 | 30.0 | 10.0 | 0.0 | 15.0 | 15.0 | 65.0 | 35.0 | 5.0 | 15.0 | 15.0 | 25.0 |
| | | 45.0 | 25.0 | 25.0 | 20.0 | 30.0 | 55.0 | 20.0 | 15.0 | 55.0 | 30.0 | 10.0 | 10.0 | 0.0 | 60.0 | 25.0 | 20.0 | 20.0 | 5.0 | 0.0 | 30.0 | 75.0 | 5.0 | 55.0 | 30.0 | | |
| *Multi-Image input LVLMs* | | | | | | | | | | | | | | | | | | | | | | | | | | | |
| InternVL2 | 50.3 | **77.8** | 41.5 | 62.8 | 24.6 | 25.3 | 35.3 | **82.5** | 59.8 | **93.5** | 47.0 | **85.5** | 92.5 | **82.0** | 73.0 | 19.0 | 77.0 | 54.5 | **83.5** | 22.0 | 86.5 | 68.5 | 33.0 | 20.5 | 26.9 | 25.0 | 88.0 |
| | | 91.5 | 40.5 | 52.0 | 25.5 | 78.0 | 35.0 | **63.0** | 28.5 | 77.5 | 41.5 | 26.0 | 20.0 | 78.4 | **55.6** | 27.5 | 25.5 | 28.0 | 20.0 | 26.5 | **43.0** | 48.5 | 13.5 | **59.5** | 51.5 | 31.0 | |
| internvl1.5-chat | 37.4 | 63.7 | 31.0 | 22.6 | 20.3 | 16.3 | 28.3 | 63.2 | 38.5 | 21.0 | 28.0 | 26.5 | 82.5 | 20.5 | 31.5 | 6.0 | 45.5 | 26.5 | 29.5 | 29.5 | 85.0 | 65.0 | 32.0 | 23.5 | 29.0 | 18.5 | 89.0 |
| | | 90.5 | 35.5 | 56.5 | 23.5 | 31.0 | 24.5 | 53.0 | 26.0 | 40.0 | 49.0 | 25.5 | 15.5 | 59.3 | 43.6 | 19.5 | 22.5 | 23.5 | 15.0 | 28.0 | 39.0 | 71.0 | 9.5 | 46.5 | 50.5 | 39.5 | |
| InternVL2-8B | 34.0 | 37.5 | 22.5 | 22.5 | 22.5 | 27.5 | 15.0 | 45.0 | 42.5 | 35.0 | 15.0 | 10.0 | 80.0 | 25.0 | 35.0 | 30.0 | 65.0 | 30.0 | 45.0 | 30.0 | 75.0 | 55.0 | 10.0 | 5.0 | 30.0 | 40.0 | 85.0 |
| | | 60.0 | 15.0 | 35.0 | 40.0 | 20.0 | 40.0 | 50.0 | 5.0 | 35.0 | 50.0 | 30.0 | 10.0 | 50.0 | 60.0 | 20.0 | 20.0 | 25.0 | 25.0 | 15.0 | 30.0 | 40.0 | 65.0 | 0.0 | 40.0 | 35.0 | |
| Mini-InternVL-chat-1.5-4B | 32.4 | 35.0 | 27.5 | 22.5 | 25.0 | 20.0 | 17.5 | 45.0 | 60.0 | 10.0 | 0.0 | 30.0 | 60.0 | 35.0 | 35.0 | 25.0 | 65.0 | 20.0 | 25.0 | 30.0 | 35.0 | 75.0 | 35.0 | 30.0 | 25.0 | 15.0 | 90.0 |
| | | 25.0 | 20.0 | 50.0 | 35.0 | 40.0 | 30.0 | 35.0 | 10.0 | 55.0 | 35.0 | 20.0 | 5.0 | 60.0 | 25.0 | 20.0 | 10.0 | 15.0 | 15.0 | 15.0 | 35.0 | 65.0 | 10.0 | 35.0 | 45.0 | 25.0 | |
| Mini-InternVL-chat-1.5-2B | 31.8 | 32.5 | 15.0 | 20.0 | 25.0 | 12.5 | 25.0 | 35.0 | 47.5 | 10.0 | 20.0 | 20.0 | 80.0 | 20.0 | 35.0 | 30.0 | 65.0 | 30.0 | 30.0 | 25.0 | 85.0 | 65.0 | 30.0 | 5.0 | 25.0 | 25.0 | **95.0** |
| | | 20.0 | 30.0 | 45.0 | 15.0 | 25.0 | 25.0 | 40.0 | 25.0 | 35.0 | 20.0 | 10.0 | 15.0 | 45.0 | 65.0 | 10.0 | 35.0 | 20.0 | 10.0 | 30.0 | 30.0 | 15.0 | 30.0 | 35.0 | 45.0 | | |
| idefics2-8b | 27.8 | 28.0 | 25.8 | 26.4 | 26.7 | 24.6 | 28.6 | 58.5 | 30.8 | 3.5 | 9.5 | 4.0 | 82.0 | 5.0 | 27.5 | **98.5** | 70.5 | 12.5 | 7.0 | 16.0 | 24.5 | 12.0 | 19.0 | 23.5 | 22.3 | 18.0 | 19.5 |
| | | 23.5 | 22.5 | 21.0 | 26.5 | 21.5 | 22.5 | 14.5 | 21.5 | 21.5 | 50.5 | 25.5 | 13.5 | 15.1 | 51.6 | 21.5 | 9.0 | 21.5 | 11.0 | 55.0 | 10.5 | 18.0 | 52.5 | **44.5** | 40.5 | | |
| xGen-MM-single-4B | 25.4 | 12.5 | 0.0 | 0.0 | 0.0 | 0.0 | 5.0 | 37.5 | 45.0 | 35.0 | 5.0 | 25.0 | 90.0 | 25.0 | 25.0 | 35.0 | 60.0 | 25.0 | 30.0 | 10.0 | 20.0 | 60.0 | 5.0 | 0.0 | 30.0 | 15.0 | 40.0 |
| | | 15.0 | 25.0 | 35.0 | 30.0 | 25.0 | 25.0 | 50.0 | 20.0 | 20.0 | 40.0 | 0.0 | 5.0 | 0.0 | 60.0 | 30.0 | 20.0 | 25.0 | 15.0 | 0.0 | 35.0 | 80.0 | 5.0 | 50.0 | 25.0 | 15.0 | |
| DeepSeek-VL-7b | 24.6 | 2.2 | 22.2 | 29.1 | 23.3 | 28.2 | 29.0 | 49.0 | 65.5 | 20.5 | 25.5 | 72.5 | 21.0 | 30.5 | 65.0 | 54.5 | 25.5 | 31.0 | 0.0 | 0.0 | 0.0 | 27.5 | 31.1 | 15.5 | 2.0 | | |
| | | 10.0 | 14.0 | 5.5 | 17.0 | 30.5 | 21.5 | 0.0 | 23.0 | 45.5 | 42.0 | 24.5 | 0.0 | 2.0 | 44.4 | 20.5 | 24.5 | 24.5 | 0.0 | 7.5 | 0.5 | 1.5 | 78.0 | 0.5 | 62.5 | 40.5 | 38.5 |
| XComposer2-7b | 23.5 | 24.5 | 23.0 | 19.1 | 16.4 | 18.4 | 10.0 | 27.8 | 27.5 | 13.0 | 12.5 | 26.0 | 55.5 | 19.5 | 33.5 | 17.0 | 54.0 | 10.5 | 1.5 | 25.0 | 59.5 | 37.0 | 25.5 | 0.0 | 24.4 | 13.0 | 68.5 |
| | | 59.0 | 28.0 | 34.0 | 25.0 | 28.5 | 17.0 | 17.5 | 0.5 | 29.5 | 48.0 | 6.0 | 7.5 | 33.2 | 41.4 | 7.0 | 0.0 | 15.5 | 17.0 | 28.0 | 2.0 | 29.0 | 33.5 | 9.0 | 27.5 | 11.5 | 3.0 |
| DeepSeek-VL-1.3b | 23.2 | 1.2 | 27.5 | 21.4 | 23.1 | 26.7 | 30.0 | 45.2 | 54.8 | 20.5 | 25.5 | 46.0 | 21.0 | 30.5 | 89.0 | 0.0 | 23.0 | 31.0 | 0.0 | 1.0 | 2.5 | 0.0 | 24.3 | 20.0 | 1.0 | | |
| | | 6.5 | 13.0 | 3.5 | 11.5 | 33.0 | 20.0 | 0.5 | 25.0 | 44.5 | 38.0 | 24.0 | 1.0 | 0.0 | **55.6** | 31.0 | 26.0 | 31.0 | 0.0 | 19.5 | 0.0 | 1.5 | 66.5 | 3.0 | 61.5 | 45.5 | 29.0 |
| flamingov2 | 22.3 | 25.5 | 25.8 | 24.6 | 21.6 | 25.0 | 28.2 | 34.5 | 49.0 | 14.5 | 9.5 | 19.0 | 13.5 | 22.5 | 17.5 | 26.0 | 39.0 | 49.0 | 20.0 | 17.5 | 16.5 | 30.0 | 20.0 | 18.7 | 24.5 | 22.5 | |
| | | 25.0 | 21.5 | 25.5 | 25.0 | 14.5 | 13.5 | 15.5 | 27.5 | 4.0 | 25.5 | 23.0 | 7.0 | 22.1 | 3.0 | 1.5 | 26.5 | 22.0 | 35.0 | 17.0 | 28.5 | 20.5 | 23.5 | 11.5 | 31.0 | 25.0 | 23.5 |
| XComposer2-1.8b | 21.9 | 24.0 | 21.0 | 10.8 | 5.8 | 0.0 | 0.0 | 34.2 | 24.0 | 14.5 | 2.5 | 23.0 | 63.5 | 19.0 | 26.0 | 14.5 | 31.0 | 9.5 | 28.5 | 31.5 | 59.5 | 44.0 | 30.0 | 4.5 | 15.5 | 12.0 | 66.0 |
| | | 55.0 | 35.0 | 42.5 | 22.5 | 2.5 | 19.0 | 20.0 | 8.0 | 15.5 | 45.0 | 0.0 | 0.0 | 20.6 | 0.0 | 16.5 | 0.0 | 7.0 | 0.0 | 4.5 | 0.0 | 33.5 | 63.0 | 1.5 | 38.5 | 42.0 | 33.0 |
| Qwen-chat | 15.9 | 20.5 | 2.5 | 13.3 | 2.5 | 9.9 | 5.9 | 3.2 | 23.8 | 10.5 | 19.5 | 12.5 | 41.0 | 5.5 | 13.5 | 29.5 | 45.0 | 3.0 | 12.0 | 10.0 | 52.5 | 18.5 | 16.5 | 2.5 | 3.6 | 5.5 | 47.0 |
| | | 29.0 | 23.0 | 18.0 | 6.0 | 6.0 | 6.0 | 32.0 | 9.0 | 13.5 | 17.0 | 15.5 | 3.5 | 40.2 | 15.8 | 16.5 | 16.5 | 22.5 | 17.5 | 13.0 | 14.5 | 14.0 | 8.0 | 3.0 | 8.5 | 1.5 | 0.5 |
| idefics-9b-instruct | 12.8 | 10.8 | 0.2 | 0.2 | 0.8 | 0.0 | 9.4 | 23.0 | 13.0 | 2.5 | 22.0 | 14.0 | 70.0 | 3.0 | 14.5 | 40.5 | 34.5 | 3.5 | 2.0 | 4.0 | 1.5 | 20.0 | 3.0 | 15.5 | 0.5 | 0.0 | |
| | | 17.0 | 27.5 | 48.5 | 23.0 | 0.0 | 5.5 | 3.0 | 9.0 | 16.0 | 0.0 | 0.0 | 6.5 | 12.8 | 0.0 | 15.5 | 10.5 | 0.5 | 36.5 | 5.5 | 2.5 | 44.5 | 1.5 | 35.0 | 0.0 | 0.0 | |
| Qwen-Base | 5.2 | 9.2 | 0.5 | 5.7 | 5.8 | 0.5 | 1.0 | 5.0 | 4.5 | 0.0 | 1.0 | 0.0 | 20.5 | 0.0 | 2.5 | 1.0 | 43.0 | 1.0 | 0.0 | 0.0 | 4.5 | 8.5 | 0.5 | 0.0 | 0.0 | 7.5 | |
| | | 24.5 | 8.0 | 29.5 | 5.0 | 5.5 | 6.5 | 2.0 | 2.0 | 8.5 | 11.5 | 0.0 | 5.5 | 5.3 | 0.0 | 0.5 | 7.0 | 0.0 | 21.5 | 0.0 | 5.5 | 2.5 | 0.0 | 0.0 | 0.0 | 0.0 | |
| *Single-Image input LVLMs* | | | | | | | | | | | | | | | | | | | | | | | | | | | |
| GLM-4v-9b | 27.0 | 32.8 | 16.0 | 31.8 | 8.7 | 9.0 | 4.7 | 59.0 | 55.8 | 31.0 | 7.5 | 19.5 | 82.0 | 23.5 | 24.5 | 81.0 | 67.0 | 25.0 | 30.0 | 7.0 | 59.5 | 53.5 | 10.5 | 5.0 | 25.9 | 10.0 | 76.0 |
| | | 55.5 | 19.0 | 34.0 | 5.0 | 11.5 | 14.5 | 26.0 | 11.5 | 35.5 | 41.5 | 16.0 | 6.5 | 25.1 | 29.3 | 9.0 | 14.0 | 14.5 | 7.0 | 0.5 | 5.5 | 27.0 | 35.0 | 7.5 | 26.0 | 48.5 | 23.5 |
| LLaVA-next-vicuna-7b | 22.2 | 22.2 | 9.2 | 11.0 | 9.1 | 7.7 | 10.5 | 37.0 | 23.2 | 7.0 | 16.5 | 8.0 | 66.0 | 5.0 | 23.5 | 88.0 | 42.5 | 13.0 | 14.5 | 5.5 | 51.0 | 42.5 | 9.5 | 10.0 | 17.1 | 6.5 | 66.0 |
| | | 50.5 | 14.5 | 38.0 | 9.0 | 9.5 | 8.5 | 31.0 | 5.0 | 28.5 | 27.0 | 8.5 | 5.0 | 22.6 | 29.3 | 6.5 | 4.0 | 4.0 | 6.0 | 8.0 | 9.5 | 32.5 | 72.0 | 1.0 | 38.0 | 42.0 | 25.0 |
| MiniCPM-Llama3-V-2-5 | 21.6 | 41.1 | 11.8 | 13.2 | 8.7 | 5.0 | 11.3 | 47.8 | 38.5 | 7.0 | 3.0 | 6.5 | 77.0 | 7.5 | 18.5 | 41.5 | 10.0 | 5.0 | 9.5 | 20.0 | 70.5 | 51.0 | 13.5 | 4.5 | 6.7 | 7.0 | 83.5 |
| | | 46.0 | 24.5 | 26.0 | 4.5 | 20.5 | 12.0 | 43.0 | 0.0 | 25.0 | 44.5 | 0.0 | 1.5 | 34.2 | 38.3 | 6.0 | 8.5 | 5.5 | 9.5 | 4.5 | 24.5 | 14.5 | 0.5 | 22.0 | 32.5 | 15.0 | |
| LLaVA-v1.5-7b | 19.2 | 14.1 | 4.2 | 13.7 | 5.8 | 1.9 | 6.9 | 27.3 | 35.0 | 6.5 | 12.5 | 12.5 | 53.0 | 10.0 | 25.5 | 66.5 | 43.0 | 19.0 | 3.5 | 2.5 | 23.5 | 36.5 | 12.0 | 16.5 | 6.7 | 7.0 | 28.0 |
| | | 24.5 | 17.5 | 40.0 | 15.0 | 21.5 | 4.0 | 26.0 | 7.5 | 26.5 | 17.5 | 5.0 | 4.5 | 25.6 | 27.1 | 8.5 | 8.0 | 4.0 | 6.0 | 14.5 | 29.5 | 66.0 | 2.0 | 35.0 | 34.5 | 28.5 | |
| sharegpt4v-7b | 18.5 | 16.4 | 5.0 | 10.8 | 6.2 | 9.0 | 2.7 | 34.2 | 28.5 | 4.5 | 10.5 | 3.5 | 57.0 | 4.0 | 12.5 | 55.5 | 44.5 | 13.5 | 5.0 | 5.0 | 26.0 | 38.0 | 14.0 | 15.5 | 10.9 | 6.0 | 25.0 |
| | | 26.5 | 19.0 | 42.0 | 7.5 | 14.0 | 7.5 | 31.5 | 7.0 | 29.0 | 18.0 | 5.0 | 1.5 | 28.1 | 23.3 | 9.5 | 3.0 | 7.0 | 6.0 | 2.0 | 8.0 | 27.5 | 65.0 | 0.0 | 44.0 | 36.5 | 31.0 |
| sharecaptioner | 16.1 | 20.7 | 22.2 | 27.2 | 10.2 | 9.1 | 21.0 | 39.5 | 37.0 | 7.0 | 5.0 | 6.0 | 47.0 | 5.0 | 17.0 | 25.0 | 35.5 | 12.5 | 13.0 | 5.5 | 14.5 | 4.5 | 3.0 | 6.0 | 18.1 | 5.5 | 21.5 |
| | | 17.0 | 22.5 | 18.5 | 12.0 | 14.5 | 11.0 | 23.5 | 7.0 | 25.5 | 22.0 | 5.5 | 2.0 | 16.1 | 43.6 | 9.0 | 2.5 | 1.5 | 1.5 | 5.5 | 8.0 | 26.5 | 47.0 | 2.0 | 38.0 | 56.5 | 9.0 |
| monkey-chat | 13.7 | 8.4 | 8.0 | 5.9 | 9.2 | 6.7 | 8.1 | 25.3 | 4.5 | 6.0 | 1.5 | 34.5 | 2.0 | 9.0 | 40.5 | 40.5 | 12.0 | 2.5 | 6.5 | 16.5 | 14.5 | 10.0 | 12.5 | 18.1 | 6.5 | 19.5 | |
| | | 10.0 | 8.5 | 17.0 | 8.0 | 13.0 | 7.5 | 15.5 | 7.0 | 27.5 | 3.0 | 10.6 | 22.6 | 9.0 | 5.5 | 8.0 | 4.0 | 0.0 | 5.5 | 7.5 | 34.5 | 51.0 | 1.5 | 17.0 | 36.0 | 8.5 | |
| *Pure Text LLMs* | | | | | | | | | | | | | | | | | | | | | | | | | | | |
| GPT-4o | 23.1 | 32.5 | 0.0 | 0.0 | 32.5 | 0.0 | 0.0 | 15.0 | 20.0 | 30.0 | 37.5 | 0.0 | 0.0 | 90.0 | 0.0 | 0.0 | 45.0 | 0.0 | 20.0 | 0.0 | 25.0 | 50.0 | 0.0 | 0.0 | 40.0 | | |
| | | 55.0 | 40.0 | 70.0 | 0.0 | 15.0 | 20.0 | 30.0 | 35.0 | 0.0 | 40.0 | 10.0 | 0.0 | 10.0 | 0.0 | 0.0 | 35.0 | 20.0 | **100.0** | 65.0 | 25.0 | 30.0 | 35.0 | 0.0 | 60.0 | 20.0 | 0.0 |
| GPT-4o + Caption | 48.7 | 67.5 | 22.5 | 52.5 | 32.5 | 2.5 | 7.5 | 77.5 | 65.0 | 45.0 | 55.0 | 80.0 | 85.0 | 100.0 | 60.0 | 15.0 | 50.0 | 70.0 | 80.0 | 30.0 | 90.0 | 70.0 | 40.0 | 25.0 | 15.0 | 90.0 | |
| | | 75.0 | 35.0 | 75.0 | 10.0 | 85.0 | 60.0 | 5.0 | 50.0 | 65.0 | 20.0 | 0.0 | 55.0 | 55.0 | 25.0 | 30.0 | 45.0 | 95.0 | 80.0 | 25.0 | 45.0 | 50.0 | 5.0 | 60.0 | 45.0 | 45.0 | |
| Llama3.1 | 29.5 | 10.0 | 0.0 | 32.5 | 30.0 | 32.5 | 17.5 | 45.0 | 37.5 | 15.0 | 20.0 | 30.0 | 25.0 | 50.0 | 45.0 | 30.0 | 35.0 | 25.0 | 55.0 | 30.0 | 55.0 | 30.0 | 15.0 | 10.0 | 15.0 | 10.0 | |
| | | 50.0 | 15.0 | 65.0 | 30.0 | 20.0 | 15.0 | 45.0 | 25.0 | 10.0 | 50.0 | 25.0 | 30.0 | 20.0 | 35.0 | 30.0 | 40.0 | 20.0 | 55.0 | 25.0 | 15.0 | 35.0 | 15.0 | | | | |
| Llama3.1 + caption | 39.2 | 55.0 | 27.5 | 30.0 | 22.5 | 37.5 | 27.5 | 77.5 | 52.5 | 20.0 | 25.0 | 25.0 | 90.0 | 40.0 | 65.0 | 45.0 | 25.0 | 30.0 | 20.0 | 55.0 | 30.0 | 30.0 | 80.0 | 20.0 | 80.0 | | |
| | | 60.0 | 35.0 | 60.0 | 5.0 | 10.0 | 35.0 | 50.0 | 30.0 | 15.0 | 55.0 | 25.0 | 40.0 | 45.0 | 20.0 | 35.0 | 20.0 | 10.0 | 55.0 | 15.0 | 70.0 | 40.0 | 35.0 | | | | |
| InternLM2.5 | 31.5 | 12.5 | 35.0 | 17.5 | 27.5 | 27.5 | 32.5 | 45.0 | 52.5 | 5.0 | 25.0 | 65.0 | 20.0 | 55.0 | 55.0 | 15.0 | 35.0 | 60.0 | 25.0 | 40.0 | 65.0 | 10.0 | 40.0 | 25.0 | | | |
| | | 45.0 | 25.0 | 60.0 | 25.0 | 15.0 | 20.0 | 25.0 | 30.0 | 25.0 | 40.0 | 35.0 | 15.0 | 35.0 | 45.0 | 15.0 | 35.0 | 30.0 | 25.0 | 5.0 | 25.0 | | | | | | |
| InternLM2.5+Caption | 38.2 | 55.0 | 22.5 | 30.0 | 40.0 | 27.5 | 30.0 | 55.0 | 70.0 | 55.0 | 20.0 | 85.0 | 50.0 | 65.0 | 40.0 | 45.0 | 25.0 | 10.0 | 30.0 | 30.0 | 35.0 | 25.0 | 50.0 | | | | |
| | | 65.0 | 25.0 | 55.0 | 10.0 | 10.0 | 55.0 | 30.0 | 20.0 | 10.0 | 55.0 | 10.0 | 10.0 | 40.0 | 45.0 | 25.0 | 10.0 | 50.5 | 30.0 | 30.0 | 25.0 | 50.0 | 50.0 | 35.0 | | | |
| Qwen2.5 | 30.9 | 17.5 | 35.0 | 32.5 | 15.0 | 20.0 | 22.5 | 45.0 | 35.0 | 5.0 | 25.0 | 25.0 | 80.0 | 20.0 | 55.0 | 40.0 | 55.0 | 10.0 | 45.0 | 15.0 | 60.0 | 45.0 | 50.0 | 30.0 | | | |
| | | 70.0 | 25.0 | 75.0 | 25.0 | 15.0 | 25.0 | 40.0 | 5.0 | 40.0 | 35.0 | 5.0 | 10.0 | 70.0 | 15.0 | 25.0 | 50.0 | 45.0 | 30.0 | 30.0 | 35.0 | 45.0 | | | | | |
| Qwen2.5+Caption | 40.3 | 55.0 | 35.0 | 55.0 | 17.5 | 20.0 | 12.5 | 67.5 | 45.0 | 25.0 | 50.0 | 40.0 | 75.0 | 80.0 | 60.0 | 10.0 | 75.0 | 50.0 | 65.0 | 25.0 | 85.0 | 50.0 | 30.0 | 25.0 | 30.0 | | |
| | | 85.0 | 30.0 | 65.0 | 40.0 | 60.0 | 20.0 | 45.0 | 25.0 | 45.0 | 20.0 | 10.0 | 35.0 | 60.0 | 15.0 | 25.0 | 15.0 | 50.0 | 15.0 | 25.0 | 15.0 | 50.0 | 30.0 | 40.0 | 25.0 | | |

images, we insert them into the text to form a coherent prompt. The complete prompt is as shown in Table 13.

## B.3 Multitask Analysis

**Differences in model capabilities across various image relationships.** As shown in Figure 8, we visualize the average performance of all models across 7 specific image relationships. Detailed analysis can be found in Section 4.3 of the main text.

**Differences in model capabilities in testmini set.** To mitigate the computational cost caused by the large number of samples, we randomly sample 20 examples from each of the 52 tasks, resulting in a testmini set with 1,040 samples while preserving the original diversity of MMIU. As shown in Table 15, we evaluate model performance on the testmini set across seven types of image relationships.

Table 15: Quantitative results across 7 image relationships on testmini set are summarized. Accuracy is the metric, and the Overall score is computed across all tasks. The maximum value of each task is bolded.

| Model | Overall | Discrete | Continuous | Low-level | High-level-sub | High-level-obj | Two-D | Three-D |
|---|---|---|---|---|---|---|---|---|
| Frequency | 30.9 | 29.1 | 29.2 | 38.3 | 29.2 | 36.1 | 26.9 | 30.4 |
| Random | 27.1 | 22.3 | 24.9 | 33.2 | 21.1 | 32.9 | 24.5 | 28.3 |
| *Closed-source LVLMs* | | | | | | | | |
| GPT-4o | 55.6 | 56.7 | 55.6 | 87.5 | 75.0 | 60.0 | 37.8 | 53.6 |
| Claude3.5 | 54.3 | 51.7 | 50.6 | 81.2 | 59.2 | 66.3 | 40.6 | 49.1 |
| Gemini1.5 | 54.5 | 50.0 | 52.5 | 81.2 | 60.8 | 66.5 | 45.0 | 45.5 |
| Gemini1.0 | 41.2 | 48.3 | 50.0 | 57.5 | 63.3 | 37.5 | 28.9 | 36.4 |
| *Adequate Multi-Image SFT LVLMs* | | | | | | | | |
| Mantis-idefics2-8B | 45.3 | 40.0 | 43.1 | 53.8 | 59.2 | 52.5 | 37.8 | 42.3 |
| Mantis-SigCLIP-8B | 41.8 | 35.8 | 40.0 | 67.5 | 54.2 | 52.5 | 26.7 | 38.2 |
| LLaVA-Next-Interleave-7B | 33.5 | 35.0 | 33.1 | 38.8 | 35.0 | 34.4 | 35.0 | 29.1 |
| xGen-MM-interleaved-4B | 28.2 | 31.7 | 23.8 | 42.5 | 42.5 | 22.5 | 30.6 | 27.7 |
| *Multi-Image input LVLMs* | | | | | | | | |
| InternVL2-Pro | 49.8 | 59.2 | 45.6 | 73.8 | 75.8 | 57.3 | 37.2 | 37.7 |
| Internvl1.5-chat | 38.7 | 45.0 | 50.6 | 41.2 | 63.3 | 27.3 | 35.6 | 35.5 |
| InternVL2-8B | 34.0 | 35.0 | 40.0 | 33.8 | 46.7 | 30.6 | 27.2 | 35.5 |
| Mini-InternVL-chat-1.5-4B | 32.5 | 33.3 | 41.9 | 33.8 | 40.8 | 29.0 | 28.9 | 29.5 |
| Mini-InternVL-chat-1.5-2B | 31.8 | 26.7 | 44.4 | 27.5 | 42.5 | 28.5 | 25.0 | 32.7 |
| idefics2-8b | 27.2 | 29.2 | 16.9 | 40.0 | 45.8 | 26.7 | 26.1 | 27.7 |
| xGen-MM-single-4B | 25.4 | 25.8 | 25.6 | 18.8 | 34.2 | 23.8 | 27.2 | 24.1 |
| DeepSeek-VL-7b | 24.6 | 13.3 | 8.8 | 28.7 | 35.8 | 38.1 | 32.2 | 16.4 |
| DeepSeek-VL-1.3b | 23.8 | 9.2 | 9.4 | 32.5 | 31.7 | 32.7 | 32.2 | 21.4 |
| XComposer2-7b | 23.4 | 29.2 | 33.1 | 26.2 | 37.5 | 19.6 | 18.9 | 16.8 |
| flamingov2 | 22.7 | 17.5 | 21.9 | 27.5 | 25.8 | 30.2 | 18.9 | 18.6 |
| XComposer2-1.8b | 22.0 | 30.0 | 33.1 | 20.0 | 39.2 | 18.3 | 18.9 | 12.3 |
| Qwen-chat | 18.0 | 13.3 | 21.9 | 26.2 | 27.5 | 17.1 | 11.7 | 20.0 |
| idefics-9b-instruct | 13.2 | 26.7 | 5.6 | 12.5 | 35.8 | 9.0 | 11.1 | 11.8 |
| Qwen-Base | 4.8 | 14.2 | 0.6 | 5.0 | 9.2 | 2.9 | 3.3 | 5.0 |
| *Single-Image input LVLMs* | | | | | | | | |
| GLM-4v-9b | 26.6 | 20.8 | 30.6 | 55.0 | 37.5 | 31.3 | 16.7 | 21.4 |
| LLaVA-next-vicuna-7b | 22.9 | 23.3 | 25.0 | 28.7 | 39.2 | 21.5 | 18.3 | 20.9 |
| MiniCPM-Llama3-V-2-5 | 21.7 | 20.0 | 32.5 | 27.5 | 40.8 | 14.8 | 10.6 | 25.9 |
| LLaVA-v1.5-7b | 18.7 | 20.8 | 16.2 | 17.5 | 31.7 | 19.6 | 16.1 | 16.8 |
| sharegpt4v-7b | 18.4 | 20.0 | 17.5 | 21.2 | 32.5 | 16.7 | 16.1 | 17.7 |
| sharecaptioner | 16.4 | 15.8 | 10.0 | 37.5 | 31.7 | 16.5 | 15.6 | 14.1 |
| monkey-chat | 14.3 | 10.0 | 13.8 | 11.2 | 25.0 | 13.7 | 17.2 | 13.2 |
| *Pure text LLMs* | | | | | | | | |
| GPT-4o | 21.8 | 29.2 | 21.9 | 6.2 | 45.8 | 12.7 | 21.7 | 25.0 |
| + Caption | 48.9 | 68.3 | 53.8 | 67.5 | 60.0 | 50.2 | 30.0 | 46.8 |
| Qwen2.5 | 30.1 | 24.2 | 33.8 | 33.8 | 39.2 | 33.7 | 28.3 | 25.0 |
| + Caption | 41.1 | 45.8 | 49.4 | 57.5 | 55.8 | 40.6 | 31.7 | 33.6 |
| InternLM2.5 | 32.1 | 32.3 | 33.4 | 40.2 | 51.5 | 30.5 | 26.2 | 30.8 |
| + Caption | 39.8 | 41.7 | 45.0 | 47.5 | 46.7 | 39.0 | 36.1 | 35.5 |
| Llama3.1 | 30.1 | 30.0 | 32.5 | 36.3 | 21.7 | 28.8 | 26.1 | 34.5 |
| + Caption | 38.3 | 42.5 | 46.2 | 53.8 | 39.2 | 38.1 | 28.9 | 35.0 |

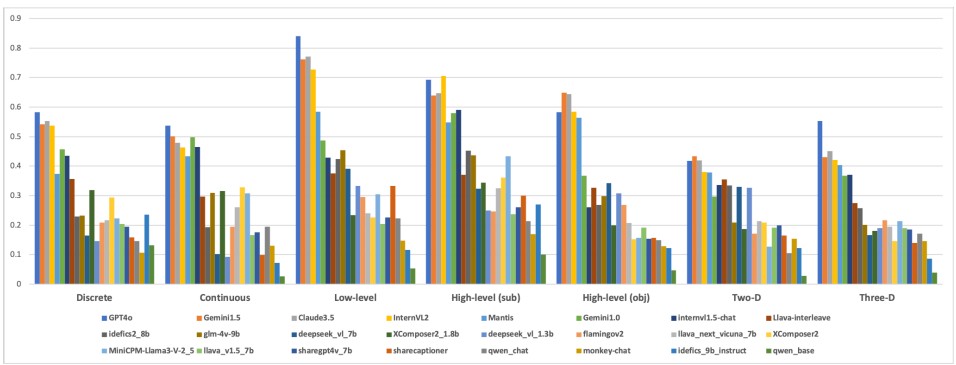

Figure 8: The average performance comparison of some LVLMs on seven specific image relationships on test set.

The results demonstrate that performance on the testmini set effectively reflects the trends observed in the full test set.

**The questions in MMIU are visual-dependent.** MMIU, as a comprehensive benchmark for evaluating multi-image understanding capabilities, heavily relies on accurate visual comprehension and cannot be solved using text alone, demonstrating distinct multimodal characteristics. Specifically, as shown in Table 3, we observe that when using state-of-the-art large language models (LLMs) with only questions and options as input, their accuracy is comparable to random selection. This indicates that MMIU fundamentally depends on visual information and that the design of distractor options is robust, preventing inference of the correct answer based solely on the options. Furthermore, we investigate extracting captions for each image using GPT-4o and then performing reasoning with the LLM. Experimental results show a significant improvement using this method, highlighting that accurate image understanding is foundational for solving MMIU. However, even with this approach, performance remains inferior to directly leveraging images for answering. For instance, the accuracy of GPT-4o with direct image input (55.5%) surpasses the performance achieved by caption extraction followed by reasoning (48.7%). This demonstrates that certain tasks in MMIU require visual processing rather than purely semantic understanding.

**Why some models are less accurate than random selection.** In MMIU, as shown in Table 3, the accuracy of some models is even lower than that of random selection. Similar result also exist in MuirBench (Wang et al., 2024) and MilesBench (Song et al., 2024) We have conducted a detailed investigation and analyzed the reasons.

**I) For multi-image input models, errors occur frequently due to limitations in model capability or instruction-following ability**, leading to significantly lower accuracy. We summarize the probability of "Z" answers for several models in the Table 16. Our findings are as follows: **1) The highest probability of generating "Z" answers occurs in spatial relationship tasks.** This is because spatial data is relatively scarce, leading to insufficient model training. Additionally, spatial tasks tend to be more complex (some involve visual tasks), and models with limited spatial ability tend to generate irrelevant responses. As shown in Table 3, even for powerful closed-source models, performance on spatial tasks remains below the overall average. **2) The lowest probability of "Z" answers occurs in temporal relationship tasks.** This is because video data is the most readily available during training, enabling models to develop relatively strong capabilities in this area. As shown in Table 3, for closed-source models, performance on temporal tasks tends to be above the overall average. **3) Instruction-following ability is equally important.** As the model size decreases, instruction-following ability weakens, leading to off-topic errors. In the error case analysis in Section B.4, Mantis' instruction-following ability is significantly lower than that of closed-source models and larger open-source models like InternVL2-Pro. This results in situations where the model provides answers not in the options. **II) For single-image models, lower accuracy or random selection errors primarily result from insufficient model capability or difficulty in generalizing long contexts.** We summarize the probability of "Z" answers for models in the Table 16. **1) When we use the concatenated image method**, we find that "Z" answers are less frequent in single-image models. Instead, these models tend to select incorrect answers. This is because the models struggle to interpret concatenated images. **2) When we test with the concatenated token method**, we observe similar issues to multi-image models. This is due to long context causing off-topic or irrelevant responses. For typical single-image models, even if long context input is supported, the training samples mostly consist of single images, which limits generalization. Specifically, we calculate the average number of images per class for different image relationships: $10.02$ for Temporal, $5.94$ for Semantic, and $7.04$ for Spatial. As the number of images increases, the probability of generating "Z" answers also increases.

Table 16: Performance comparison of various models with multi-image input data on different tasks.

| Model | Temporal | Semantic | Spatial | Overall |
|---|---|---|---|---|
| XComposer2-7b | 13.6 | 20.1 | 46.1 | 28.3 |
| flamingov2 | 19.1 | 10.7 | 17.3 | 15.5 |
| xgen-mm-single | 15.1 | 29.8 | 35.8 | 27.9 |
| llava-v1.5-7b (concat img) | 0.6 | 0.1 | 0.9 | 0.5 |
| sharecaptioner (concat img) | 15.1 | 3.3 | 8.6 | 8.5 |
| llava-v1.5-7b (concat token) | 55.0 | 5.1 | 38.1 | 29.5 |
| sharecaptioner (concat token) | 38.9 | 25.0 | 34.1 | 32.2 |

**High resolution and dynamic resolution matter.** High resolution and dynamic resolution are critical factors. Research has shown that increasing resolution and employing dynamic resolution training significantly enhance image understanding performance (Lin et al., 2023) This is because low resolution tends to overlook critical visual details, while training with a fixed resolution can lead to image distortion or inefficiencies in training and inference.In multi-image tasks, we observe that top-performing models (e.g., InternVL2-8B, Mantis-idefics2, LLaVA-Next-Interleave) consistently adopt high-resolution settings (e.g., 448, 384, and 336 respectively) combined with dynamic resolution training strategies. This approach allows the models to better comprehend image content and adapt to various input scales.

Table 17: Performance comparison of various models with multi-image training data in different stage on MMIU.

| Model | Multi-image Pretrained | Multi-image SFT | Size | Semantic | Temporal | Spatial | Overall |
|---|---|---|---|---|---|---|---|
| Mini-InternVL1.5-chat | No | No | 2B | 30.4 | 36.2 | 26.5 | 30.5 |
| Mini-InternVL1.5-chat | No | No | 4B | 32.2 | 36.8 | 28.5 | 32.1 |
| InternVL1.5-chat | No | No | 26B | 33.4 | 45.2 | 35.5 | 37.4 |
| LLaVA-Next | No | No | 7B | 23.0 | 24.1 | 20.3 | 22.2 |
| LLaVA-Next-Interleave | No | Yes | 7B | 35.7 | 32.2 | 31.1 | 32.4 |
| Mantis-SigCLIP | No | Yes | 8B | 53.6 | 38.4 | 35.7 | 42.6 |
| xGen-MM-Phi3 | Yes | No | 4B | 25.1 | 25.7 | 23.1 | 24.5 |
| idefics2 | Yes | No | 8B | 31.6 | 20.9 | 29.2 | 27.8 |
| InternVL2 | Yes | Only video | 8B | 35.2 | 39.5 | 31.2 | 34.8 |
| Mantis-idsfics2 | Yes | Yes | 8B | 56.3 | 40.8 | 39.2 | 45.6 |
| xGen-MM-Phi3-Interleaved | Yes | Yes | 4B | 28.2 | 28.7 | 29.1 | 28.7 |

**Powerful vision encoder matters.** The InternVL series models demonstrate outstanding performance on MMIU. For instance, both Mini-InternVL-Chat-1.5-4B and xGen-MM-Interleave-4B use Phi3-mini (Abdin et al., 2024) as their LLM, yet InternVL achieves significantly better results. Notably, the performance of Mini-InternVL-Chat-1.5-4B even matches that of LLaVA-Next-Interleave-7B. We attribute this advantage partly to InternViT (Chen et al., 2024b), the vision encoder trained for InternVL. As a powerful encoder, InternViT outperforms various CLIP-releated models across multiple benchmarks. This highlights the critical role of a strong vision encoder in improving performance on multi-image tasks.

**Discussion on the effectiveness of multi-image training.** As shown in Table 17, by categorizing these representative models based on whether multi-image data is included during pretraining or SFT, we observe the following: **1) Multi-image SFT effectively improves performance regardless of whether multi-image pretraining is conducted.** On the one hand, LLaVA-Next-Interleave, which applies interleaved visual instruction tuning on LLaVA-Next (without multi-image pretraining), still achieves performance gains. On the other hand, xGen-MM-phi3-Interleaved, built on xGen-MM-phi3 with multi-image pretraining, also benefits significantly from multi-image SFT. **2) Multi-image SFT achieves greater improvements when combined with multi-image pretraining.** Both Mantis-SigCLIP and Mantis-idesfics2 use the same multi-image SFT dataset, but the latter outperforms the former because its model incorporates multi-image pretraining. **3) Multi-image pretraining alone is insufficient.** Models such as xGen-MM-phi3 [†] and idefics2, which only undergo multi-image pretraining without substantial multi-image SFT, do not outperform models with just multi-image SFT (e.g., LLaVA-Next-Interleave, Mantis-SigCLIP). This indicates that the potential of multi-image pretraining requires multi-image SFT to be fully realized. **4) Models still exhibit some generalization to multi-image tasks even without multi-image data.** Surprisingly, the Mini-InternVL1.5 series, trained without any multi-image data, performs relatively well on multi-image tasks. This can be attributed to InternViT as a strong visual encoder and configurations like dynamic resolution, enabling superior image understanding (e.g., Mini-InternVL1.5-chat-4B matches the performance of LLaVA-Next-34B on MME (Fu et al., 2023)). However, these models still lag significantly behind those SFT with multi-image data (e.g., Mantis).

**Practical Applicability Discussion.** MMIU involves plenty of tasks highly related to real-life applications. To this end, we summarize representative LVLMs' performance on tasks highly related to practical applications in Table 18 and detail our discussions from the following four perspectives. 1) **MMIU provides extensive exploration of the visual capabilities needed for high-level planning in embodied systems and their application in robotics**, making it highly significant in practical terms. For instance, as shown in Table 18, on the task of Egocentric Video Question Answering

---

[†]xGen-MM-Phi3 has only a few multi-image SFT data related to spot the difference

Table 18: Performance of models on various tasks related to practical applications.

| Model | Egocentric Video Question Answering | 3D Depth Estimation | Vehicle Retrieval | Person Re-ID | GUI App Recognition | GUI Next Action Prediction |
|---|---|---|---|---|---|---|
| GPT-4o | 57.5 | 24.0 | 68.0 | 61.5 | 73.5 | 46.5 |
| Claude3.5-Sonnect | 53.5 | 23.5 | 79.0 | 77.5 | 88.5 | 55.0 |
| Mantis-idefics2-8B | 54.5 | 23.5 | 70.5 | 57.5 | 78.0 | 34.0 |
| InternVL2-Pro | 63.0 | 25.5 | 83.5 | 82.0 | 91.5 | 40.5 |

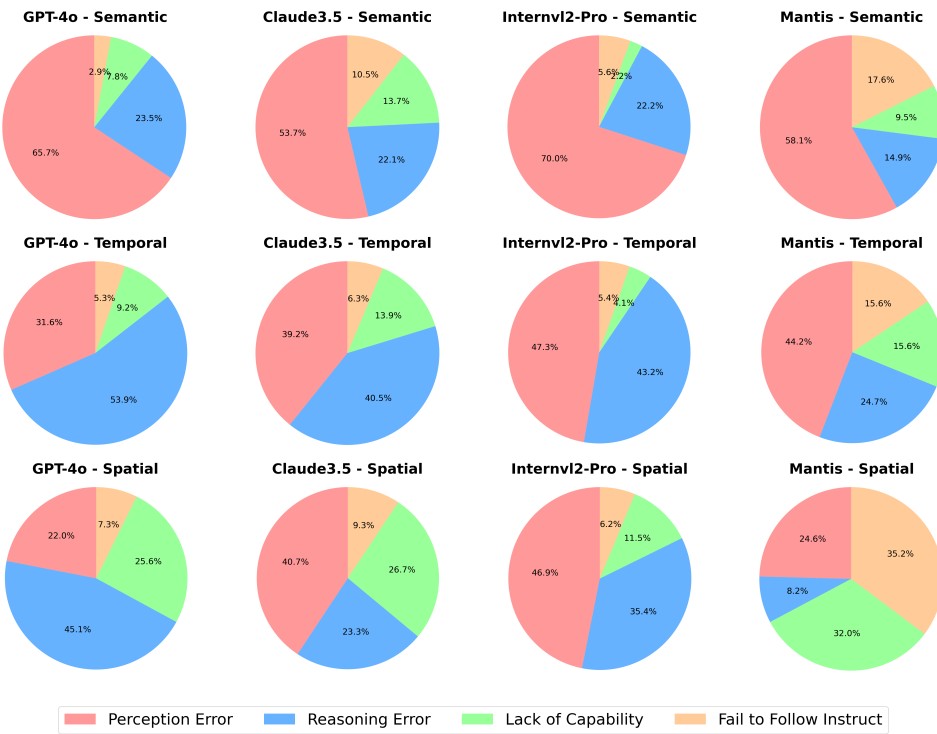

Figure 9: Distribution of error types for GPT-4o, Claude3.5 Sonnet, InternVL2-Pro and Mantis-idesifcs2 towards three main image relationship.

(EVQA), InternVL2-Pro outputformed other LVLMs, demonstrates its development potential for physical interactions with the environment. **2) MMIU also contains spatial tasks playing pivotal roles in autonomous driving.** For example, 3D depth estimation can provide vehicles with accurate, real-time spatial awareness and situational understanding of dynamic environments. Among our evaluation results in Table 18, we find that most LVLMs faced difficulties delivering strong performance in this task, indicating their limitations on spatial reasoning capabilities. **3) MMIU consists of semantic tasks within the surveillance domain**, such as Vehicle Re-Identification and Person Re-Identification, which plays a crucial role in enhancing the effectiveness and capabilities of surveillance systems. In our evaluations of Table 18, InternVL2-Pro outputformed other LVLMs in these two tasks, including the closed-source model GPT-4o and the multi-image SFT model Mantis, showcasing its potential applications within the surveillance domain. **4) MMIU has significant practival value for autonomous control applications.** Because it includes temporal tasks like gui app recognition and gui next action prediction. In Table 18, we find that Claude3.5 demontrate relatively better performance compared to other LVLMs in these tasks, highlighting its significant growth potential for future development. In conclusion, MMIU provides a comprehensive benchmark encompassing a diverse range of tasks, offering critical insights into the strengths and limitations of current LVLMs across spatial, semantic, and temporal dimensions, ultimately driving advancements in real-world applications.

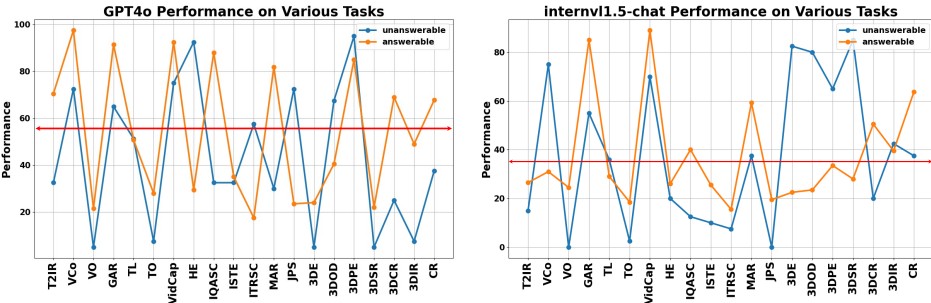

Figure 10: Comparison of GPT-4o and InternVL1.5 on unanswerable and answerable questions, with the red line representing the model's average accuracy across all tasks.

## B.4  ERROR ANALYSIS

To more vividly demonstrate the performance of MMIU and the models on different tasks, we analyze error cases for the four best-performing models: GPT-4o, Claude3.5-Sonnet, InternVL2-Pro and Mantis-idefics2. Through an analysis of error types in MMIU, we categorize them into four types: Fail to Follow Instruction, Lack of Capability, Perception Error, and Reasoning Error. Specifically, **Perception Error** refers to instances where LVLMs struggle with object recognition, classification, or detection in images due to limitations in the representational capacity of their visual encoders, making it the most common error. **Reasoning Error** occurs when models correctly recognize and interpret visual content, but errors arise during the reasoning process, leading to incorrect answers. **Lack of Capability** refers to cases where models refuse to answer questions, citing insufficient information despite having all necessary inputs. **Fail to Follow Instruction** indicates that LVLMs misinterpret instructions, resulting in incorrect responses.

As illustrated in Figure 9, we make the following observations regarding different types of visual relationships (e.g., semantic, temporal, and spatial relationships): **1) Semantic tasks tend to exhibit more perception errors compared to temporal and spatial relationships.** We attribute this to the greater diversity often inherent in semantic tasks (e.g., image retrieval), which poses additional challenges for multi-image understanding. In contrast, temporal and spatial tasks typically involve images with inherent consistency, such as variations in time or viewpoint. **2) Temporal and spatial tasks are more prone to reasoning errors.** Temporal tasks often require models to infer the dynamics of objects over time (e.g., ordering tasks or next image prediction), while spatial tasks demand spatial cognition and reasoning (e.g., Image-Captioning-with-Spatial-Context). We observe that models consistently struggle with these forms of reasoning. **3) Spatial tasks exhibit a significant number of errors caused by lack of capacity.** This is particularly evident in tasks requiring advanced visual understanding, such as 3D pose estimation. Current models often produce irrelevant or incoherent responses in these tasks, likely due to insufficient training for specialized visual challenges. On the other hand, when analyzing performance across models, we find the following: **4) Open-source models are more prone to perception errors compared to closed-source models.** As discussed in Appendix B.3, we believe the capability of the visual encoder is critical for multi-image understanding. Enhancing the performance of visual encoders could effectively mitigate these errors. **5) Smaller open-source models (e.g., Mantis) are more likely to fail to follow instructions.** This can be attributed to the limited size of the language model, which impairs its ability to handle long-context tasks involving multiple images. This suggests that improving LLM capacity, particularly for long-context understanding, is crucial for better performance in multi-image tasks.

Finally, for each type of image relationship, we visualize one error case. Since the model responses or questions might be lengthy, we mostly extract the important parts. The images are shown in Figure 12, Figure 13, Figure 14, Figure 15, Figure 16, Figure 17, and Figure 18.

## B.5  ABLATION STUDY

**Impact of Unanswerable Questions on Model Performance.** We have constructed 19 tasks, each including 40 questions. We tested a series of models on these questions, with full results refer-

Table 19: Quantitative results for LVLMs across 21 tasks with unanswerable and answerable questions are summarized. Accuracy is the metric, and the Overall score is computed across all tasks.

| Model | Overall | T2IR | VCo | VO | GAR | TL | TO | VidCap | HE | IQASC | ISTE | ITRSC | MAR | JPS | 3DE | 3DOD | 3DPE | 3DSR | 3DCR | 3DIR | CR |
|---|---|---|---|---|---|---|---|---|---|---|---|---|---|---|---|---|---|---|---|---|---|
| **Answerable** | | | | | | | | | | | | | | | | | | | | | |
| GPT-4o | 54.2 | 70.5 | 97.5 | 21.5 | 91.5 | 50.8 | 28.0 | 92.5 | 29.5 | 88.0 | 35.0 | 17.5 | 81.9 | 23.5 | 24.0 | 40.5 | 85.0 | 22.0 | 69.0 | 49.0 | 67.8 |
| Claude3.5 | 49.5 | 65.5 | 67.5 | 38.5 | 80.5 | 43.5 | 23.0 | 91.0 | 23.0 | 78.5 | 32.0 | 4.5 | 64.8 | 31.5 | 23.5 | 41.0 | 99.5 | 21.5 | 53.5 | 36.5 | 70.2 |
| Mantis | 38.9 | 46.5 | 14.0 | 20.0 | 81.0 | 23.8 | 27.0 | 85.0 | 23.0 | 66.0 | 23.5 | 13.0 | 71.4 | 27.5 | 23.5 | 24.0 | 22.5 | 25.0 | 59.0 | 40.5 | 61.5 |
| internvl1.5-chat | 37.5 | 26.5 | 31.0 | 24.5 | 85.0 | 29.0 | 18.5 | 89.0 | 26.0 | 40.0 | 25.5 | 15.5 | 59.3 | 19.5 | 22.5 | 23.5 | 33.5 | 28.0 | 50.5 | 39.5 | 63.7 |
| idefics2-8b | 24.0 | 12.5 | 21.5 | 22.5 | 24.5 | 22.3 | 18.0 | 19.5 | 21.5 | 31.0 | 25.5 | 13.5 | 15.1 | 27.5 | 26.0 | 21.5 | 21.5 | 23.0 | 44.5 | 40.5 | 28.0 |
| GLM-4v-9b | 23.3 | 25.0 | 11.5 | 14.5 | 59.5 | 25.9 | 10.0 | 76.0 | 11.5 | 35.5 | 16.0 | 6.5 | 25.1 | 9.0 | 14.0 | 14.5 | 0.5 | 5.5 | 48.5 | 23.5 | 32.8 |
| DeepSeek-VL-7b | 19.3 | 25.5 | 30.5 | 21.5 | 6.0 | 31.1 | 15.5 | 2.0 | 23.0 | 45.5 | 24.5 | 0.0 | 2.0 | 20.5 | 24.5 | 24.5 | 7.5 | 0.5 | 40.5 | 38.5 | 2.2 |
| XComposer2-1.8b | 19.5 | 10.5 | 28.5 | 17.0 | 59.5 | 24.4 | 13.0 | 68.5 | 0.5 | 29.5 | 6.0 | 7.5 | 33.2 | 7.0 | 0.0 | 15.5 | 28.0 | 2.0 | 11.5 | 3.0 | 24.5 |
| DeepSeek-VL-1.3b | 20.1 | 23.0 | 33.0 | 20.0 | 1.0 | 26.4 | 20.0 | 1.0 | 25.0 | 44.5 | 24.0 | 1.0 | 0.0 | 31.0 | 26.0 | 31.0 | 19.5 | 0.0 | 45.5 | 29.0 | 1.2 |
| flamingov2 | 19.0 | 20.0 | 14.5 | 13.5 | 13.5 | 18.7 | 24.5 | 22.5 | 27.5 | 4.0 | 23.0 | 7.0 | 22.1 | 1.5 | 26.5 | 22.0 | 17.0 | 28.5 | 25.0 | 23.5 | 25.5 |
| LLaVA-next-vicuna-7b | 18.1 | 13.0 | 9.5 | 8.5 | 51.0 | 17.1 | 6.5 | 66.0 | 5.0 | 28.5 | 8.5 | 5.0 | 22.6 | 6.5 | 4.0 | 4.0 | 8.0 | 9.5 | 42.0 | 25.0 | 22.2 |
| XComposer2 | 17.8 | 9.5 | 2.5 | 19.0 | 59.5 | 15.5 | 12.0 | 66.0 | 8.0 | 15.5 | 0.0 | 0.0 | 20.6 | 16.5 | 0.0 | 7.0 | 4.5 | 0.0 | 42.0 | 33.0 | 24.0 |
| MiniCPM-Llama3-V-2-5 | 20.6 | 10.0 | 20.5 | 12.0 | 70.5 | 17.6 | 5.0 | 83.5 | 0.0 | 25.0 | 0.0 | 1.5 | 34.2 | 6.0 | 8.5 | 5.5 | 20.0 | 4.5 | 32.5 | 15.0 | 41.1 |
| LLaVA-v1.5-7b | 14.8 | 19.0 | 21.5 | 4.0 | 23.5 | 6.7 | 7.0 | 28.0 | 7.5 | 26.5 | 5.0 | 4.5 | 25.6 | 8.5 | 8.0 | 4.0 | 6.0 | 14.5 | 34.5 | 28.5 | 14.1 |
| sharegpt4v-7b | 14.3 | 13.5 | 14.0 | 7.5 | 26.0 | 10.9 | 6.0 | 25.0 | 7.0 | 29.0 | 5.0 | 1.5 | 28.1 | 9.5 | 3.0 | 7.0 | 2.0 | 8.0 | 36.5 | 31.0 | 16.4 |
| sharecaptioner | 11.3 | 12.5 | 14.5 | 11.0 | 14.5 | 18.1 | 5.5 | 21.5 | 7.0 | 25.5 | 5.5 | 2.0 | 16.1 | 9.0 | 2.5 | 1.5 | 5.5 | 8.0 | 16.5 | 9.0 | 20.7 |
| Qwen-chat | 15.5 | 3.0 | 6.0 | 6.0 | 52.5 | 3.6 | 5.5 | 47.0 | 9.0 | 13.5 | 15.5 | 3.5 | 40.2 | 16.5 | 16.5 | 22.5 | 13.0 | 14.5 | 1.5 | 0.5 | 20.5 |
| monkey-chat | 11.8 | 12.0 | 13.0 | 7.5 | 16.5 | 18.1 | 6.5 | 19.5 | 7.0 | 27.5 | 5.5 | 3.0 | 10.6 | 9.0 | 5.5 | 8.0 | 5.5 | 7.5 | 36.0 | 8.5 | 8.4 |
| idefics-9b-instruct | 6.1 | 3.5 | 0.0 | 5.5 | 1.5 | 0.5 | 3.0 | 10.0 | 3.0 | 9.0 | 0.0 | 0.0 | 6.5 | 1.0 | 15.5 | 10.5 | 36.5 | 5.5 | 0.0 | 0.0 | 10.8 |
| Qwen-Base | 3.7 | 1.0 | 5.5 | 6.5 | 4.5 | 0.0 | 0.0 | 7.5 | 2.0 | 8.5 | 0.0 | 0.0 | 0.5 | 0.0 | 0.5 | 7.0 | 21.5 | 0.0 | 0.0 | 0.0 | 9.2 |
| **Unanswerable** | | | | | | | | | | | | | | | | | | | | | |
| GPT-4o | 43.4 | 32.5 | 72.5 | 5.0 | 65.0 | 51.3 | 7.5 | 75.0 | 92.5 | 32.5 | 32.5 | 57.5 | 30.0 | 72.5 | 5.0 | 67.5 | 95.0 | 5.0 | 25.0 | 7.5 | 37.5 |
| Claude3.5 | 36.2 | 45.0 | 80.0 | 0.0 | 62.5 | 38.5 | 0.0 | 70.0 | 65.0 | 35.0 | 30.0 | 62.2 | 62.5 | 5.0 | 2.5 | 32.5 | 42.5 | 5.0 | 20.0 | 7.5 | 58.8 |
| Mantis | 48.3 | 17.5 | 7.5 | 22.5 | 62.5 | 61.5 | 0.0 | 80.0 | 75.0 | 37.5 | 50.0 | 2.5 | 35.0 | 2.5 | 87.5 | 67.5 | 57.5 | 57.5 | 82.5 | 87.5 | 72.5 |
| internvl1.5-chat | 37.7 | 15.0 | 75.0 | 0.0 | 55.0 | 35.9 | 2.5 | 70.0 | 20.0 | 12.5 | 10.0 | 7.5 | 37.5 | 0.0 | 82.5 | 80.0 | 65.0 | 85.0 | 20.0 | 42.5 | 37.5 |
| idefics2-8b | 26.4 | 5.0 | 90.0 | 0.0 | 30.0 | 20.5 | 2.5 | 32.5 | 0.0 | 45.0 | 2.5 | 27.5 | 25.0 | 0.0 | 100.0 | 27.5 | 0.0 | 32.5 | 20.0 | 32.5 | 35.0 |
| GLM-4v-9b | 9.1 | 5.0 | 10.0 | 0.0 | 12.5 | 28.2 | 0.0 | 30.0 | 0.0 | 15.0 | 0.0 | 7.5 | 0.0 | 0.0 | 0.0 | 12.5 | 0.0 | 0.0 | 10.0 | 20.0 | 31.2 |
| DeepSeek-VL-7b | 20.0 | 0.0 | 2.5 | 2.5 | 37.5 | 0.0 | 0.0 | 22.5 | 2.5 | 25.0 | 0.0 | 27.5 | 27.5 | 0.0 | 67.5 | 22.5 | 25.0 | 5.0 | 42.5 | 77.5 | 12.5 |
| XComposer2-1.8b | 39.9 | 17.5 | 0.0 | 2.5 | 25.0 | 2.6 | 32.5 | 47.5 | 52.5 | 47.5 | 47.5 | 27.5 | 27.5 | 2.5 | 92.5 | 62.5 | 55.0 | 27.5 | 95.0 | 95.0 | 38.8 |
| DeepSeek-VL-1.3b | 21.6 | 2.5 | 27.5 | 7.5 | 22.5 | 0.0 | 10.0 | 10.0 | 0.0 | 17.5 | 15.0 | 15.0 | 30.0 | 0.0 | 37.5 | 35.0 | 25.0 | 7.5 | 77.5 | 75.0 | 16.2 |
| flamingov2 | 23.2 | 5.0 | 17.5 | 2.5 | 72.5 | 0.0 | 0.0 | 62.5 | 35.0 | 42.5 | 0.0 | 42.5 | 20.0 | 0.0 | 27.5 | 0.0 | 0.0 | 12.5 | 40.0 | 32.5 | 51.2 |
| LLaVA-next-vicuna-7b | 6.6 | 0.0 | 0.0 | 0.0 | 27.5 | 0.0 | 0.0 | 30.0 | 0.0 | 0.0 | 0.0 | 0.0 | 7.5 | 0.0 | 0.0 | 0.0 | 0.0 | 0.0 | 10.0 | 47.5 | 8.8 |
| XComposer2 | 38.0 | 35.0 | 50.0 | 12.5 | 52.5 | 35.9 | 0.0 | 70.0 | 30.0 | 65.0 | 5.0 | 30.0 | 80.0 | 0.0 | 47.5 | 22.5 | 25.0 | 42.5 | 47.5 | 50.0 | 60.0 |
| MiniCPM-Llama3-V-2-5 | 13.1 | 5.0 | 15.0 | 0.0 | 42.5 | 23.1 | 0.0 | 47.5 | 15.0 | 7.5 | 10.0 | 0.0 | 5.0 | 0.0 | 0.0 | 0.0 | 0.0 | 0.0 | 47.5 | 20.0 | 23.8 |
| LLaVA-v1.5-7b | 19.1 | 0.0 | 0.0 | 0.0 | 50.0 | 0.0 | 0.0 | 37.5 | 0.0 | 0.0 | 20.0 | 12.5 | 15.0 | 0.0 | 30.0 | 12.5 | 2.5 | 10.0 | 60.0 | 85.0 | 46.2 |
| sharegpt4v-7b | 19.1 | 0.0 | 0.0 | 0.0 | 40.0 | 0.0 | 0.0 | 15.0 | 7.5 | 12.5 | 17.5 | 17.5 | 10.0 | 0.0 | 32.5 | 5.0 | 0.0 | 15.0 | 77.5 | 87.5 | 43.8 |
| sharecaptioner | 10.8 | 0.0 | 0.0 | 0.0 | 2.5 | 0.0 | 0.0 | 10.0 | 0.0 | 2.5 | 0.0 | 0.0 | 2.5 | 0.0 | 0.0 | 12.5 | 0.0 | 0.0 | 70.0 | 80.0 | 36.2 |
| Qwen-chat | 22.0 | 12.5 | 10.0 | 0.0 | 60.0 | 25.6 | 25.0 | 60.0 | 7.5 | 10.0 | 17.5 | 35.0 | 17.5 | 20.0 | 32.5 | 20.0 | 17.5 | 22.5 | 12.5 | 2.5 | 31.2 |
| monkey-chat | 12.2 | 0.0 | 0.0 | 0.0 | 2.5 | 0.0 | 0.0 | 20.0 | 7.5 | 12.5 | 0.0 | 2.5 | 15.0 | 0.0 | 25.0 | 5.0 | 0.0 | 10.0 | 42.5 | 75.0 | 26.2 |
| idefics-9b-instruct | 56.1 | 77.5 | 52.5 | 85.0 | 80.0 | 86.8 | 60.0 | 87.5 | 30.0 | 57.5 | 45.0 | 97.5 | 60.0 | 27.5 | 42.5 | 7.5 | 27.5 | 12.5 | 67.5 | 60.0 | 57.5 |
| Qwen-Base | 44.4 | 72.5 | 67.5 | 27.5 | 65.0 | 79.5 | 60.0 | 45.0 | 47.5 | 50.0 | 30.0 | 65.0 | 95.0 | 62.5 | 17.5 | 30.0 | 5.0 | 0.0 | 2.5 | 5.0 | 61.3 |

enced in Table 19 in the Appendix. As shown in Figure 10, we selected GPT-4o and InternVL1.5 as representative models for analysis. We observed that for some tasks where the models generally performed well, such as GAR (General Action Recognition), both GPT-4o and InternVL1.5 experienced performance degradation. However, for tasks that are inherently challenging for the models, as indicated by tasks below the red line in the figure, there is no significant pattern in the change of accuracy between answerable and unanswerable questions. We believe the reasons are as follows. 1) For tasks with high accuracy, introducing unanswerable questions confuses the models, increasing difficulty and thereby reducing accuracy. 2) For tasks with low accuracy, since the models already struggle with the original questions, the addition of unanswerable options might lead the models to directly choose the unanswerable option when uncertain, or the increased difficulty might further hinder their performance.

**Impact of Different Testing Methods on Model Performance.** We test the effectiveness of single-image input models in completing multi-image tasks using concatenated visual tokens or concatenated images, and we record the final results in Table 20.

## C    TASK MAP

A task map determines the similarity between tasks based on their inherent characteristics. By combining the task map with model performance, we aim to analyze which types of tasks current models perform well or poorly on. This approach avoids the bias introduced by using meta-task analysis alone, providing a more comprehensive conclusion through complementary methods. Thanks to the extensive number of tasks in MMIU, we can construct a comprehensive multi-image task map. As

Table 20: Quantitative results for single-image LVLMs across 52 mtasks with token-concat or image-concat are summarized. Accuracy is the metric, and the Overall score is computed across all tasks.

| Model | Overall | CR GuAR | ER GNAP | FD TC | FC VClz | SC VCo | VCor VO | VQA EVQA | VGR HE | FR IQASC | HR ICSC | I2IR ISTE | MIC ITRSC | PR MAR | S2IR MR | STD JPS | STS 3DE | T2IR 3DOD | VR 3DOT | AQA 3DPE | GAR 3DSR | MVU 3DQA | MEV PT | NIP RPM | TL SOT | TO 3DCR | VidCap 3DIR |
|---|---|---|---|---|---|---|---|---|---|---|---|---|---|---|---|---|---|---|---|---|---|---|---|---|---|---|---|
| **image-concat** | | | | | | | | | | | | | | | | | | | | | | | | | | | |
| GLM-4v-9b | 27.0 | 32.8 | 16.0 | 31.8 | 8.7 | 9.0 | 4.7 | 59.0 | 55.8 | 31.0 | 7.5 | 19.5 | 82.0 | 23.5 | 24.5 | 81.0 | 67.0 | 25.0 | 30.0 | 7.0 | 59.5 | 53.5 | 10.5 | 5.0 | 25.9 | 10.0 | 76.0 |
| | | 55.5 | 19.0 | 34.0 | 5.0 | 11.5 | 14.5 | 26.0 | 11.5 | 35.5 | 41.5 | 16.0 | 6.5 | 25.1 | 29.3 | 9.0 | 14.0 | 14.5 | 7.0 | 0.5 | 5.5 | 27.0 | 35.0 | 7.5 | 26.0 | 48.5 | 23.5 |
| LLaVA-next-vicuna-7b | 22.2 | 22.2 | 9.2 | 11.0 | 9.1 | 7.7 | 10.5 | 37.0 | 23.2 | 7.0 | 16.5 | 8.0 | 66.0 | 5.0 | 23.5 | 88.0 | 42.5 | 13.0 | 14.5 | 5.5 | 51.0 | 42.5 | 9.5 | 10.0 | 17.1 | 6.5 | 66.0 |
| | | 50.5 | 14.5 | 38.0 | 9.0 | 9.5 | 8.5 | 31.0 | 5.0 | 28.5 | 27.0 | 8.5 | 5.0 | 22.6 | 29.3 | 6.5 | 4.0 | 6.0 | 8.0 | 9.5 | 32.5 | 72.0 | 1.0 | 38.0 | 42.0 | 25.0 | |
| LLaVA-v1.5-7b | 19.2 | 14.1 | 4.2 | 13.7 | 5.8 | 1.9 | 6.9 | 27.3 | 35.0 | 6.5 | 12.5 | 12.5 | 53.0 | 10.0 | 25.5 | 66.5 | 43.0 | 19.0 | 3.5 | 2.5 | 23.5 | 36.5 | 12.0 | 16.5 | 6.7 | 7.0 | 28.0 |
| | | 24.5 | 17.5 | 40.0 | 15.0 | 21.5 | 4.0 | 26.0 | 7.5 | 26.5 | 17.5 | 5.0 | 4.5 | 25.6 | 27.1 | 8.5 | 8.0 | 4.0 | 6.0 | 6.0 | 14.5 | 29.5 | 66.0 | 2.0 | 35.0 | 34.5 | 28.5 |
| sharegpt4v-7b | 18.5 | 16.4 | 5.0 | 10.8 | 6.2 | 9.0 | 2.7 | 34.2 | 28.5 | 4.5 | 10.5 | 3.5 | 57.0 | 4.0 | 12.5 | 55.5 | 44.5 | 13.5 | 5.0 | 5.0 | 26.0 | 38.0 | 14.0 | 15.5 | 10.9 | 6.0 | 25.0 |
| | | 26.5 | 19.0 | 42.0 | 7.5 | 14.0 | 7.5 | 31.5 | 7.0 | 29.0 | 18.0 | 5.0 | 1.5 | 28.1 | 23.3 | 9.5 | 3.0 | 7.0 | 6.0 | 2.0 | 8.0 | 27.5 | 65.5 | 0.0 | 44.0 | 36.5 | 31.0 |
| sharecaptioner | 16.1 | 20.7 | 22.2 | 27.2 | 10.2 | 9.1 | 21.0 | 39.5 | 37.0 | 7.0 | 5.0 | 6.0 | 47.0 | 5.0 | 17.0 | 25.0 | 35.5 | 12.5 | 13.0 | 5.5 | 14.5 | 4.5 | 3.0 | 6.0 | 18.1 | 5.5 | 21.5 |
| | | 17.0 | 22.5 | 18.5 | 12.0 | 14.5 | 11.0 | 23.5 | 7.0 | 25.5 | 22.0 | 5.5 | 2.0 | 16.1 | 43.6 | 9.0 | 2.5 | 1.5 | 5.5 | 8.0 | 26.5 | 47.0 | 2.0 | 28.0 | 16.5 | 9.0 | |
| monkey-chat | 13.7 | 8.4 | 8.0 | 5.9 | 9.2 | 6.7 | 8.1 | 23.5 | 25.3 | 4.5 | 6.0 | 1.5 | 34.5 | 2.0 | 9.0 | 40.5 | 40.5 | 12.0 | 2.5 | 6.5 | 16.5 | 14.5 | 10.0 | 12.5 | 18.1 | 6.5 | 19.5 |
| | | 10.0 | 8.5 | 17.0 | 8.0 | 13.0 | 7.5 | 15.5 | 7.0 | 27.5 | 17.0 | 5.5 | 3.0 | 10.6 | 22.6 | 9.0 | 5.5 | 8.0 | 6.0 | 5.5 | 7.5 | 34.5 | 51.0 | 1.5 | 17.0 | 36.0 | 8.5 |
| **token-concat** | | | | | | | | | | | | | | | | | | | | | | | | | | | |
| GLM-4v-9b | 26.7 | 61.2 | 9.8 | 14.1 | 9.1 | 12.2 | 14.4 | 27.5 | 54.0 | 13.0 | 18.0 | 9.0 | 79.5 | 12.5 | 19.5 | 64.5 | 37.5 | 12.5 | 15.0 | 7.5 | 81.5 | 63.0 | 11.5 | 9.0 | 14.0 | 6.5 | 91.5 |
| | | 55.0 | 22.0 | 41.0 | 5.0 | 4.5 | 3.5 | 53.0 | 13.0 | 39.5 | 41.0 | 16.0 | 6.0 | 76.9 | 29.3 | 8.0 | 4.5 | 11.0 | 5.5 | 1.0 | 7.0 | 34.5 | 42.0 | 3.5 | 25.5 | 49.5 | 21.5 |
| LLaVA-next-vicuna-7b | 5.9 | 0.0 | 0.5 | 0.0 | 4.7 | 0.0 | 0.9 | 1.2 | 11.0 | 0.5 | 0.0 | 0.0 | 0.5 | 0.0 | 2.5 | 62.5 | 45.0 | 0.0 | 0.0 | 0.0 | 0.0 | 0.0 | 0.5 | 0.0 | 14.0 | 0.0 | 0.0 |
| | | 2.0 | 2.5 | 1.0 | 0.0 | 2.5 | 1.0 | 0.0 | 0.5 | 10.5 | 34.5 | 0.0 | 0.0 | 0.0 | 2.5 | 45.0 | 0.0 | 0.0 | 0.0 | 0.0 | 0.0 | 0.0 | 77.0 | 0.0 | 18.5 | 12.5 | 0.0 |
| LLaVA-v1.5-7b | 13.5 | 0.0 | 4.5 | 10.4 | 10.2 | 8.8 | 7.5 | 30.2 | 20.0 | 11.0 | 19.0 | 10.0 | 58.5 | 18.0 | 30.5 | 35.5 | 45.5 | 25.5 | 14.5 | 0.0 | 0.0 | 0.0 | 0.0 | 0.0 | 15.5 | 2.1 | 0.0 |
| | | 2.0 | 8.0 | 4.0 | 2.5 | 27.0 | 8.5 | 0.0 | 7.5 | 7.0 | 26.5 | 6.5 | 0.5 | 0.0 | 29.3 | 8.0 | 17.5 | 5.5 | 0.0 | 11.0 | 2.0 | 0.5 | 69.5 | 0.0 | 36.0 | 26.0 | 12.0 |
| sharegpt4v-7b | 14.0 | 0.2 | 4.5 | 15.9 | 9.2 | 9.2 | 5.1 | 26.8 | 27.3 | 13.5 | 19.5 | 21.5 | 58.0 | 15.0 | 22.5 | 30.5 | 45.0 | 25.5 | 22.5 | 0.0 | 0.0 | 0.0 | 0.0 | 18.0 | 5.7 | 7.5 | 0.0 |
| | | 2.0 | 3.0 | 4.0 | 1.0 | 24.0 | 5.5 | 0.0 | 7.5 | 6.5 | 28.5 | 7.0 | 0.0 | 0.0 | 29.3 | 6.0 | 17.0 | 5.0 | 0.0 | 0.0 | 1.5 | 2.5 | 75.0 | 0.0 | 40.5 | 34.5 | 26.5 |
| sharecaptioner | 9.4 | 0.3 | 8.0 | 13.5 | 16.4 | 11.1 | 21.5 | 23.8 | 33.0 | 0.0 | 0.0 | 1.0 | 35.5 | 0.5 | 1.5 | 4.0 | 3.5 | 10.5 | 1.0 | 0.0 | 0.5 | 2.5 | 0.0 | 5.0 | 0.0 | 8.5 | 0.0 |
| | | 6.0 | 12.0 | 2.5 | 9.0 | 18.0 | 10.5 | 0.0 | 7.0 | 19.5 | 30.0 | 6.0 | 1.5 | 0.0 | 22.6 | 7.5 | 2.5 | 4.0 | 0.0 | 6.5 | 0.0 | 0.0 | 62.0 | 0.0 | 14.0 | 22.5 | 23.5 |
| monkey-chat | 11.1 | 0.0 | 2.5 | 9.4 | 10.6 | 12.4 | 8.1 | 19.8 | 34.2 | 5.5 | 7.5 | 2.5 | 46.5 | 4.5 | 13.5 | 28.0 | 42.0 | 17.0 | 10.5 | 0.0 | 0.0 | 0.0 | 10.5 | 8.5 | 5.2 | 9.0 | 0.0 |
| | | 7.5 | 7.0 | 9.0 | 8.5 | 7.5 | 7.5 | 0.0 | 6.5 | 30.0 | 25.5 | 5.5 | 0.5 | 0.5 | 41.4 | 5.5 | 5.5 | 5.0 | 7.0 | 5.5 | 8.5 | 0.0 | 42.0 | 0.0 | 10.5 | 16.5 | 4.5 |

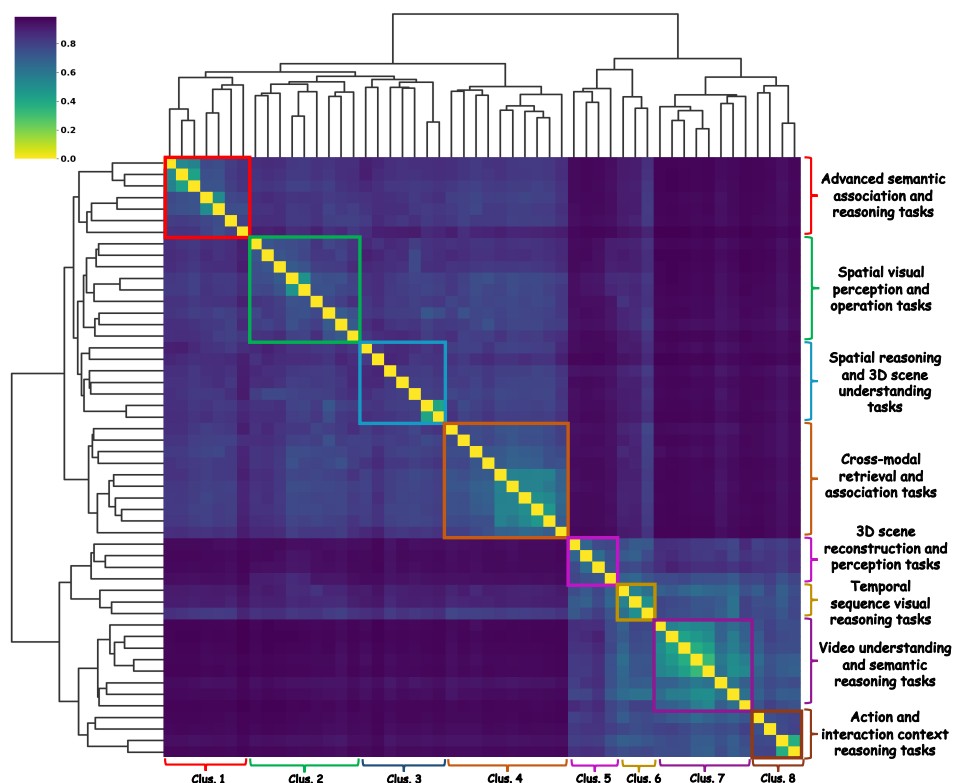

Figure 11: Visualization of task maps and hierarchical clustering along with the task map.

shown in Figure 11, using the task map, we cluster all tasks into 8 categories. The tasks in each category, along with detailed descriptions, are provided in Table 21.

## C.1 CONSTRUCTION

Inspired by the methodology outlined in TaskVec (Ilharco et al., 2023), and benefiting from the extensive and diverse set of tasks in MMIU, we aim to analyze tasks and model performance across different tasks using a task map. Specifically: 1) We extract task vectors similarly to the approach in TaskVec. Using QwenVL-chat as a probing network, we fine-tune it on the multi-choice VQA samples of 52 tasks in MMIU. Formally, a task vector is defined by the weight variation between the

weight fine-tuned on task data $D^t$ and the initial weight $W_0$ of a probing model with the minimum loss, as given by

$$V^t = \arg\min_W \mathcal{L}(W|D^t) - W_0 \tag{1}$$

Notice that for most tasks, we train for 20 epochs, while for a subset of tasks with lower accuracy after initial training, we extend the training to 60 epochs to obtain accurate task vectors. Given the large parameter size of QwenVL-chat, we use LORA (Hu et al., 2021) for fine-tuning, which reduces the length of the task vector from 9.6B to 3.5M and consumes fewer storage resources. 2) For any two task vectors $V^s$ and $V^t$, we compute the cosine distance between their task vectors, where $G^{st} = 1 - \cos(V^s, V^t)$. This process results in a 52x52 task map. 3) As shown in Figure 11, the task map for MMIU reveals that similar tasks cluster together, such as GNAP and GuAR, which are related to GUI, and VO, TO, and VCo, which involve designing image sequence reasoning tasks, which show that the constructed task map aligns with intuition. A comprehensive breakdown of task map can be found in Table 22 and Table 21.

Table 21: Details of task clustering on the task map of our MMT-Bench.

| Cluster ID | Tasks | # Number | Overall Description |
|---|---|---|---|
| 1 | emotion-recognition, forensic-detection, visual-quality-assessment, visually-grounded-reasoning, visual-correspondence, semantic-correspondence, functional-correspondence, | 7 | Advanced semantic association and reasoning tasks: This cluster focuses on semantic understanding and functional correspondence within visual data. Tasks include emotion recognition, semantic and visual correspondence, and visually grounded reasoning, emphasizing deep semantic inference from static images. |
| 2 | spot-the-diff, Multiples-image-captioning, Homography-estimation, single-object-tracking, point-tracking, jigsaw-puzzle-solving, threeD-Pose-Estimation, Image-Captioning-with-Spatial-Context, Image-Spatial-Transformation-Estimation | 9 | Spatial visual perception and operation tasks: Tasks in this cluster involve spatial perception and object tracking, such as multi-image captioning, pose estimation, and point tracking. They require models to interpret spatial changes and structures while interacting with multiple images to achieve operational goals. |

Table 21 – continued from previous page

| Cluster ID | Tasks | # Number | Overall Description |
|---|---|---|---|
| 3 | next-img-prediction, spot-the-similarity, ravens-progressive-matrices, threed-indoor-recognition, threed-cad-recognition, Multiview-reasoning, threeD-Depth-Estimation | 7 | High-level spatial reasoning and 3D scene understanding tasks: This cluster emphasizes 3D scene reasoning and multi-view analysis, including depth estimation and object recognition. Tasks focus on deriving object properties and spatial relationships from 3D structures or multiple perspectives, showcasing advanced spatial reasoning. |
| 4 | temporal-localization, visual-cloze, person-reid, text2image-retrieval, vehicle-retrieval, face-retrieval, sketch2image-retrieval, image2image-retrieval, handwritten-retrieval, Icon-Question-Answering-with-Spatial-Context | 10 | Cross-modal retrieval and association tasks: This cluster addresses cross-modal associations between vision and text, covering tasks like image-to-text retrieval, handwritten retrieval, and icon-based question answering. The focus is on efficiently mapping semantic and visual features for precise retrieval and understanding. |
| 5 | mevis, threeD-Scene-Reconstruction, threeD-Object-Detection, threeD-Object-Tracking | 4 | 3D scene reconstruction and visual perception tasks: Tasks in this cluster specialize in 3D scene perception and reconstruction, such as object detection, tracking, and scene reconstruction. These tasks demand precise spatial understanding and the ability to process 3D geometric structures. |
| 6 | visual-coherence, visual-ordering, temporal-ordering | 3 | Temporal sequence visual reasoning tasks: This cluster involves temporal reasoning in visual data, including visual coherence, ordering, and temporal sequence analysis. Tasks challenge models to infer temporal relationships between visual elements while maintaining logical consistency in sequential data. |

Table 21 – continued from previous page

| Cluster ID | Tasks | # Number | Overall Description |
|---|---|---|---|
| 7 | meme-vedio-understanding, action-quality-assessment, video-captioning, general-action-recognition, textual-cloze, casuality-reasoning, Image-text-retrieval-with-Spatial-Context, Egocentric-Video-QuestionAnswering | 8 | Multimodal video understanding and semantic reasoning tasks: This cluster focuses on multimodal semantic inference and temporal video understanding, including video captioning, action quality assessment, and cross-modal semantic retrieval. Tasks require models to interpret dynamic scenes and synthesize temporal and semantic information. |
| 8 | gui-next-action-prediction, gui-app-recognition, Multiview-Action-Recognition, threeD-question-answering | 4 | Action and interaction context reasoning tasks: This cluster emphasizes contextual reasoning in user interaction, such as GUI operation, action recognition, and 3D question answering. Tasks demand models to comprehend and predict dynamic actions within interactive scenarios. |

## C.2 ANALYSIS

We perform hierarchical clustering on the task map and analyze each cluster. Unlike the previous method of classification through multiple relationships, this approach leverages Qwen-VL as a probe network, allowing for a more objective segmentation based on the intrinsic attributes of the tasks themselves. Combining the model's performance in each cluster, as shown in Figure 4, with the task map presented in Figure 4, we begin by analyzing the in-domain tasks where the model demonstrates strong performance. Next, we examine the out-domain tasks where the model underperforms. Finally, we propose using Taskmap to assess task difficulty, guiding future model development.

**In-Domain Tasks Analysis.** In-domain tasks are tasks that most current multimodal large models can handle correctly. For multi-image tasks, the model generally struggles to achieve satisfactory results, with most models performing worse than random selection. Consequently, the model can only achieve good performance on a limited number of tasks. Specifically, as shown in Table 22, for tasks in clusters 7, 8, and some tasks in cluster 2, which involve recognition or captioning (e.g., video captioning, action recognition), the model per-

Table 22: The number of tasks within each cluster after hierarchical clustering, and the Spearman correlation $r$ between the average performance of the model on these tasks and the overall performance of the model.

| # Cluster | 1 | 2 | 3 | 4 | 5 | 6 | 7 | 8 |
|---|---|---|---|---|---|---|---|---|
| # Tasks | 7 | 9 | 7 | 10 | 4 | 3 | 8 | 4 |
| # $r$ | 0.94 | 0.85 | 0.92 | 0.96 | 0.83 | 0.93 | 0.74 | 0.73 |
| # Acc | 27.9 | 34.7 | 26.3 | 26.8 | 17.2 | 20.2 | 32.3 | 33.7 |

forms relatively well. We believe this is because these multi-image tasks focus on overall image perception, requiring less comparison and reasoning between images. Additionally, the model has already demonstrated strong capabilities in high-level perception tasks involving single images.

**Out-of-Domain Tasks Analysis.** Out-of-Domain Tasks refer to tasks where most models perform poorly. Specifically, as shown in Table 22, we find that models struggle with tasks in clusters 4, 5, and 6. Upon analysis, we discover that tasks in clusters 4 and 6 involve modeling semantic relationships or sequential order among multiple images, which requires strong memory capabilities and advanced perceptual and reasoning skills. Most open-source models underperform on these tasks, especially in image sequencing problems (e.g., temporal ordering tasks), where even closed-

source models struggle to achieve satisfactory results. Tasks in cluster 5 pertain to visual tasks involving 3D spatial relationships, such as detection and tracking. Although models show some proficiency in handling 2D visual tasks, they lack optimization for 3D data, making them capable of handling only simple 3D tasks that involve basic semantic understanding, but inadequate for accurately modeling complex 3D visual tasks.

**Analysis of Different Model Categories.** As shown in Figure 4, we categorize the models into four types. It is evident that across different task categories, the performance trend follows: closed-source models outperforming models with extensive multi-image SFT, which in turn outperform multi-image input models, and finally single-image input models. This indicates: **1) Extensive multi-image SFT is crucial for multi-image performance.** A more detailed analysis of this is provided in Appendix B.3. **2) While some single-image models (e.g., GLM-4V) can generalize to multi-image tasks to a certain extent, their average performance remains inferior to models that support multi-image input.** This suggests that most single-image models struggle to generalize to multi-image tasks and require instruction fine-tuning with multi-image data.

## D    COMPUTAIONAL RESOURCES

Table 23: Resource consumption of some models evaluated on the testmini set of MMIU.

| Model | Batch Size | Resources | Times | Memory Utilization Per GPU |
|---|---|---|---|---|
| GPT-4o | 4 | USD 8.57 | 0.51h | - |
| LLaVA-Next-Interleave-7B | 3 | 1 x A100-80GB | 0.04h | 60381MiB |
| Mantis-idefics2-8B | 6 | 1 x A100-80GB | 0.15h | 73896MiB |
| xGen-MM-4B | 7 | 1 x A100-80GB | 0.14h | 75082MiB |
| InternVL-Chat-V1.5-26B | 8 | 1 x A100-80GB | 0.89h | 71619MiB |
| InternVL2-8B | 1 | 1 x A100-80GB | 0.97h | 79292MiB |
| LLaVA-1.5-7B | 4 | 1 x A100-80GB | 0.02h | 64462MiB |

As shown in Table 23, our benchmark enables efficient evaluation across all tested models, regardless of their size or complexity. The results show that most models can be tested on a single A100 card within 1 hour; even the closed-source model GPT-4o can be tested within 1 hour with only 8.57 USD. Additionally, our framework's support for multi-GPU distributed inference ensures that even resource-intensive models achieve high efficiency. This makes our benchmark suitable for comprehensive multi-model comparisons while maintaining practical evaluation times across diverse architectures.

## Low-Level Semantics: VQA

Question: Compared to the second image, how is the lighting situation in the first image?
Options:
A)  Similar
B)  Slightly better
C)  Slightly worse

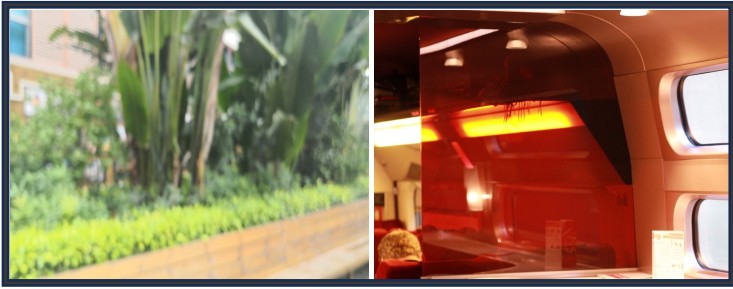

**GPT-4o**: C. The first image is blurry with poor sharpness, likely due to suboptimal lighting or motion issues. In contrast, the second image has well-distributed artificial lighting, providing clear visibility and detail, with distinct objects, colors, and reflections…

**Claude3.5**: C. The first image is noticeably blurry and lacks sharp focus, which suggests poor lighting conditions or camera shake due to low light.. In contrast, the second image has much clearer definition and vivid colors, particularly the bright reds and yellows of the train interior…

**InternVL2**: C. In the first image (Image-1), The primary light source seems natural, possibly from an overcast sky or a shaded area, which results in less contrast and a generally darker atmosphere. On the other hand, Image-2 shows artificial lighting with bright overhead lights that cast direct illumination on various elements within the scene…

Figure 12: Error cases in low-level semantic image relations (visual quality assessment). Red highlights the wrong answer.

**High-Level(obj) Semantics: S2IR**

Question: Please retrieve the most similar image to the Query Image in the candidate images. The query image is the first image..
Options:
A)  the second image
B)  the third image
C)  The forth image

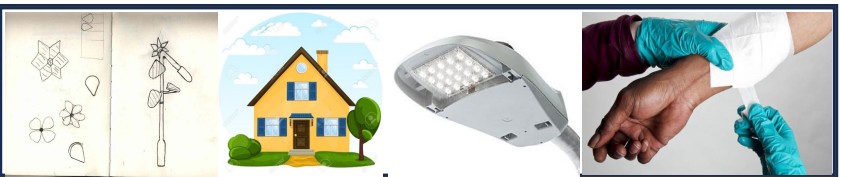

**GPT-4o**: **A.** The query image contains sketch-like drawings in a notebook, primarily of flowers and petals. the second image, which is an illustration of a house with a somewhat simple, illustrated style, is relatively closer in terms of being an illustrative drawing. The other images (a streetlight and a medical scenario) do not share this illustrative nature.

**Claude3.5**: **C.** The query image (first image) shows hand-drawn sketches of various objects, including flowers and what appears to be a pinwheel or windmill-like shape. Among the candidate images, the fourth image is the most similar because: It also depicts hands, which aligns with the hand-drawn nature of the query sketch…

**InternVL2**: **C.** The first image, which appears to be a collection of simple line drawings including flowers and leaves, has an artistic or illustrative style. Among the given options, Image-4 also shows elements that can be considered as illustrations or diagrams related to human anatomy (showing a hand being bandaged). This visual representation aligns more closely with the artistic nature of the first image compared to the other two images, making it the most similar in terms of its illustrative quality.…

Figure 13: An error case in high-level(obj) semantic image relations (sketch2image retrieval). Red highlights the wrong answer.

## High-Level(sub) Semantics: ER

Question: Now you are given four images, please examine the details and tell which one of them has emotions of the characters that are different from the others. Which image is most likely to show a different emotion from the other images?
Options:
A)    the first image                C) the third image
B)    the second image              D) the forth image

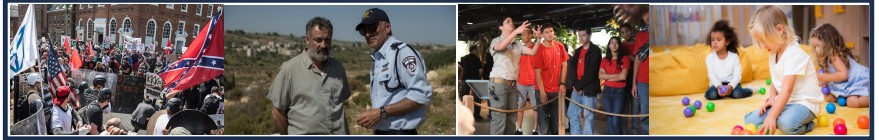

**GPT-4o**: C. The first, second, and fourth images seem to portray moments with more intense or serious emotions, likely involving protests, official interactions, or children deeply engaged. The third image, however, shows students in what appears to be an educational setting, listening attentively and interacting calmly. This setting and the visible emotions differ from the more charged or formal emotions in the other images....

**Claude3.5**: D. The fourth image depicts young children playing with colorful balls in what appears to be a daycare or preschool setting. The children seem engaged in calm, focused play activities. Their body language and interactions suggest curiosity, contentment, and peaceful exploration as they handle the toys. In contrast, the other images show more tense or serious situations:...

**InternVL2**: D. **Image (A):** This image depicts a scene of conflict or protest. The individuals involved appear to be engaged in intense confrontation, with visible signs of tension and aggression. **Image (B):** The second image shows two men standing outdoors, seemingly engaged in conversation or perhaps observing something together. **Image (C):** The third image presents a group of people being addressed or guided through what appears to be a zoo or museum exhibit. **Image (D):** The primary emotion captured here is that of innocence, playfulness, and happiness. Image (D) stands out because it captures emotions related to childhood playfulness—joy, curiosity, and comfort—which starkly contrasts with the more serious, tense, or investigative moods evident in the other images.

Figure 14: An error case in high-level(sub) semantic image relations (emotion recognition). Red highlights the wrong answer.

## Discrete Temporal: GNAP

Question: You are given a GUI navigation task that includes current screenshot images, historical screenshot images, and corresponding actions. The task is: Open the Firefox Browser to search for the best video blogs on travel vlogs. Then, go to the Setting app to turn up the brightness on your phone. \nThe historical actions are… The last image represents the current screenshot and the preceding images are historical screenshots. Which action should I do next?
Options:
A)    SCROLL: LEFT        C) SCROLL: DOWN
B)    SCROLL: UP.          D) CLICK: (31, 960)

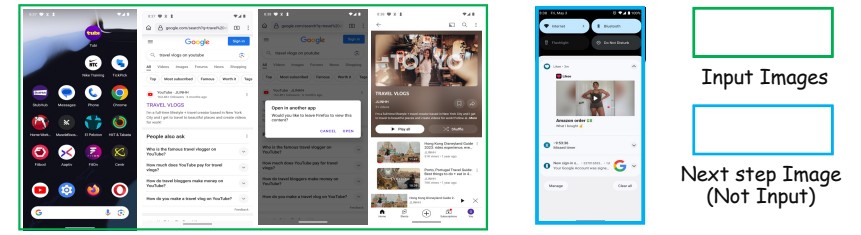

Input Images

Next step Image
(Not Input)

**GPT-4o**: D The next action should be to adjust the brightness on your phone by going to the Settings app, as per the task instructions. After the previously listed actions, which included opening the Firefox Browser and searching for travel vlogs, the current screenshot shows the mobile home screen, with apps including the Settings app visible. Considering the task involves navigating the GUI and focusing on brightness settings, the most logical next step is to click on the Settings app icon. the selection associated with clicking the Settings app is: CLICK: (31, 960).

**Claude3.5**: D Explanation:1. The current screen shows a YouTube channel page for travel vlogs, which indicates we've already completed the first part of the task (searching for travel vlogs)… 5. This action will allow you to navigate away from the YouTube app and move towards completing the next part of the task, which is accessing the Settings to adjust brightness.Therefore, clicking the Home button (option D) is the most appropriate next action.

**InternVL2**: D. Coordinates (31, 960) aligns with typical placement for a system navigation button or perhaps a menu/options area that could provide access to other applications like Firefox. Therefore, based on common smartphone interface design principles and the location indicated by option D, clicking at coordinates (31, 960) would be the logical next step to proceed towards opening the Firefox browser…

Figure 15: An error case in discrete temporal image relations (GUI APP action prediction). Red highlights the wrong answer.

**Continuous Temporal: TO**

Question: Please predict the order of the following pictures, and give each picture a sequential index. This index starts from 0. The larger the index, the later the order.
Options:
A) 2,0,1,3
B) 1,0,3,2
C) 3,1,0,2
D) 0,2,1,3

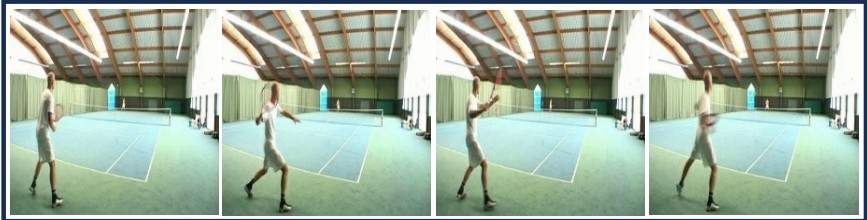

**GPT-4o**: *C*: Following this sequence, The correct choice is :C: [3, 1, 0, 2]
Explanation:
- Picture 3 shows the player in the initial preparation.
- Picture 1 shows the player starting to swing.
- Picture 0 shows the player in mid-swing.
- Picture 2 shows the player finishing the swing.

**Claude3.5**: B: Image 1 (index 0): The player is in the initial stance, preparing to serve. Image 0 (index 1): The player has tossed the ball up and is reaching back with the racket. Image 3 (index 2): The player is at the peak of the serve motion, with the racket raised high.Image 2 (index 3): The player has completed the serve, with the racket following through.

**InternVL2**: *C*: Image-3 (Index 0): The player begins initiating the backhand swing. Image-1 (Index 1):As the swing progresses, he transitions into the follow-through preparation phase. Image-2 (Index 2): Following the initial swing setup, he completes the backhand stroke with full extension...

Figure 16: An error case in continuous temporal image relations (temporal ordering). Red highlights the wrong answer.

## 2D Spatial: JPS

Question: The patches in the middle of the image might be disordered. Please state the correct order of the number indexes based on the given patches, following the sequence: top left, top right, bottom left, bottom right.
Options:
A) 1, 2, 3, 4
B) 2, 4, 1, 3
C) 4, 1, 3, 2
D) 2, 1, 4, 3

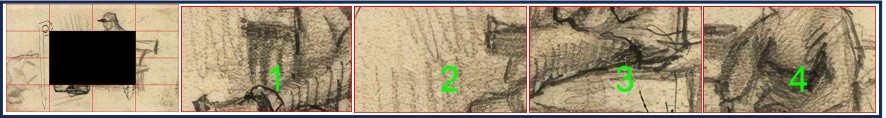

**GPT-4o**: D Patch 2 shows some vertical lines that seem to match the top center lines in the larger image. This makes it appropriate for the top left position. Patch 4 shows a portion of the head and a hat, matching the top right corner of the larger image. Patch 1 shows less identifiable features but the patterns and shading might fit the bottom left corner of the larger image. Patch 3 shows straight horizontal and vertical lines that fit well into the bottom right corner of the larger image, matching the sketch patterns there....

**Claude3.5**: D Top left: The patch labeled '2' shows faint sketch lines, which matches the top left corner of the main image. Top right: The patch labeled '1' contains a sketched figure wearing a hat, corresponding to the top right corner of the main image. Bottom left: The patch labeled '4' shows a more heavily shaded area with curved lines, which aligns with the bottom left corner of the main image.
Bottom right: The patch labeled '3' depicts what appears to be a sketched body or torso, fitting the bottom right corner of the main image....

**InternVL2**: D Top Left (Patch 1):This patch shows a part of what appears to be an animal or reptile's body with scales. The orientation and details suggest it should be placed at the top right. Top Right (Patch 2)**: This patch also contains similar scaled textures but seems to fit logically as a continuation from the previous piece, indicating that it belongs at the top left.Bottom Left (Patch 3)**: This patch has detailed lines and features consistent with the texture seen in the other pieces, suggesting continuity from the top sections. It fits best in the bottom right. Bottom Right (Patch 4): This patch includes more defined edges and structures which align well when placed after the third piece, fitting correctly in the bottom left position....

Figure 17: An error case in 2D spatial image relations (jigsaw puzzle solving). Red highlights the wrong answer.

## 3D Spatial: 3DIR

Question: What is the category of the point cloud based on the multi-view of the point cloud?
Options:
A) Cabinet
B) Bin
C) Bed
D) sink

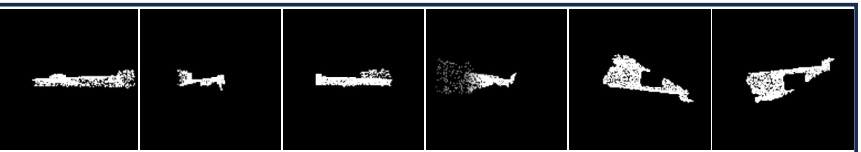

**GPT-4o**: C: The multi-view images of the point cloud show a structure that looks fairly flat and elongated with some irregularities on one side. This shape is most consistent with the features of a "bed" (Option C).
A cabinet (Option A) would typically have a more rectangular block-like appearance from multiple views.
A bin (Option B) would likely appear more box-shaped or cylindrical and less elongated horizontally…

**Claude3.5**: B: Based on the multiple views of the point cloud provided, we can deduce that the object represented is most likely a bin. Here's why: Shape: The point cloud shows an object with a generally elongated, rectangular shape in most views, which is consistent with the typical form of a bin or container. Open top: Several views, particularly the top-down perspectives, show an open top area…

**InternVL2**: A: The point cloud in the images displays several characteristics that suggest it represents a cabinet: **Structure**: The overall shape appears to be boxy with distinct edges and corners, which is typical for a cabinet. Cabinets often have rectangular or square shapes with flat surfaces for doors and compartments.**Proportions**: The proportions shown in the multiple views indicate height and depth consistent with common cabinet sizes. Cabinets are designed to fit against walls and provide storage space…

Figure 18: An error case in 3D spatial image relations (3D indoor recognition). Red highlights the wrong answer.

