# OpenReview forum: "MMIU: Multimodal Multi-image Understanding for Evaluating Large Vision-Language Models"
_ICLR.cc/2025/Conference — ICLR 2025 Poster_

### Official Review · Reviewer_g41s · 2024-10-26

**Soundness:** 3
**Presentation:** 3
**Contribution:** 3
**Rating:** 6
**Confidence:** 3

**Summary:**

The article introduces the MMIU benchmark for evaluating large vision-language models on multi-image tasks. It includes various tasks and a large dataset with over 77K images and 11K questions. The study finds that even advanced models struggle with complex multi-image understanding

**Strengths:**

1、Comprehensive Evaluation: The MMIU benchmark provides a thorough assessment of LVLMs by covering a wide range of multi-image tasks and relationships, making it one of the most extensive benchmarks in this domain.

2、Diverse Task Coverage: It includes a variety of tasks that require different cognitive processes, from low-level visual perception to high-level reasoning, offering a holistic view of a model's capabilities.

3、Insightful Analysis: The article presents detailed analytical experiments that identify key performance gaps and limitations in current models, providing valuable insights for future research and development in the field of multimodal AI.

**Weaknesses:**

1、Typo: 173 line multimodal large language models (LVLMs), abbreviation error.

2、The details of writing can be further polished, such as using Fig. in some places, Some places use Figure

**Questions:**

1、Are there any duplicates between the samples in your dataset and the current multi graph dataset?

2、Why do models without multi-imgae SFT perform better than those with multi-imgae SFT on your benchmark?

---

> ### Author Response · Authors · 2024-11-21
> **Response to Reviewer g41s (Q1)**
>
> **In order to observe the changes in the PDF more clearly, we have marked the changes in the new PDF with purple marks based on your comments.**
>
> Q1: some writing issues
>
> A1: Thank you for your correction! We have made the necessary changes and thoroughly reviewed the entire text again. Let us know if there is anything else we can refine or clarify further.

---

> ### Author Response · Authors · 2024-11-21
> **Response to Reviewer g41s (Q2)**
>
> Q2: Are there any duplicates between the samples in your dataset and the current multi graph dataset?
>
> A2: Compared to other multi-image benchmarks such as **MuirBench** [1] and **MilesBench** [2] , MMIU features test data that is highly distinct, establishing itself as an independently developed benchmark.
>
> To validate this, we calculate the visual similarity between the images in MMIU and those in MuirBench and MilesBench. Specifically, we use **LPIPS** [3] as the similarity metric. Many image editing works adopt LPIPS as a standard measure, and based on reported results, an LPIPS value below **0.1** is generally considered indicative of relatively similar images [4,5].
>
> As shown in **Table 11 below**, when using a threshold of **0.1**, the image duplication rates between MMIU and MuirBench or MilesBench are only **0.1%** and **1.6%**, respectively. This demonstrates that MMIU is a highly independent and novel evaluation benchmark.
>
> Let me know if this aligns with your expectations!
>
> | **LPIPS ↓**    | 0.1  | 0.2  | 0.3  | 0.4   |
> | -------------- | ---- | ---- | ---- | ----- |
> | **MuirBench**  | 0.1% | 1.8% | 6.4% | 11.8% |
> | **MilesBench** | 1.6% | 3.3% | 7.8% | 12.9% |
>
> ​																Table11: LPIPS between MMIU and MuirBench, MilesBench
>
>
>
> [1] MuirBench: A Comprehensive Benchmark for Robust Multi-image Understanding
>
> [2] MileBench: Benchmarking MLLMs in Long Context
>
> [3] The Unreasonable Effectiveness of Deep Features as a Perceptual Metric
>
> [4] Diffree: Text-guided shape free object inpainting with diffusion model
>
> [5] Inversion-Free Image Editing with Language-Guided Diffusion Models

---

> ### Author Response · Authors · 2024-11-25
> **Looking forward to your reply !**
>
> We would like to extend our heartfelt gratitude for the invaluable suggestions provided for this paper. In light of your feedback, we have diligently provided comprehensive and elaborate explanations. If you have any further inquiries, we are more than delighted to offer additional clarifications and conduct further experiments. We eagerly anticipate your response!

---

> ### Author Response · Authors · 2024-11-26
> **Response to Reviewer g41s (Q3)**
>
> Q3: Why do models without multi-imgae SFT...
>
> A3: Thanks to the reviewer for his suggestion. We miss this question before and now we give a detailed answer. For the original paper, InternVL2-Pro (only used video data for "multi-image SFT") and InternVL-Chat-1.5 surpassed some multi-image SFT models, but since the sizes of these two models (**78B and 26B**) are large, we add more models for comparison. Specifically, Below we comprehensively discuss the effect of multi-image data at different training stages. These contents are also involved in [Response to Reviewer bVMZ (Q5)](https://openreview.net/forum?id=WsgEWL8i0K&noteId=ZiG52vKojd) and we provide a more detailed analysis in **Appendix B.3: Discussion on the Effectiveness of Multi-Image Training**.
>
> We consider the two most common training stages: **pretraining** and **SFT (supervised fine-tuning)**. Based on the inclusion of multi-image data at these stages, we categorized models into four groups.  As shown in **Table 12 below**, we find the following:
>
> 1. **Models exhibit some generalization to multi-image tasks without multi-image data**:
>    Models trained without multi-image data, such as **Mini-InternVL1.5-chat**, still achieve reasonable performance on multi-image tasks, and even surpass the model trained with multi-image SFT: XGen-MM-Interleaved, which we attribute to their strong visual encoders leading to **excellent image understanding capabilities**. (e.g. Mini-InternVL1.5-4B even outperforms LLaVA-Next-34B on the multimodal comprehensive evaluation benchmark **MME**). However, even so, their performance is still lower than that of models trained with multi-image SFT such as Mantis, highlighting the importance of multi-image SFT
>
> 2. **Multi-image SFT consistently improves performance**:
>    Regardless of whether multi-image pretraining is conducted, models like **LLaVA-Next-Interleave** and **XGen-MM-Phi3-Interleaved** show significant improvements when multi-image SFT is applied.
>
> 3. **Combining multi-image SFT with pretraining yields better results**:
>    Models that incorporate multi-image data at both stages, such as **Mantis-idesfics2**, outperform those relying solely on SFT, such as **Mantis-SigCLIP**.
>
> 4. **Multi-image pretraining alone is insufficient**:
>    Models that only perform multi-image pretraining, like **XGen-MM-Phi3-Single** and **idefics2**, underperform compared to those that only apply multi-image SFT (e.g. **Mantis-SigCLIP**), highlighting the importance of multi-image SFT.
>
>
>
>
> If you have further questions, please do not hesitate to let me know immediately. We sincerely hope that you will consider raising your rating and we will do our best to answer your questions.
>
> | **Model**                    | **Multi-image Pretrained** | **Multi-image SFT** | **Size** | **Semantic** | **Temporal** | **Spatial** | **Overall** |
> | ---------------------------- | -------------------------- | ------------------- | -------- | ------------ | ------------ | ----------- | ----------- |
> | Mini-InternVL1.5-chat        | No                         | No                  | 2B       | 30.4         | 36.2         | 26.5        | 30.5        |
> | Mini-InternVL1.5-chat        | No                         | No                  | 4B       | 32.2         | 36.8         | 28.5        | 32.1        |
> | InternVL1.5-chat             | No                         | No                  | 26B      | 33.4         | 45.2         | 35.5        | 37.4        |
> | LLaVA-Next                   | No                         | No                  | 7B       | 23.0         | 24.1         | 20.3        | 22.2        |
> | LLaVA-Next-Interleave        | No                         | Yes                 | 7B       | 35.7         | 32.2         | 31.1        | 32.4        |
> | Mantis-SigCLIP               | No                         | Yes                 | 8B       | 53.6         | 38.4         | 35.7        | 42.6        |
> | xGen-MM-Phi3-Single [1]      | Yes                        | No                  | 4B       | 25.1         | 25.7         | 23.1        | 24.5        |
> | idefics2                     | Yes                        | No                  | 8B       | 31.6         | 20.9         | 29.2        | 27.8        |
> | InternVL2                    | Yes                        | Only video          | 8B       | 35.2         | 39.5         | 31.2        | 34.8        |
> | Mantis-idsfics2              | Yes                        | Yes                 | 8B       | 56.3         | 40.8         | 39.2        | 45.6        |
> | XGen-MM-Phi3-Interleaved [1] | Yes                        | Yes                 | 4B       | 28.2         | 28.7         | 29.1        | 28.7        |
>
> Table12: Performance comparison of various models with multi-image training data in different stage on MMIU.
>
>
> [1] xGen-MM (BLIP-3): A Family of Open Large Multimodal Models

---

> ### Author Response · Authors · 2024-11-30
> **Looking forward to your reply !**
>
> As the discussion period is nearing its end, we greatly value your feedback and are more than happy to provide further clarifications if needed. If you have any additional questions, please feel free to reach out immediately. We sincerely hope you will consider revising your rating, and we will do our utmost to address any concerns you may have.

---

### Official Review · Reviewer_JjzH · 2024-10-27

**Soundness:** 3
**Presentation:** 3
**Contribution:** 2
**Rating:** 6
**Confidence:** 3

**Summary:**

The paper introduces the Multimodal Multi-Image Understanding (MMIU) benchmark, which evaluates large vision-language models (LVLMs) on complex multi-image tasks, including spatial, semantic, and temporal relationships. Covering 52 task types with 77,000 images and 11,000 curated questions, MMIU is a comprehensive benchmark that reveals significant challenges, particularly in spatial and temporal tasks, for leading models such as GPT-4o.

**Strengths:**

1. Multi-image understanding is indeed a meaningful and impactful problem within multimodal research, this paper effectively addresses this need by establishing MMIU, a benchmark specifically designed to evaluate large vision-language models (LVLMs) across diverse multi-image tasks.

2. The presentation is clear and well-structured, figures and tables are effectively used to enhance understanding.

**Weaknesses:**

1. The benchmark evaluates numerous LVLMs but could benefit from deeper comparisons across models with distinct visual encoding strategies. This would provide more specific insights into how different architectures impact performance on complex spatial and temporal tasks, which is particularly valuable for the multi-image context.

2. While MMIU successfully identifies gaps in LVLM performance, the paper does not sufficiently explore or propose alternative methods to address these gaps, such as novel training techniques or architectural adjustments. This limits the benchmark’s practical utility for guiding future model development.

**Questions:**

This paper is overall good, it would be better to present deeper analysis on model performance variations across specific multi-image relationships, particularly spatial, temporal, and high-level semantic tasks. Currently, the analysis highlights general performance trends but could benefit from a more granular examination of how and why certain models excel or struggle with specific relationship types.

---

> ### Author Response · Authors · 2024-11-21
> **Response to Reviewer JjzH (Q1)**
>
> **In order to observe the changes in the PDF more clearly, we have marked the changes in the new PDF with orange marks based on your comments.**
>
> Q1: ... more specific insights into how different architectures impact ...
>
> A1: We sincerely thank the reviewer for the insightful comments. Indeed, architecture plays a significant role in influencing model performance. However, given the differences in training datasets for the tested models, it is challenging to conduct detailed controlled experiments. To address this, we select some of the best-performing models and analyze commonalities in their architectures that may contribute to better performance on multi-image tasks.
>
> **1) High resolution and dynamic resolution are crucial.**
> Research suggests that increasing resolution and incorporating dynamic resolution during training significantly improve image understanding performance (e.g., SPHINX [1]). This is because low resolutions tend to miss critical visual details, and training on a single resolution size can lead to image distortions or inefficiencies in training and inference. For multi-image tasks, we observe that top-performing models (e.g., **InternVL2-8B, Mantis-idefics2, LLaVA-Next-Interleave**) adopt high resolutions (448, 384, and 336, respectively) alongside dynamic resolution training strategies. This enables the models to better comprehend visual inputs.
>
> **2) A strong visual encoder makes a difference.**
> The **InternVL** series demonstrates outstanding performance on MMIU. For instance, both **Mini-InternVL-Chat-1.5-4B** and **XGen-MM-Interleave-4B** [2] use **Phi3-mini** [3] as their LLM, yet InternVL consistently performs better. Remarkably, **Mini-InternVL-Chat-1.5-4B** achieves performance on par with the larger **LLaVA-Next-Interleave-7B**. We attribute this partly to **InternViT**, the encoder used in InternVL models. InternViT is a powerful vision encoder that outperforms CLIP-based alternatives across multiple benchmarks. Thus, having a robust vision encoder is instrumental in enhancing performance on multi-image tasks.
>
> We provide a more detailed analysis in **Appendix B.3**.
>
> Thank you again for your constructive comments, which have helped us refine and deepen our analysis.
>
>
>
>
>
> [1] SPHINX: The Joint Mixing of Weights, Tasks, and Visual Embeddings for Multi-modal Large Language Models
>
> [2] xGen-MM (BLIP-3): A Family of Open Large Multimodal Models
>
> [3] Phi-3 Technical Report: A Highly Capable Language Model Locally on Your Phone

---

> ### Author Response · Authors · 2024-11-21
> **Response to Reviewer JjzH (Q2)**
>
> Q2: While MMIU successfully identifies gaps...
>
> A2: We sincerely thank the reviewer for the valuable suggestions. Indeed, MMIU does not involve training a specific model or proposing a new method to solve multi-image understanding tasks. However, training a model is not the only way to contribute. MMIU provides a significant contribution by testing a wide range of models on comprehensive multi-image tasks, conducting detailed analyses, and summarizing potential approaches to improve multi-image performance:
>
> 1. **Sufficient multi-image SFT greatly enhances performance:**
>    For example, **Mantis-idesfics2**, after performing multi-image SFT on the base model **idefics2**, achieves a remarkable **17.8% accuracy improvement**. Even without multi-image pretraining, models like **Mantis-SigCLIP** still achieve a relatively high accuracy of **42.6%**, showing the effectiveness of multi-image SFT.
> 2. **Combining multi-image SFT with multi-image pretraining yields better results:**
>    For instance, **Mantis-idesfics2** outperforms **Mantis-SigCLIP** because the base model of the former underwent interleaved image-text pretraining, resulting in higher accuracy.
> 3. **Multi-image pretraining alone may not be cost-effective:**
>    Models like **idefics2** and **xGen-MM-Single**, despite undergoing multi-image pretraining, show suboptimal performance without sufficient multi-image SFT. This highlights the importance of coupling pretraining with fine-tuning for optimal results.
> 4. **High resolution and dynamic resolution matter:**
>    Please refer to our response to **[Response to Reviewer JjzH (Q1)](https://openreview.net/forum?id=WsgEWL8i0K&noteId=GypD7Nx1i2)** for further details.
> 5. **A strong visual encoder is crucial:**
>    Please refer to our response to **[Response to Reviewer JjzH (Q1)](https://openreview.net/forum?id=WsgEWL8i0K&noteId=GypD7Nx1i2)** for further discussion.
>
> We provide a more detailed explanation of these findings in **Appendix B.3**. Once again, we thank the reviewer for their insightful comments. We fully agree that proposing and validating solutions is also an important contribution, and we will prioritize this as part of our future work. Thank you for helping us improve the scope and depth of our analysis!

---

> ### Author Response · Authors · 2024-11-21
> **Response to Reviewer JjzH (Q3)**
>
> Q3: ... more granular examination of how and why certain models excel or struggle with specific relationship types
>
> A3: Thank you for your question. We conduct a statistical error case analysis on four representative models across different multi-image relationships, and we summarize the findings as follows:
>
> **1）Semantic tasks are more prone to perception errors** compared to temporal and spatial tasks. This is attributed to the greater diversity inherent in semantic tasks, such as image retrieval, which poses additional challenges for multi-image understanding. In contrast, temporal and spatial tasks typically involve images with inherent consistency, like changes in time or viewpoint, making them less susceptible to perception errors.
>
> **2）Reasoning errors are more common in temporal and spatial tasks.** Temporal tasks often require models to infer object dynamics over time (e.g., ordering tasks or next image prediction), while spatial tasks demand reasoning about spatial relationships and cognition (e.g., Image-Captioning-with-Spatial-Context). Current models consistently struggle with these complex reasoning requirements.
>
> **3）Spatial tasks are particularly affected by capacity-related errors.** Tasks requiring advanced visual understanding, such as 3D pose estimation, exhibit a significant number of errors caused by the model’s limited capacity. This is likely due to insufficient training for specialized visual challenges, resulting in irrelevant or incoherent responses.
>
> **4）Open-source models are more prone to perception errors than closed-source models.** This disparity is largely due to the capabilities of the visual encoder, which is critical for multi-image understanding. Enhancing the performance of visual encoders in open-source models could effectively reduce these perception errors.
>
> **5）Smaller open-source models often fail to follow instructions,** particularly in long-context multi-image tasks. Their limited language model size impairs their ability to process and understand complex multi-image inputs. This highlights the importance of improving LLM capacity, especially for tasks requiring long-context understanding, to achieve better performance in multi-image applications.
>
> We visualize the error types for different models under various image relationships in **Figure 9** and provide a detailed analysis in **Appendix B.4**.

---

> ### Author Response · Authors · 2024-11-25
> **Looking forward to your reply !**
>
> Thank you very much for your insightful suggestions on this paper. We have responded to each of these in great detail. And we kindly request that you consider updating the score accordingly. If you have any other questions, we are more than happy to provide additional clarification as well as experiments, and look forward to your reply !

---

> ### Author Response · Authors · 2024-11-30
> **Looking forward to your reply !**
>
> As the discussion period is nearing its end, we truly appreciate your feedback and are eager to provide any additional clarifications if necessary. Should you have any further questions, please don't hesitate to contact me right away. We sincerely hope you will reconsider your rating, and we will make every effort to address your concerns.

---

### Official Review · Reviewer_4Ca4 · 2024-10-28

**Soundness:** 3
**Presentation:** 3
**Contribution:** 3
**Rating:** 6
**Confidence:** 4

**Summary:**

This paper presents MMIU, a comprehensive benchmark designed to evaluate large vision-language models (LVLMs) on multi-image tasks, addressing limitations in existing single-image-focused evaluations. MMIU includes 52 tasks spanning seven image relationship types, such as semantic, spatial, and temporal, with a total of 77,659 images and 11,698 multi-choice questions.

**Strengths:**

- Originality: The paper introduces a novel benchmark, MMIU, specifically designed for evaluating large vision-language models (LVLMs) on multi-image tasks. This fills a critical gap, as existing benchmarks predominantly focus on single-image understanding. The inclusion of diverse image relationships—semantic, spatial, and temporal—is a creative approach that highlights areas where LVLMs struggle, especially in tasks requiring complex multi-image reasoning.

- Quality: The experimental methodology is robust and thorough. The authors test 24 popular LVLMs, both open-source and closed-source, providing a comprehensive view of current capabilities. Their use of multi-dimensional analysis tools, including supervised fine-tuning (SFT) and task maps, provides depth to the evaluation, highlighting model strengths and weaknesses across various task types. This approach demonstrates rigorous scientific methodology and enhances the reliability of the conclusions drawn.

- Significance: MMIU has high potential impact for the vision-language research community, particularly given the increasing relevance of multi-image tasks in real-world applications (e.g., video understanding, 3D reasoning). By providing a benchmark with a broad set of tasks and image relationships, MMIU allows for a more nuanced understanding of LVLMs’ strengths and limitations. The insights from this benchmark can guide future LVLM improvements, pushing the field towards more sophisticated models that handle complex visual contexts and multi-image data effectively.

**Weaknesses:**

- Reliability of GPT-4o-Generated Data: Although MMIU’s large dataset is a significant contribution, its reliance on GPT-4o for generating questions and responses raises concerns about potential bias. Given that GPT-4o itself may have inherent preferences or limitations in multi-image reasoning, this could inadvertently affect the construction of questions and answer choices. Additionally, if GPT-4o has limitations in multi-image reasoning (as noted by the authors), this brings into question whether the generated tasks and responses are fully reliable. While human verification can ensure the accuracy of correct answers, it may not catch subtle model biases embedded in the answer choices, potentially skewing results. A clearer discussion or additional safeguards against GPT-4o’s influence on task design would improve the benchmark’s objectivity and robustness.

- Limited Discussion on Practical Applicability: While MMIU’s tasks are inspired by real-world applications, the paper could strengthen its relevance by addressing practical implications more explicitly. For instance, the authors could highlight specific MMIU tasks that align with high-stakes applications—such as high-level reasoning and long-term memory tasks relevant to autonomous driving, surveillance, and medical imaging. Connecting benchmark tasks to these domains would make the work more relatable and impactful for applied researchers. Additionally, discussing how MMIU might be adapted or extended to cover emerging application areas would further enhance the benchmark’s future-facing value and adaptability.

**Questions:**

- Clarification on GPT-4o’s Role in Data Generation: Could you elaborate on how GPT-4o’s limitations in multi-image reasoning are managed during data generation? Specifically:

- How do you ensure that GPT-4o’s generated questions and answer choices are unbiased and accurately reflect the task complexity, given that it might have inherent reasoning limitations?

- To what extent was human verification involved, especially in preventing GPT-4o's possible biases from influencing the multiple-choice options? Further clarification on this process would strengthen confidence in the benchmark’s objectivity.

---

> ### Author Response · Authors · 2024-11-21
> **Response to Reviewer 4Ca4 (Q1)**
>
> **In order to observe the changes in the PDF more clearly, we have marked the changes in the new PDF with green marks based on your comments.**
>
> Thank you for raising a series of insightful questions. Considering that GPT-4o was only used to generate part of the incorrect options in the dataset, we have summarized your concerns into the following three key questions。
> - **Q1:** What measures were taken to ensure that the generated incorrect options are genuinely incorrect and sufficiently challenging.
>
> - **Q2:** Limited Discussion on Practical Applicability。
>
> - **Q3:** How do you ensure the high quality of incorrect options generated by GPT-4o
>
> Q1: What measures were taken to ensure that the generated incorrect options are genuinely incorrect and sufficiently challenging.
>
> A1: In generating incorrect options, we employ a variety of task-specific methods, carefully designed to ensure the quality and challenge of the distractors:
>
> 1. **For tasks where the ground truth is a numerical value (e.g., object tracking):**
>    We use Python scripts to introduce random perturbations to the ground truth values within a fixed standard deviation. This approach generates plausible but incorrect options that remain close to the original values, increasing the difficulty of the task.
> 2. **For tasks where the ground truth is a long text (e.g., multi-image captioning):**
>    We use a **few-shot prompting strategy** with GPT-4o. The prompts are designed to help GPT-4o understand what constitutes a challenging incorrect option. For example, given a ground truth caption such as *"The woman and man were eating some street food,"* we provide examples of incorrect options with varying quality scores, as shown in **Table 9 [in the response to reviewer 4Ca4 (Q3)](https://openreview.net/forum?id=WsgEWL8i0K&noteId=a2Fc45rwoU)** . GPT-4o is then tasked with generating high-quality distractors based on these examples. Experiments (as discussed in our **A3 response**) and manual inspections confirm that this method produces high-quality incorrect options.
>
> Throughout this process, we tailor the generation of incorrect options to the unique characteristics of each task type. This ensures that the multiple-choice questions in MMIU are not **naive or trivial**. Our approach aligns with methodologies used in SEED-Bench [1] and MMT-Bench [2], which demonstrate that combining human design with LLM assistance reliably produces high-quality incorrect options.
>
>
>
> [1] SEED-Bench: Benchmarking Multimodal LLMs with Generative Comprehension
>
> [2] MMT-Bench: A Comprehensive Multimodal Benchmark for Evaluating Large Vision-Language Models Towards Multitask AGI

---

> ### Author Response · Authors · 2024-11-21
> **Response to Reviewer 4Ca4 (Q2)**
>
> Q2: Limited Discussion on Practical Applicability...
>
> A2: Thank you for your valuable suggestion. We greatly appreciate your insights, which help us emphasize the practical relevance of MMIU. As shown in **Table 7 below**, we discuss this from four perspectives:
>
> 1. **Embodied Systems and Robotics**
>    Unlike other benchmarks that focus primarily on semantic-related tasks, MMIU includes a rich set of spatial tasks that enable extensive exploration of the visual capabilities required for high-level planning in **embodied systems** and their applications in **robotics**. For example, in the **Egocentric Video Question Answering (EVQA)** task, **InternVL2** outperforms other LVLMs, demonstrating its potential for physical interactions with the environment.
> 2. **Autonomous Driving**
>    MMIU also includes spatial tasks that are critical for **autonomous driving**, such as **3D depth estimation**, which provides vehicles with accurate, real-time spatial awareness and situational understanding in dynamic environments. Our evaluations show that most LVLMs face challenges in this task, highlighting their current limitations in spatial reasoning capabilities.
> 3. **Surveillance Systems**
>    MMIU incorporates semantic tasks such as **Vehicle Re-Identification** and **Person Re-Identification**, which are essential for improving the effectiveness and capabilities of **surveillance systems**. In these tasks, **InternVL2** outperforms other LVLMs, including the closed-source model GPT-4o and the multi-image SFT model Mantis, showcasing its strong potential in the surveillance domain.
> 4. **Autonomous Control Applications**
>    MMIU includes temporal tasks like **GUI app recognition** and **GUI next action prediction**, which have significant practical value for autonomous control systems relying on visual inputs. Our results show that **Claude3.5** demonstrates relatively better performance in these tasks compared to other LVLMs, highlighting its growth potential for future development.
>
> **Conclusion**
> MMIU serves as a comprehensive benchmark encompassing diverse tasks, providing critical insights into the strengths and limitations of current LVLMs across spatial, semantic, and temporal dimensions. By doing so, it contributes to driving advancements in real-world applications.
>
> We provide further details in **Appendix B.3: Practical Applicability Discussion**. Once again, thank you for your insightful suggestions, which have been instrumental in refining our analysis and highlighting the broader significance of our work.
>
>
>
>
>
> | Model              | Egocentric Video Question Answering | 3D Depth Estimation | Vehicle Retrieval | Person Re-ID | GUI App Recognition | GUI Next Action Prediction |
> | ------------------ | ----------------------------------- | ------------------- | ----------------- | ------------ | ------------------- | -------------------------- |
> | GPT-4o             | 57.5                                | 24.0                | 68.0              | 61.5         | 73.5                | 46.5                       |
> | Claude3.5-Sonnect  | 53.5                                | 23.5                | 79.0              | 77.5         | 88.5                | 55.0                       |
> | Mantis-idefics2-8B | 54.5                                | 23.5                | 70.5              | 57.5         | 78.0                | 34.0                       |
> | InternVL2-Pro      | 63.0                                | 25.5                | 83.5              | 82.0         | 91.5                | 40.5                       |
>
> ​															Table7: The performance of some models on MMIU's practical tasks

---

> ### Author Response · Authors · 2024-11-21
> **Response to Reviewer 4Ca4 (Q3)**
>
> Q3: How do you ensure the high quality of incorrect options generated by GPT-4o
>
> A3: On the one hand, we explain that although GPT-4o has only moderate multi-image reasoning capabilities, it can still generate competitive incorrect options using the approach described in **A1**:
>
> 1. **The task is to generate incorrect options, not solve the problem:**
>    Instead of requiring GPT-4o to solve the question, we provide it with the question and the correct answer to generate incorrect options. By presenting the correct answer, the difficulty of understanding the question is effectively reduced. This is because GPT-4o can identify which image or region the question focuses on and then perform targeted reasoning based on the correct answer. Once the question is understood, GPT-4o can generate incorrect options by reasoning about semantic differences between the options and the images.
> 2. **Task-specific few-shot prompts for incorrect option generation:**
>    We design specific few-shot prompts tailored to different tasks to enhance the quality of incorrect options. For example, given a ground truth caption such as *"The woman and man were eating some street food,"* we provide examples of incorrect options with varying quality scores, as shown in **Table 9 below**. GPT-4o is then tasked with generating high-quality distractors based on these examples.
>
> This targeted approach allows GPT-4o to produce challenging and semantically relevant incorrect options, even with its moderate multi-image reasoning abilities.
>
> On the other hand, We demonstrate that the incorrect options in MMIU are of high quality through two experiments and manual evaluation:
>
> 1. **Performance without Visual Input:**
>    As shown in **Table 3** of the main text, when images are removed and LLMs are tasked with answering questions using only the questions and options, the overall accuracy of multiple LLMs approaches random guessing. This indicates that it is difficult to distinguish between correct and incorrect options solely based on the options, demonstrating that the incorrect options are sufficiently challenging.
>
> 2. **Semantic Similarity Analysis:**
>    We randomly select **500 samples** whose incorrect options are generated by GPT-4o. Using OpenAI’s **text-embedding-3-small** model, we extract embeddings for both the correct and incorrect options and compute their cosine similarity. As a baseline, we randomly select incorrect options from other samples within the same task, ensuring that these incorrect options differ from the correct option of the current sample.
>
>    As shown in **Table 8 below**, the results show that the semantic similarity between GPT-generated incorrect options and the correct options is significantly higher than the baseline. This indicates that GPT-generated incorrect options are more semantically similar to the correct options, making them more challenging.
>
> | Similarity | Semantic | Temporal | Spatial |
> | ---------- | -------- | -------- | ------- |
> | GPT        | 0.64     | 0.56     | 0.54    |
> | Random     | 0.21     | 0.18     | 0.15    |
>
> ​											Table8: Comparison of semantic similarity between wrong options and correct options

---

> ### Author Response · Authors · 2024-11-21
> **Response to Reviewer 4Ca4 (Q3)**
>
> 3. **Human Evaluation of Incorrect Options:**
>    We use the **500 samples** selected in experiment 2 and recruit **5 senior undergraduate students** to evaluate the quality of the incorrect options. Each sample is assigned to all 5 annotators, who score the quality of the incorrect options based on the provided guidelines (examples shown in **Table 9 below**). Annotators evaluate the options on a standardized scale. The cost of this manual evaluation is detailed in **Table 10 below**.
>
>    The results show an average score of **0.93 (after normalization)** across all annotators, with no samples receiving a score of **0**. Additionally, the average variance among annotators’ scores is **0.06**, indicating strong agreement among annotators. This consistency demonstrates that GPT-4o-generated incorrect options in MMIU are widely considered to be challenging and effective distractors.
>
>
>
> | Images      | Question Description                                   | Correct Answer                                 | Incorrect Option                               | Quality of Incorrect Options | Reason                                                       |
> | ----------- | ------------------------------------------------------ | ---------------------------------------------- | ---------------------------------------------- | ---------------------------- | ------------------------------------------------------------ |
> | Images list | Select the option with describe these images correctly | The woman and man were eating some street food | The man and woman were cooking dinner at home  | Moderate (1)                 | The option is indeed wrong, but the behavior and scenario have changed |
> |             |                                                        |                                                | The couple were eating some food in the street | Poor (0)                     | The option is same with the correct option                   |
> |             |                                                        |                                                | The couple were eating some food in the park   | High (2)                     | The option is indeed wrong, and only scenario has changed    |
>
> ​																Table9: Example of incorrect option quality annotation
>
>
>
>
>
> | Samples | Avg. time (per sample) | Num. People | Background      | Cost     | Total Cost |
> | ------- | ---------------------- | ----------- | --------------- | -------- | ---------- |
> | 500     | 30s-1min               | 5           | College Student | 8USD / h | 200USD     |
>
> ​																				Table10: Cost of manual inspection

---

> ### Author Response · Authors · 2024-11-25
> **Looking forward to your reply !**
>
> We express our sincere appreciation for the valuable suggestions regarding this paper. In response, we have provided thorough and detailed explanations. And we kindly request that you consider updating the score accordingly. If there are any further questions, we are delighted to offer additional clarifications and conduct further experiments!

---

> ### Author Response · Authors · 2024-11-30
> **Looking forward to your reply !**
>
> As the discussion period is nearing its end, we sincerely thank you for your valuable feedback and are happy to offer any further explanations if needed. If you have additional questions, please feel free to reach out at any time. We genuinely hope you will reconsider your rating, and we are committed to addressing any issues you may have.

---

### Official Review · Reviewer_avPS · 2024-11-02

**Soundness:** 3
**Presentation:** 3
**Contribution:** 3
**Rating:** 6
**Confidence:** 4

**Summary:**

This work introduces MMIU, a benchmark to evaluate a LVLM's capability of understanding information when given multiple images simultaneously. MMIU covers a wide range of established cognitive tasks on semantics and compositionality in fine categories.

**Strengths:**

This work offers meticulous details to the design of the benchmark. The authors have shown great effort in the data curation process while following a logical main plot line. I particularly find the results that show the LVLMs pre-trained with only single-image content may outperform their multi-image counterparts of great research insight.

**Weaknesses:**

Please find my two major concerns down below:

1. My bigger issue about this paper is regarding the presentation of experimental results. **I find the benchmarking results are presented in a very cumbersome way.** Table 3 is the best example to illustrate this major weakness. The authors put in all the fine-category results by every single subtask. Although the comprehensiveness is appreciated, such overkill-style of presentation makes it hard to follow, so that we readers may not obtain a clear intuition about where and how well each baseline model thrives or lags against one another. This 'way-too-much-details' problem is also present in Figure 4 of the task maps. I highly recommend the authors show more summarized performance on the coarse categories, such the 3 big ones of Semantic/Spatial/Temporal in the main text, while leaving the finer categories in the Appendix.

2. **Potential position bias in the unanswerable set is unaddressed.** . MMIU's QA test cases are all multiple-choices. However, according to Zheng et al, 2023, the position bias is a significant issue when constructing benchmarks for evaluating LLMs (esp. on non-GPT methods). Basically speaking, the position bias will be prominent if the 'None of the above' option for the unanswerable set is always Option D, while known LLMs tend to favor the first option. This means models intend to give away deflated results shown in Table 12 for the unanswerable tasks. Have the authors tried shuffling the option orders? I am not seeing particular evidence for addressing the position bias in Section 3.2 as supposed to be.

Zheng et al, 2023. Judging LLM-as-a-Judge with MT-Bench and Chatbot Arena

**Questions:**

Please find my concerns and suggestions in the Weakness section above.

Typo: Line 1389, Table reference is missing.

Update (Nov 23): Raising my ratings according to the authors' response

---

> ### Author Response · Authors · 2024-11-21
> **Response to Reviewer avPS (Q1)**
>
> **In order to observe the changes in the PDF more clearly, we have marked the changes in the new PDF with red marks based on your comments.**
>
> Q1: My bigger issue about this paper is regarding the presentation of experimental results ...
>
> A2: Thank you for your valuable suggestions regarding our writing. We implement numerous revisions in the resubmitted paper to improve its readability and clarity:
>
> 1. **Revised Main Results Table**
>    We replace the main results table in the main text with the average accuracy of LVLMs across seven types of image relationships. In **Section 4.2**, we analyze the overall trends across different models, while in **Section 4.3.1**, we provide a detailed performance analysis of different models under varying image relationships.
> 2. **Enhanced Focus in Figure 4**
>    - We highlight the clustering results of the **task map** from the original **Figure 4(a)** by emphasizing the shared characteristics of each cluster. These details, along with annotations, are now presented in **Appendix Figure 11**, as our primary focus is the analysis of model performance based on clustering.
>    - For the original **Figure 4(b)**, which presents model performance, we simplify the heatmap to create the current **Figure 4** by categorizing tasks into clusters and grouping models. This improves readability while maintaining essential insights. Additional analyses are provided in **Appendix C.2**.
> 3. **Separated Small Tables for Independent Analysis**
>    For smaller tables, such as **Appendix Table 16** and **Table 17**, we extract key results (e.g., representative models' performance on different image relationships or tasks) and conduct more detailed discussions in **Appendix B.3**.
> 4. **Expanded Visualizations for Important Content**
>    For example, we revise **Appendix Figure 9** to include more detailed visualizations categorized by image relationships. This refinement addresses the broad scope of **Section 4.4 (Error Analysis)** by providing a more targeted and visually interpretable breakdown, thereby improving the presentation of key findings.
>
> We appreciate your constructive feedback, which significantly contributes to enhancing the clarity and presentation of our work.

---

> > ### Comment · Reviewer_avPS · 2024-11-23
> > **Impressive updates, with a follow-up question**
> >
> > I appreciate the authors' effort in addressing the concerns raised by not only me but also all other reviewers. The updated draft is now in a way better shape. I can now see a clear merit in the new Table 3, where a large number of existing LVLMs can't even beat random guessing on all categories of MMIU - I believe this point is worth being emphasized in the main result section.
> >
> > And I would like to have a minor follow-up question - Is there any baseline performance by human? What are the *theoretical upper bounds* by human performance? It's okay if the authors do not have them given the scale of MMIU. But if they do, it would further clarify the potential of MMIU on how much room is left for improvements in future work.
> >
> > With all being said, I am impressed and will update my assessments very soon.

---

> > > ### Author Response · Authors · 2024-11-25
> > > **Response to Reviewer avPS (Q1)**
> > >
> > > Q1: Why some models are less accurate than random selection
> > >
> > > A1: Thank you for your insightful question. This is indeed an important phenomenon, and we provide a detailed explanation for it. We have also included this discussion in **Appendix B.3 (Why some models are less accurate than random selection)** of the paper.
> > >
> > > - **Irrelevant answers**: Due to occasional irrelevant answers from the models (including answers not in the options, garbled text, repeated issues, and repeated responses until cutoff), we label answers as **"Z"** when they do not correspond to any of the options in the question. This approach is also adopted by **VLMEvalKit**. This is one of the main reasons why the model’s accuracy is lower than random selection. Below, we provide a detailed analysis of some representative models.
> > > - **Multi-image input models**: Errors occur frequently due to limitations in model capability or instruction-following ability, leading to significantly lower accuracy. We summarize the probability of "Z" answers for several models in the **Table 1 below.** Our findings are as follows:
> > >   - **Spatial relationship tasks**: The highest probability of generating "Z" answers occurs in spatial relationship tasks. This is because spatial data is relatively scarce, leading to insufficient model training. Additionally, spatial tasks tend to be more complex (some involve visual tasks), and models with limited spatial ability tend to generate irrelevant responses. As shown in **Table 3 in main text**, even for powerful closed-source models, performance on spatial tasks remains below the overall average.
> > >   - **Temporal relationship tasks**: The lowest probability of "Z" answers occurs in temporal relationship tasks. This is because video data is the most readily available during training, enabling models to develop relatively strong capabilities in this area. As shown in **Table 3 in main text**, for closed-source models, performance on temporal tasks tends to be above the overall average.
> > >   - **Instruction-following ability**: As the model size decreases, instruction-following ability weakens, leading to off-topic errors. In the error case analysis in **Appendix B.4**, Mantis' instruction-following ability is significantly lower than that of closed-source models and larger open-source models like **InternVL2-Pro**. This results in situations where the model provides answers not in the options.
> > > - **Single-image models**: Lower accuracy or random selection errors primarily result from insufficient model capability or difficulty in generalizing long contexts. We give the main result in **Table 1 below.**
> > >   - **Concatenated image input**: When we use the concatenated image method, we find that "Z" answers are less frequent in single-image models. Instead, these models tend to select incorrect answers. We believe this is because the models struggle to **interpret concatenated images**.
> > >   - **Concatenated token input**: When we test with the concatenated token method, we observe similar issues to multi-image models. This is due to long context causing off-topic or irrelevant responses. For some single-image models with average capabilities, even if long context input is supported, the training samples mostly consist of single images, which limits generalization. Specifically, we calculate the average number of images per class for different image relationships: **10.02** for **Temporal**, **5.94** for **Semantic**, and **7.04** for **Spatial**. As the number of images increases, the probability of generating "Z" answers also increases.
> > >
> > >
> > > | Model                           | Temporal | Semantic | Spatial | Overall |
> > > | ------------------------------- | -------- | -------- | ------- | ------- |
> > > | XComposer2_1.8b                 | 13.6     | 20.1     | 46.1    | 28.3    |
> > > | flamingov2                      | 19.1     | 10.7     | 17.3    | 15.5    |
> > > | xgen_mm_single                  | 15.1     | 29.8     | 35.8    | 27.9    |
> > > | llava_v1.5_7b（concat img）     | 0.6      | 0.1      | 0.9     | 0.5     |
> > > | sharecaptioner（concat img）    | 15.1     | 3.3      | 8.6     | 8.5     |
> > > | llava_v1.5_7b （concat token）  | 55.0     | 5.1      | 38.1    | 29.5    |
> > > | sharecaptioner （concat token） | 38.9     | 25.0     | 34.1    | 32.2    |
> > >
> > > Table 1: The probability of different models being judged as Z under 7 image conditions.

---

> ### Author Response · Authors · 2024-11-21
> **Response to Reviewer avPS (Q2)**
>
> Q2: Potential position bias in the unanswerable set is unaddressed...
>
> A2: Thank you for your question, and we apologize for any confusion caused.
>
> In fact, we consider this aspect not only when designing the unanswerable set but also throughout the overall benchmark. The correct options in all tasks are always randomized in order. As shown in **Table 6 below**, for the unanswerable set, the placement of the unanswerable option is also consistently randomized. This approach minimizes the potential **position bias** issue you mention to the greatest extent possible.
>
> Additionally, we provide further clarification in **Section 4.1 (Evaluation Method)** to ensure this point is clearly addressed.
>
> Thank you again for raising this important consideration.
>
> | Position | A     | B     | C     | D     |
> | -------- | ----- | ----- | ----- | ----- |
> | Portion  | 24.6% | 24.7% | 25.6% | 25.1% |
>
> ​								Table 6: The probability of unanswerable options appearing in different positions is almost equal

---

> ### Author Response · Authors · 2024-11-25
> **Response to Reviewer avPS (Q2)**
>
> Q2: Is there any baseline performance by human
>
> A2: We would like to thank the reviewer for their insightful comments. These suggestions have been very helpful, and we have given them detailed consideration. Below, we address each point in turn.
>
> 1. **Recruitment of Human Evaluators**: We continue to invite the five students employed at **[Response to Reviewer 4Ca4 (Q3)](https://openreview.net/forum?id=WsgEWL8i0K&noteId=a2Fc45rwoU)** to continue the task of answering questions. Given time constraints, we select 41 tasks out of the original 52 which are suitable for human evaluation. Because some tasks in MMIU, such as tracking-related tasks, are more suited for machines to answer, while human evaluation is overly labor-intensive. From each task, we sample 10 instances, resulting in a total of 410 samples for human evaluation. For each sample, we assign 5 students to answer. The average accuracy across 7 relationships is then calculated, as shown in **Table 2 below**. The manual effort involved in this process is recorded in **Table 3 below**.
>
> 2. **Evaluation Results**: The results show that the current model performs relatively well for Low-Level image relationships and High-Level (subjective) tasks, approaching human scores. However, the overall performance still lags behind human performance, particularly in spatial relationships, where the gap between model and human ratings is most pronounced. This highlights the current limitations of the model in handling multi-image tasks.
>
> 3. **Future Work**: We will expand the sample size for further human evaluation. For some visual tasks, we plan to invite more senior annotators and use **specialized visual models** (e.g., Depth Anything) to test the accuracy of these tasks. This will provide a more comprehensive evaluation of MMIU's **theoretical upper bounds**.
>
> Once again, thank you for your valuable feedback. If the reviewer has any further questions or concerns, please do not hesitate to ask, and we will do our best to address them.
>
> | Samples | Avg. time (per sample) | Num. People | Background      | Cost     | Total Cost |
> | ------- | ---------------------- | ----------- | --------------- | -------- | ---------- |
> | 420     | 1min-2min              | 5           | College Student | 8USD / h | 384 USD    |
>
> ​																				Table2: The cost of human annotators.
>
> | Overall | Overall | Discrete | Continuous | Low-level | High-level(sub) | High-level(obj) | Two-D | Three-D |
> | ------- | ------- | -------- | ---------- | --------- | --------------- | --------------- | ----- | ------- |
> | Human   | 92.8    | 89.5     | 92.8       | 98.5      | 96.6           | 94.6            | 88.0  | 92.0    |
> | GPT-4o  | 55.5    | 58.2     | 53.7       | 84.0      | 69.2            | 57.5            | 41.7  | 55.4    |
>
> Table 3: Comparison of human results and GPT-4 results across 7 image relationships.

---

> > ### Comment · Reviewer_avPS · 2024-11-27
> > **Very thoughtful.**
> >
> > Much obliged for the additional clarifications. I see all of my concerns have been rightfully addressed. Good job for completing the impressive work!

---

> ### Author Response · Authors · 2024-11-28
> **Response to Reviewer avPS**
>
> Thanks for your interest in our research, which have been instrumental to our work. We are deeply grateful for your thorough and constructive review! If there are any further questions, we are delighted to offer additional clarifications and conduct further experiments. We eagerly look forward to your reply!

---

### Official Review · Reviewer_bVMZ · 2024-11-04

**Soundness:** 3
**Presentation:** 3
**Contribution:** 3
**Rating:** 6
**Confidence:** 4

**Summary:**

This paper introduces the Multimodal Multi-image Understanding (MMIU) benchmark, a comprehensive evaluation suite designed to assess Large Vision-Language Models (LVLMs) across a wide range of multi-image tasks. MMIU encompasses seven types of multi-image relationships, including semantic, temporal, and spatial aspects, and consists of 52 tasks, 77K images, and 11K meticulously curated multiple-choice questions. The authors evaluate 24 popular LVLMs, including both open-source and proprietary models, on the MMIU benchmark and identify significant challenges in multi-image comprehension, particularly in tasks involving spatial understanding. Through multi-faceted analytical experiments, they provide valuable insights for future model and data improvements.

**Strengths:**

1. The paper is well-organized and clearly presented, with a detailed background section, methodological descriptions, and experimental results.
2. MMIU is the first benchmark to comprehensively evaluate the performance of LVLMs on multi-image tasks. It covers a wide range of multi-image relations and tasks, including semantic, temporal and spatial relations, as well as various image modalities.
3. The design of the MMIU allows it to be easily extended to new multi-image tasks and image modalities. This makes it a useful tool for future research.

**Weaknesses:**

1. MMIU is large in scale, including 77,659 images and 11,698 multiple-choice questions. This may require a lot of computing resources to run experiments and perform analysis.
2. The paper shows that multi-graph understanding requires the capabilities of semantic, temporal and spatial relations. However, most current models do not model the latter two capabilities in the pre-training phase, which can lead to a lack of practical significance in the comparison of various models

**Questions:**

1. The authors mention how they collected and standardized the data to create the MMIU. However, they do not provide detailed information about the data collection process, such as which data sources and data processing techniques they used.
2. The strong capability in single-image understanding is the foundation of multi-image comprehension. As we know, the ability of single-image understanding comes from powerful language models, such as MMMU benchmark, can you give some performance of pure language models.
3. I hope to provide some model performance that add interleave data or video data in the pre-training or post-training stage.

---

> ### Author Response · Authors · 2024-11-21
> **Response to Reviewer bVMZ (Q1)**
>
> **In order to observe the changes in the PDF more clearly, we have marked the changes in the new PDF with blue marks based on your comments.**
>
> Q1: MMIU ... This may require a lot of computing resources
>
> A1: To mitigate the computational cost caused by the large number of tasks, similar to MathVista [1], we construct a smaller test set, **testmini**, by randomly sampling 20 examples from each of the 52 tasks, resulting in a total of 1,040 examples. While preserving the diversity of MMIU, our experiments show that the results on **testmini** effectively reflect those on the full test set. These findings appear in **Table 3** in the main text and **Table 15** in the appendix, where we report results on both the full test set and **testmini**.
>
> Additionally, we perform a quantitative analysis of the testing cost for **testmini**. As shown in **Table 1 below**, most models complete the evaluation on a single A100 GPU within one hour. Even for the closed-source model, GPT-4o, the testing cost remains manageable, requiring approximately $10 and one hour to complete.
>
> Moving forward, we plan to evaluate more models on **testmini** and provide a **convenient interface** to enable the community to test and compare models on MMIU seamlessly.
>
> Once again, thank you for your thoughtful suggestions, which are instrumental in improving our work.
>
> | Model                 | Size | BS（per device） | Resources     | Times | Memory   |
> | --------------------- | ---- | ---------------- | ------------- | ----- | -------- |
> | GPT-4o                | -    | 4                | USD 8.57      | 0.51h | -        |
> | LLaVA-Next-Interleave | 7B   | 3                | 1 x A100-80GB | 0.04h | 60381MiB |
> | Mantis-idefics2       | 8B   | 6                | 1 x A100-80GB | 0.15h | 73896MiB |
> | XGen-MM               | 4B   | 7                | 1 x A100-80GB | 0.14h | 75082MiB |
> | InternVL-Chat-V1.5    | 26B  | 1                | 1 x A100-80GB | 0.89h | 71619MiB |
> | InternVL2             | 8B   | 1                | 1 x A100-80GB | 0.97h | 79292MiB |
> | LLaVA-1.5             | 7B   | 4                | 1 x A100-80GB | 0.02h | 64462MiB |
>
> ​															Table1: Resource consumption of some models in MMIU's testmini set
>
> [1] MathVista: Evaluating Math Reasoning in Visual Contexts

---

> ### Author Response · Authors · 2024-11-21
> **Response to Reviewer bVMZ (Q2)**
>
> Q2: The paper shows that ... lack of practical significance
>
> A2: Thank you for your constructive feedback. We have carefully considered your comments and would like to address them in detail below:
>
> **1）Multi-Image Training for Temporal and Spatial Relationships in LVLMs**
> We acknowledge that earlier single-image models like LLaVA and MiniGPT-4 primarily focused on modeling semantic relationships. However, with the advancement of large vision-language models (LVLMs), many now incorporate multi-image data during training to capture temporal and spatial relationships. For instance:
>
> - **Mantis** utilizes **Mantis-Instruct** [1] for multi-image supervised fine-tuning (SFT), which includes datasets like VIST [2] and NExT-QA [3], emphasizing temporal relationships.
> - **LLaVA-Next-Interleave** employs **M4-Instruct** [4] for multi-image SFT, incorporating embodied tasks such as 3D grounding that highlight spatial reasoning.
> - **InternVL2** utilizes **OmniCorpus** [5] during pretraining, a dataset with 10 billion interleaved image-text pairs, encompassing diverse semantic, temporal, and spatial relationships. Additionally, in its Stage 2 training, it incorporates video data (e.g., **EgoTaskQA** [6]), enabling it to effectively handle tasks involving both spatial and temporal relationships.
>
> These advancements demonstrate the growing focus on multi-image capabilities, aligning with the evolving needs of LVLMs.
>
> **2) Practical Significance of MMIU Task Design**
> We also emphasize the practical value of MMIU’s task design. Beyond general high-level semantic tasks, MMIU incorporates tasks with real-world applications, relevant to fields like robotics and autonomous driving. As shown in **Table 2 below**, we analyzed selected tasks and model performances, highlighting the following:
>
> - MMIU evaluates high-level visual capabilities for **embodied systems**, such as robotics. For example, **InternVL2-Pro** demonstrates strong potential in egocentric tasks like EVQA.
>
> - MMIU includes spatial reasoning tasks critical for **autonomous driving**. Results reveal that current LVLMs face challenges in tasks like 3D depth estimation.
>
> - MMIU addresses semantic tasks relevant to **surveillance applications**. **InternVL2-Pro** excels in Vehicle and Person Re-Identification, outperforming other LVLMs in these scenarios.
>
> - MMIU incorporates temporal tasks for **autonomous control**, where **Claude 3.5** shows potential in GUI-related applications.
>
> We provide a more detailed discussion of these findings in **Appendix B.3: Practical Applicability Discussion**.
>
> Once again, thank you for your insightful suggestions, which have been instrumental in refining our analysis and emphasizing the broader significance of our work.
>
> | Model              | Egocentric Video Question Answering | 3D Depth Estimation | Vehicle Retrieval | Person Re-ID | GUI App Recognition | GUI Next Action Prediction |
> | ------------------ | ----------------------------------- | ------------------- | ----------------- | ------------ | ------------------- | -------------------------- |
> | GPT-4o             | 57.5                                | 24.0                | 68.0              | 61.5         | 73.5                | 46.5                       |
> | Claude3.5-Sonnect  | 53.5                                | 23.5                | 79.0              | 77.5         | 88.5                | 55.0                       |
> | Mantis-idefics2-8B | 54.5                                | 23.5                | 70.5              | 57.5         | 78.0                | 34.0                       |
> | InternVL2-Pro      | 63.0                                | 25.5                | 83.5              | 82.0         | 91.5                | 40.5                       |
>
> ​										Table2: The performance of some models on MMIU's practical tasks
>
>
>
>
>
>
>
> [1] Mantis: Interleaved Multi-Image Instruction Tuning
>
> [2] Visual Storytelling
>
> [3] NExT-QA: Next Phase of Question-Answering to Explaining Temporal Actions
>
> [4] LLaVA-NeXT-Interleave: Tackling Multi-image, Video, and 3D in Large Multimodal Models
>
> [5] OmniCorpus: A Unified Multimodal Corpus of 10 Billion-Level Images Interleaved with Text
>
> [6] EgoTaskQA: Understanding Human Tasks in Egocentric Videos

---

> ### Author Response · Authors · 2024-11-21
> **Response to Reviewer bVMZ (Q3)**
>
> Q3: The authors ... data processing techniques used
>
> A3: Thank you for pointing out this issue. We have supplemented our explanation with more detailed descriptions as follows:
>
> In **Section 3.2**, we outlined the general process of data collection, which involves the following steps:
>
> 1. **Defining tasks based on image relationships.**
>    We first define specific tasks corresponding to the relationships among images. Detailed explanations of these tasks and their descriptions are provided in **Table 7** in the Appendix.
> 2. **Identifying datasets that reflect the tasks.**
>    We then identify datasets suitable for each defined task, primarily through manual collection from sources such as Kaggle, GitHub, and Papers with Code. The specific **data sources** for each task are listed in **Table 7** of the Appendix.
> 3. **Transforming datasets into multiple-choice questions.**
>    Next, we convert the collected datasets into a unified multiple-choice format:
>    - First, the datasets are standardized into a unified metadata format, with an example provided in **Table 9** of the Appendix.
>    - Then, we generate incorrect options based on the ground truth. For tasks where the ground truth is numerical (e.g., object detection), we use algorithmic perturbations to create distractors. For tasks involving text generation (e.g., captions), we employ a few-shot prompting approach with GPT-4o to generate high-quality distractors. For instance, given the ground truth "The woman and man were eating some street food," we provide three examples of distractors with different quality scores in **Table 9 in the [Response to Reviewer 4Ca4 (Q3)](https://openreview.net/forum?id=WsgEWL8i0K&noteId=a2Fc45rwoU)**, guiding GPT-4o to generate high-quality incorrect options.
>
> Detailed descriptions of the key steps in data generation, such as metadata standardization and distractor generation, are provided in **Appendix A.4**.
>
> We hope this additional information clarifies our data collection and processing methodology. Thank you again for your valuable feedback, which has helped us improve the clarity of our work.

---

> ### Author Response · Authors · 2024-11-21
> **Response to Reviewer bVMZ (Q4)**
>
> Q4: ...can you give some performance of pure language models.
>
> A4: Thank you for your suggestions. We have supplemented this part with additional experiments, as detailed below.
>
> 1. **Performance without Visual Input**
>    As shown in **Table 3 below**, when images are excluded and only questions and options are provided, the scores of all LLMs are nearly equivalent to random guessing. This suggests that the distractors generated by GPT-4o are sufficiently challenging and meaningful. Furthermore, it highlights that MMIU tasks require visual information, demonstrating their distinct multimodal characteristics.
> 2. **Caption-Based Reasoning**
>    We also explore an approach where GPT-4o extracted captions for each input image, and then LLMs were tasked with answering the questions based on the captions and options. As shown in **Table 4 below**, this approach led to significant improvement compared to random guessing. However, it still falls short compared to direct inference using LVLMs. For instance, GPT-4o’s direct inference achieves an overall accuracy of **55.5%**, outperforming the caption-based reasoning accuracy of **48.7%**. This finding underscores that accurately understanding the semantic information in images is essential for multi-image understanding tasks, while some tasks in MMIU require more **low-level visual information** to be effectively solved.
>
> We provide a more detailed analysis of these findings in **Appendix B.3**.
>
> Thank you again for your valuable feedback, which has helped enhance the depth and clarity of our evaluation.
>
> | Model       | Temporal | Semantic | Spatial | Overall |
> | ----------- | -------- | -------- | ------- | ------- |
> | Random      | 24.0     | 30.9     | 26.6    | 27.1    |
> | GPT-4o      | 28.2     | 16.1     | 25.7    | 23.1    |
> | Llama3.1    | 29.3     | 31.1     | 28.3    | 29.5    |
> | Qwen2.5     | 28.6     | 35.3     | 28.5    | 30.9    |
> | InternLM2.5 | 32.2     | 31.4     | 31.2    | 31.5    |
>
> ​																	Table3: Some LLMs performance on MMIU
>
> | Model       | Temporal | Semantic | Spatial | Overall |
> | ----------- | -------- | -------- | ------- | ------- |
> | GPT-4o      | 53.9     | 53.8     | 42.5    | 48.7    |
> | Llama3.1    | 40.0     | 43.1     | 35.2    | 39.2    |
> | Qwen2.5     | 44.4     | 45.2     | 34.2    | 40.3    |
> | InternLM2.5 | 42.3     | 42.1     | 33.2    | 38.2    |
>
> ​										Table4: Some LLMs with caption generated by GPT-4o performance on MMIU

---

> ### Author Response · Authors · 2024-11-21
> **Response to Reviewer bVMZ (Q5)**
>
> Q5: ... add interleave data or video data in the pre-training or post-training stage.
>
> A5: Your question is highly insightful, and we have conducted additional tests with a broader range of models to analyze the role of multi-image data at different training stages.
>
> We consider the two most common training stages: **pretraining** and **SFT (supervised fine-tuning)**. Based on the inclusion of multi-image data at these stages, we categorized models into four groups. As shown in **Table 5 below**, we find the following:
>
> 1. **Multi-image SFT consistently improves performance**:
>    Regardless of whether multi-image pretraining is conducted, models like **LLaVA-Next-Interleave** and **XGen-MM-Phi3-Interleaved** show significant improvements when multi-image SFT is applied.
> 2. **Combining multi-image SFT with pretraining yields better results**:
>    Models that incorporate multi-image data at both stages, such as **Mantis-idesfics2**, outperform those relying solely on SFT, such as **Mantis-SigCLIP**.
> 3. **Multi-image pretraining alone is insufficient**:
>    Models that only perform multi-image pretraining, like **XGen-MM-Phi3-Single** and **idefics2**, underperform compared to those that only apply multi-image SFT (e.g. **Mantis-SigCLIP**), highlighting the importance of multi-image SFT.
> 4. **Models exhibit some generalization to multi-image tasks without multi-image data**:
>    Models trained without multi-image data, such as **Mini-InternVL1.5**, still achieve reasonable performance on multi-image tasks, attributed to their strong visual encoders. However, their performance lags behind models with multi-image SFT.
>
> We provide a more detailed analysis in **Appendix B.3: Discussion on the Effectiveness of Multi-Image Training**.
>
> Thank you for your valuable feedback, which helped us deepen our analysis of multi-image training effectiveness.
>
> | **Model**                    | **Multi-image Pretrained** | **Multi-image SFT** | **Size** | **Semantic** | **Temporal** | **Spatial** | **Overall** |
> | ---------------------------- | -------------------------- | ------------------- | -------- | ------------ | ------------ | ----------- | ----------- |
> | Mini-InternVL1.5-chat        | No                         | No                  | 2B       | 30.4         | 36.2         | 26.5        | 30.5        |
> | Mini-InternVL1.5-chat        | No                         | No                  | 4B       | 32.2         | 36.8         | 28.5        | 32.1        |
> | InternVL1.5-chat             | No                         | No                  | 26B      | 33.4         | 45.2         | 35.5        | 37.4        |
> | LLaVA-Next                   | No                         | No                  | 7B       | 23.0         | 24.1         | 20.3        | 22.2       |
> | LLaVA-Next-Interleave        | No                         | Yes                 | 7B       | 35.7         | 32.2         | 31.1        | 32.4        |
> | Mantis-SigCLIP               | No                         | Yes                 | 8B       | 53.6         | 38.4         | 35.7        | 42.6        |
> | xGen-MM-Phi3-Single [1]      | Yes                        | No                  | 4B       | 25.1         | 25.7         | 23.1        | 24.5        |
> | idefics2                     | Yes                        | No                  | 8B       | 31.6         | 20.9         | 29.2        | 27.8        |
> | InternVL2                    | Yes                        | Only video          | 8B       | 35.2         | 39.5         | 31.2        | 34.8        |
> | Mantis-idsfics2              | Yes                        | Yes                 | 8B       | 56.3         | 40.8         | 39.2        | 45.6        |
> | XGen-MM-Phi3-Interleaved [1] | Yes                        | Yes                 | 4B       | 28.2         | 28.7         | 29.1        | 28.7        |
>
> ​		Table5: Performance comparison of various models with multi-image training data in different stage on MMIU.
>
>
>
> [1] xGen-MM (BLIP-3): A Family of Open Large Multimodal Models

---

> > ### Comment · Reviewer_bVMZ · 2024-11-25
> >
> > I appreciate the authors' effort in conducting the comprehensive experiments during rebuttal period. I strongly recommend the authors to add those experiments to the main paper in their final version. Given that the authors have addressed most of my concerns, I would like to keep my original rating. I hope to supplement some model results of real multi-image training or video training, such as Gemini-flash/Qwen2-VL, etc.

---

> ### Author Response · Authors · 2024-11-25
> **Response to Reviewer bVMZ**
>
> Thank you for your feedback. We have updated the latest version of the paper with additional content based on your suggestions, including the **Testmini settings** and **an analysis of the role of multi-image data at different training stages** (please refer to the **[Paper Update](https://openreview.net/forum?id=WsgEWL8i0K&noteId=XgIP7czmVN)** for specifics).
>
> For models that have undergone multi-image training, such as Gemini-Flash and Qwen2-VL, we have now tested over **30 LVLMs**, which provides a more comprehensive evaluation compared to other benchmarks. This includes models like **LLaVA-Interleave** and **InternVL2**, both of which have been trained with multi-image and video data. Additionally, we have included more recent models such as **XGen-MM**, **LLaVA-Interleave**, and **Mini-InternVL**. Absolutely, We plan to test even more models and provide convenient testing interfaces for the community to test on their own in the future.
>
> If you have any further questions, please don’t hesitate to let us know. We will make every effort to address them. And we kindly request that you consider updating the score accordingly. Thank you again for your valuable suggestions, which we believe have significantly enhanced the quality of the paper.

---

> ### Author Response · Authors · 2024-11-28
> **Response to Reviewer bVMZ**
>
> We would like to express our sincere gratitude for your insightful feedback. In fact, we have already conducted tests on **Gemini1.5Flash**, as well as on **InternVL2**, which is similar to **Qwen-VL2**. Both models have been extensively trained on **video data** and **interleaved image-and-text datasets**. The detailed results are summarized in **Table 1** below.
>
> Notably, We have evaluated **over 30 LVLMs and LLMs** on MMIU, compared to other benchmark papers, we think this is a fairly comprehensive evaluation. As for Gemini and InternVL: **Gemini1.5Flash**, **InternVL2-Pro**, and **InternVL2-8B** have achieved superior performance compared to other models, highlighting the importance of **multi-image training**. However, it is also important to note that these models still fall short when compared to **human scores** (referencing **[Response to Reviewer avPS (Q2)](https://openreview.net/forum?id=WsgEWL8i0K&noteId=mcGk8PFa6Z))**. This suggests that there is still significant room for improvement in the models' **multi-image capabilities**.
>
> In **Appendix B.3**, we provide a detailed analysis of both **architectural** and **data-related approaches** to enhance multi-image performance. Additionally, in **Appendix B.4**, we include a thorough **error case analysis**. We hope these findings can contribute to the development of more advanced multi-image models.
>
> Given the contributions of our work and the insights shared, we kindly ask if you might consider raising your score accordingly. If you have any further questions, we would be more than happy to provide additional clarifications.
>
>
>
> | Model          | Overall | Discrete | Continuous | Low-level | High-level-sub | High-level-obj | Two-D | Three-D |
> |----------------|---------|----------|------------|-----------|----------------|----------------|-------|---------|
> | Gemini1.5-Flash         | 54.5    | 54.2     | 50.1       | 76.1      | 63.9           | 64.9       | 43.3 | 43.0    |
> | InternVL2-Pro  | 49.8    | 53.8     | 46.3       | 72.7      | 70.6           | 58.5           | 38.1  | 42.1    |
> | InternVL2-8B   | 34.0    | 34.2     | 43.4       | 36.7      | 47.3           | 32.1           | 30.0  | 32.2    |
> | Human          | **92.8** | **89.5** | **92.8**   | **98.5**  | **96.6**       | **94.6**       | **88.0** | **92.0** |
>
> Table 1: Some model performance and human performance on MMIU

---

> ### Author Response · Authors · 2024-11-30
> **Looking forward to your reply !**
>
> As the discussion period is nearing its end, we would like to sincerely thank you for your valuable feedback. If you have any further questions, please do not hesitate to contact me immediately. We are fully committed to addressing any concerns you may have.

---

### Author Response · Authors · 2024-11-21
**Paper Update**

We deeply appreciate the time and effort invested by all reviewers in providing valuable feedback on our manuscript. Their insightful comments and suggestions have guided substantial improvements to our work. In the revised manuscript, we have employed a color-coding system (**blue, red, green, orange, and purple**) to highlight modifications addressing the specific comments from **Reviewers bVMZ, avPS, 4Ca4, JjzH, and g41s**, respectively. The comprehensive revisions have significantly enhanced the paper's clarity, depth, and overall quality.

The key changes included:

- Introduction of our proposed testmini test set for efficient evaluation in **Section 3**
- Addition of LLM experimental results in **Section 4.1**
- Clarification of MMIU's shuffle mechanism during construction to avoid position bias in **Section 4.1**
- Replacement of **Table 3** with a focused performance analysis across 7 types of image relationships, with the comprehensive task performance table moved to the appendix
- Enhancement of **Figure 4** by separating the TaskMap and model performance heatmap visualizations, with TaskMap relocated to the appendix for better focus
- Documentation of the MMIU data collection process in **Appendix A.4**, including metadata formats and distractor generation techniques
- Analysis of MMIU's independence through comparison with other benchmarks in **Appendix A.4**
- Expansion of experimental analyses in **Appendix B.3**, covering:
  - Performance analysis of testmini
  - Visual signal requirements in MMIU questions
  - Architectural impact on multi-image performance
  - Training stage effects of multi-image data introduction
  - Discussion of MMIU's practical applications
- Detailed error analysis across models and image relationships in **Appendix B.4**
- Addition of task cluster descriptions in **Appendix C Table 20**
- Analysis of the revised task map in **Appendix C.2**
- Inclusion of MMIU test overhead tables and analysis in **Appendix D**



These revisions have addressed the reviewers' concerns while strengthening the paper's contributions and methodological rigor. We believe these changes have markedly improved the manuscript's quality and clarity. We sincerely thank the reviewers for their constructive feedback that has helped shape this improved version of our work.

---

### Meta-Review · Area_Chair_u9i7 · 2024-12-21

**Metareview:**

The paper introduces a benchmark designed to evaluate VLMs across a wide range of multi-image tasks, including 7 types of multi-image relationships, 52 tasks, and 11K meticulously curated multiple-choice questions. With the proposed benchmark, the paper evaluates more than 30 existing VLM models, and analyzes their strengths and weaknesses in multi-image tasks.

MMIU provides a useful and comprehensive benchmark for multi-image understanding. Multi-image understanding is a very important capability for modern VLMs, but the corresponding comprehensive benchmarks are limited, especially the evaluation across diverse tasks and scenarios, e.g., semantic, temporal and spatial relations.

The reviews pointed out weaknesses in insufficient analysis, comparison, and reliability.
The authors responded to each point in details.
The final reviewers are all positive about the paper, with final scores of 6, 6, 6, 6, 6.

I recommend accepting this paper because the carefully designed benchmark for multi-image understanding meets current demands and the extensive analysis on a variety of VLM models is insightful and offers valuable insights into future research.

**Additional Comments On Reviewer Discussion:**

In the initial reviews, five reviewers raised concerns about insufficient analysis, comparison, and reliability of the proposed MMIU benchmark. Details are as below.

- Reviewer bVMZ: computation cost caused by many tasks, lack of practical significance, insufficient details about data processing, insufficient experiments on pure language models and models with interleave data or video data training.

- Reviewer avPS: poor writing and presentation of experimental results, potential position bias in the unanswerable set

- Reviewer 4Ca4: reliability of data generated by GPT-4o, limited discussion on practical applicability

- Reviewer JjzH: insufficient analysis in the aspects of architecture and training,  how and why certain models excel or struggle with specific relationship types, limited practical guidance for reducing the gap discussed in the benchmark.

- Reviewer g41s:  details of writing can be further polished, duplication between MMIU and existing multi-image benchmarks,  models without multi-image SFT outperform multi-image SFT models.

The authors provided detailed responses to each point, including revision of writing, more analysis from different aspects, additional discussion and comparison.

Reviewer bVMZ and avPS recognized the rebuttal and keep the scores of 6.
Reviewer 4Ca4, JjzH, g41s didn't make clear response and maintain the original scores of 6.

The final scores are all marginally above the acceptance threshold: 6, 6, 6, 6, 6.

---

### Decision · Program_Chairs · 2025-01-22

Accept (Poster)